# Variance-aware decision making with linear function approximation under heavy-tailed rewards

**Xiang Li**                                                        *lx10077@pku.edu.cn*
*School of Mathematical Sciences*
*Peking University*

**Qiang Sun**                                                       *qiang.sun@utoronto.ca*
*Department of Statistical Sciences*
*University of Toronto*

**Reviewed on OpenReview:** *https://openreview.net/forum?id=8bnsoL2IyJ&noteId=mUmJtQzBhL*

## Abstract

This paper studies how to achieve variance-aware regrets for online decision-making in the presence of heavy-tailed rewards with only finite variances. For linear stochastic bandits, we address the issue of heavy-tailed rewards by modifying the adaptive Huber regression and proposing AdaOFUL. AdaOFUL achieves a state-of-the-art regret bound of $\widetilde{\mathcal{O}}\big(d\big(\sum_{t=1}^{T}\nu_t^2\big)^{1/2}+d\big)$[1] as if the rewards were uniformly bounded, where $\nu_t^2$ is the conditional variance of the reward at round $t$, $d$ is the feature dimension, and $T$ is number of online rounds. Building upon AdaOFUL, we propose VARA for linear MDPs, which achieves a variance-aware regret bound of $\widetilde{\mathcal{O}}(d\sqrt{H\mathcal{G}^*K})$. Here, $H$ is the length of episodes, $K$ is the number of episodes, and $\mathcal{G}^*$ is a smaller instance-dependent quantity that can be bounded by other instance-dependent quantities when additional structural conditions on the MDP are satisfied. Overall, our modified adaptive Huber regression algorithm may serve as a useful building block in the design of algorithms for online problems with heavy-tailed rewards.

## 1 Introduction

In many real-world scenarios, data exhibit heavy-tailed behaviors, which deviate significantly from classical assumptions in statistical analyses. Examples include stock returns in financial markets (Cont, 2001; Hull, 2012), microarray data analysis (Posekany et al., 2011), and advertiser values in online advertising (Arnosti et al., 2016). Such heavy-tailed distributions pose challenges to conventional algorithmic designs that often hinge upon uniformly bounded or sub-Gaussian reward assumptions.

Many works studying decision-making under uncertainty focus on the multi-arm bandit problem and its extension, the linear bandits. Regret analysis in this domain seeks to understand the suboptimality of algorithmic choices. However, traditional analyses often limit their applicability by assuming uniformly bounded or sub-Gaussian rewards. Some recent approaches address heavy-tailed behaviors by truncating rewards to achieve sub-linear worst-case regret bounds (Bubeck et al., 2013; Medina & Yang, 2016; Shao et al., 2018; Xue et al., 2021). Nevertheless, these truncation-based methods encounter estimation errors dependent on absolute moments of observations, not their central moments, suggesting suboptimality, especially in noiseless situations.

While linear bandits offer a general enough setting for understanding decision-making with heavy-tailed data, reinforcement learning (RL) elevates this understanding to long-horizon decision-making processes. In RL, agents not only make decisions but also navigate through potentially infinite state and action spaces

---

[1] $\widetilde{\mathcal{O}}$ hides constant factors and logarithmic dependence on $T$.

over a given horizon (Sutton & Barto, 2018). RL demonstrates remarkable empirical successes in various applications, including robotics (Lillicrap et al., 2015), dialogue systems (Li et al., 2016), and Go play (Silver et al., 2016). Recent theoretical advances have expanded RL's applicability, especially with linear function approximation in the context of linear Markov decision processes (MDPs) (Yang & Wang, 2020; Jin et al., 2020b;a; Wagenmaker et al., 2022b; Zanette et al., 2020; Ayoub et al., 2020; Zhou et al., 2021). More recently, the shift from worst-case regret analysis to variance-aware regret analysis in RL offers more nuanced insights into agent performance (Pananjady & Wainwright, 2020; Khamaru et al., 2021; Li et al., 2023; Yin & Wang, 2021; Min et al., 2021). More specifically, variance-aware regrets depend on the variances of rewards and value functions and provide finer guarantees than worst-case bounds by characterizing problem-dependent performances across different problem instances. Yet, the challenge posed by heavy-tailed rewards remains.

This paper explores the intersection of decision-making under heavy-tailed rewards, ranging from linear bandits to RL applications. We use the term *heavy-tailed rewards* throughout the paper to refer to the rewards that have only finite variances. We aim to achieve the variance-awareness and address the heavy-tailed issue simultaneously. A desirable algorithm should have the following two properties. First, it should possess the flexibility to function as a module, enhancing algorithms originally designed for bounded rewards to accommodate heavy-tailed rewards. Second, it should attain tight variance-aware regret bounds based on central moments, rather than absolute moments.

## 1.1 Our contributions

We provide a particular algorithm satisfying the mentioned characteristics. Our solution is motivated by adaptive Huber regression (Sun et al., 2020; Sun, 2021), which was originally proposed for analyzing offline independently and identically distributed (i.i.d.) data. It uses the (pseudo-) Huber loss to estimate the unknown coefficient with a universal robustification parameter. We adapt this method for online bandits and carefully choose different robustification parameters to handle non-i.i.d. data. The resulting algorithm, called AdaOFUL, short for Adaptive Huber regression based OFUL, achieves the state-of-the-art regret bound $\widetilde{\mathcal{O}}\left(d\sqrt{\sum_{t\in[T]}\nu_t^2}+d\right)$ for linear bandits with heavy-tailed rewards, where $\nu_t^2$ is the observed conditional variance of the random reward at step $t$ and $d$ is the feature dimension. Here $\widetilde{\mathcal{O}}(\cdot)$ hides constant factors and logarithmic dependence on $T$. Such a variance-aware regret bound has only been obtained in the literature of linear bandits with sub-Gaussian or uniformly bounded rewards (Kirschner & Krause, 2018; Zhou & Gu, 2022). In contrast, truncation-based methods are suboptimal due to their estimation errors that depend on absolute moments instead of central moments. For example, the truncation-based algorithms from (Shao et al., 2018; Xue et al., 2021) yield regret in the form of $\widetilde{\mathcal{O}}\left(vd\sqrt{T}\right)$ where $v^2$ is the bound for the second moment of random rewards. Our regret bound depends on the central moment instead and is thus tighter.

Using AdaOFUL as a building block, we then propose the Variance-Aware Regret via the Adatptive Huber regression (VARA) algorithm for linear MDPs with heavy-tailed rewards. In essence, VARA integrates AdaOFUL with the state-of-the-art worst-case algorithm LSVI-UCB++ from (He et al., 2022), enhancing regret performance through more careful analysis and resulting in a regret bound of $\widetilde{\mathcal{O}}(d\sqrt{H\mathcal{G}^*K})$. Here $H$ is the horizon length and $\mathcal{G}^*$ is a variance-aware quantity bounded by the sum of weighted per-step conditional variances. Our regret bound is superior to the current state-of-the-art bounds in three ways. First, it depends on a tighter instance-dependent quantity $\mathcal{G}^*$ without knowing the value of $\mathcal{G}^*$ in advance and has optimal dependence on $d$ and $H$. Second, assuming additional structural conditions on the underlying MDP, we can obtain further instance-dependent bounds of $\mathcal{G}^*$, including range-dependent, first-order, and concentrability-dependent bounds. Third, our regret bound $\widetilde{\mathcal{O}}(d\sqrt{H\mathcal{G}^*K})$ is valid even when rewards have only finite variances, which achieves a level of generality that is unmatched by previous works.

Our findings indicate that heavy-tailed rewards do not pose a limitation for developing online decision-making with linear function approximations. Our proposed modified adaptive Huber regression algorithm can be used as a general approach to adapt existing online algorithms designed for light-tailed rewards to handle heavy-tailed ones while maintaining tight dependence on variance for regret bounds.

**Overview** The rest of the paper proceeds as follows. We state our main results for heavy-tailed linear bandits in Section 2 and for linear MDPs in Section 3. We review related work in Section 4 and conclude in Section 5. Most proofs are collected in the appendix.

**Notation** We use $\|\cdot\|$ to denote the $\ell_2$-norm in $\mathbb{R}^d$, and $\mathrm{Ball}_d(B)$ the $\ell_2$-norm ball in $\mathbb{R}^d$ with radius $B > 0$. For a positive definite matrix $\boldsymbol{H} \in \mathbb{R}^{d \times d}$, $\|\boldsymbol{x}\|_{\boldsymbol{H}} = \sqrt{\boldsymbol{x}^\top \boldsymbol{H} \boldsymbol{x}}$ for a vector $\boldsymbol{x} \in \mathbb{R}^d$. For two semidefinite positive matrices $\boldsymbol{H}_1, \boldsymbol{H}_2$, we denote $\boldsymbol{H}_1 \succeq \boldsymbol{H}_2$ if $\boldsymbol{H}_2 - \boldsymbol{H}_1$ is semidefinite positive. For an integer $K \in \mathbb{N}^+$, let $[K] := \{1, 2, \cdots, K\}$. For a set $\mathcal{A}$, $|\mathcal{A}|$ denotes its cardinality. For real numbers $a \leqslant b$ and $x \in \mathbb{R}$, we use $x_{[a,b]} := \max\{a, \min\{x, b\}\}$ to denote the projection of $x$ onto the closed interval $[a, b]$.

## 2 Variance-aware Regret for Heavy-tailed Linear Bandits

In this section, we first introduce the heavy-tailed linear bandit and then present the AdaOFUL algorithm, showing it achieves state-of-the-art variance-aware regret even when faced with heavy-tailed rewards.

### 2.1 Heavy-tailed Stochastic Linear Bandit

**Definition 2.1** (Heavy-tailed stochastic linear bandit)**.** Let $\{\mathcal{D}_t\}_{t \geqslant 1}$ denote a fixed sequence of decision sets and $\{\mathcal{F}_t\}_{t \geqslant 1}$ a filtration. At round $t$, the agent chooses $\boldsymbol{\phi}_t \in \mathcal{D}_t$ and then observes the reward $y_t$ and its conditional variance $\nu_t^2$. We assume $y_t = \langle \boldsymbol{\phi}_t, \boldsymbol{\theta}^* \rangle + \varepsilon_t$ where $\boldsymbol{\theta}^* \in \mathbb{R}^d$ is a vector unknown to the agent and $\varepsilon_t \in \mathbb{R}$ is a martingale difference random noise such that $\mathbb{E}[\varepsilon_t | \mathcal{F}_{t-1}] = 0$ and $\mathbb{E}[\varepsilon_t^2 | \mathcal{F}_{t-1}] = \nu_t^2$. Both $\nu_t$ and $\boldsymbol{\phi}_t$ are $\mathcal{F}_{t-1}$-measurable and $\|\boldsymbol{\phi}_t\| \leqslant L$. We assume $\|\boldsymbol{\theta}^*\| \leqslant B$ with $B$ known *a priori*. The agent aims to minimize the regret, formally defined as

$$\mathrm{Reg}(T) := \sum_{t=1}^{T} \left[ \sup_{\boldsymbol{\phi} \in \mathcal{D}_t} \langle \boldsymbol{\phi}, \boldsymbol{\theta}^* \rangle - \langle \boldsymbol{\phi}_t, \boldsymbol{\theta}^* \rangle \right]. \tag{2.1}$$

In heavy-tailed stochastic linear bandits, the mean-zero random noises $\varepsilon_t$ have only bounded variances. We emphasize that in linear bandits, data are collected adaptively, and therefore, the distribution of $\varepsilon_t$ depends on $\boldsymbol{\phi}_t$. Moreover, the choice of $\boldsymbol{\phi}_t$ depends on all past observations $(\boldsymbol{\phi}_s, y_s, \nu_s)_{s < t}$.

### 2.2 Algorithm Description

This section presents the AdaOFUL algorithm for heavy-tailed linear bandits. The AdaOFUL algorithm is given in Algorithm 1. AdaOFUL follows the principle of Optimism in the Face of Uncertainty (OFU) (Abbasi-Yadkori et al., 2011) to solve the heavy-tailed heterogeneous linear bandit problem. At each round $t$, it maintains a confidence set defined in equation 2.2 such that $\boldsymbol{\theta}^* \in \mathcal{C}_t$ uniformly for all $t \geqslant 1$ with high probability when the exploration radius $\beta_{t-1}$ is properly chosen. Unlike the standard OFUL algorithm (Abbasi-Yadkori et al., 2011) which directly selects the most optimistic estimator $\widetilde{\boldsymbol{\theta}}_t$ to make an arm selection $\boldsymbol{\phi}_t$, AdaOFUL uses adaptive Huber regression to compute a new estimator $\boldsymbol{\theta}_t$ that takes into account the heavy-tailed rewards. The agent then selects the arm $\boldsymbol{\phi}_t$ that maximizes the inner product $\langle \boldsymbol{\phi}, \boldsymbol{\theta} \rangle$ over $\boldsymbol{\theta} \in \mathcal{C}_{t-1}$. After playing the selected arm, the agent observes the reward $y_t$ and its conditional variance $\nu_t$. The last step of round $t$ updates the exploration radius $\beta_t$ and the shape matrix $\boldsymbol{H}_t$ for the confidence set construction.

**Adaptive pseudo-Huber regression** The pseudo-Huber loss (Hastie et al., 2009; Sun, 2021) is defined as

$$\ell_\tau(x) = \tau(\sqrt{\tau^2 + x^2} - \tau), \tag{2.5}$$

which is a smooth approximation to the well-known Huber loss (Huber, 1964). Similar to the Huber loss, the pseudo-Huber loss resembles a quadratic function for small values of $|x|$ and is approximately linear when $x$ is large in magnitude, making the loss strongly convex when close to the origin and less sensitive to changes in the tails. The parameter $\tau$ controls the balance between the quadratic and linear regions and is referred

---

**Algorithm 1** Adaptive Huber regression based OFUL (AdaOFUL)

---

**Initialization:** $\boldsymbol{H}_0 = \lambda \boldsymbol{I}, \boldsymbol{\theta}_0 = \boldsymbol{0}, \beta_0 = \sqrt{\lambda} B, c_0 = \frac{1}{6\sqrt{3\log\frac{2T^2}{\delta}}}, c_1 = \frac{1}{42 \cdot \log\frac{2T^2}{\delta}}, \sigma_{\min} = \frac{1}{\sqrt{T}}.$

---

**1 for** $t = 1$ **to** $T$ **do**

**2**     Construct the confidence set $\mathcal{C}_{t-1}$ as in

$$\mathcal{C}_{t-1} := \left\{ \boldsymbol{\theta} \in \mathrm{Ball}_d(B) : \|\boldsymbol{\theta} - \boldsymbol{\theta}_{t-1}\|_{\boldsymbol{H}_{t-1}} \leqslant \beta_{t-1} \right\}. \tag{2.2}$$

**3**     Solve $(\boldsymbol{\phi}_t, \cdot) = \mathrm{argmax}_{\boldsymbol{\phi} \in \mathcal{D}_t, \boldsymbol{\theta} \in \mathcal{C}_{t-1}} \langle \boldsymbol{\phi}, \boldsymbol{\theta} \rangle.$

**4**     Play $\boldsymbol{\phi}_t$ and observe $(y_t, \nu_t)$.

**5**     Set $\sigma_t, w_t$ and $\tau_t$ according to the following equation and record $\{\sigma_s, w_s, \tau_s : 1 \leqslant s \leqslant t\}$.

$$\sigma_t = \max\left\{\nu_t, \sigma_{\min}, \frac{\|\boldsymbol{\phi}_t\|_{\boldsymbol{H}_{t-1}^{-1}}}{c_0}, \frac{\sqrt{LB}\|\boldsymbol{\phi}_t\|_{\boldsymbol{H}_{t-1}^{-1}}^{\frac{1}{2}}}{c_1^{\frac{1}{4}} d^{\frac{1}{4}}}\right\}, w_t = \left\|\frac{\boldsymbol{\phi}_t}{\sigma_t}\right\|_{\boldsymbol{H}_{t-1}^{-1}}, \ \tau_t = \tau_0 \frac{\sqrt{1 + w_t^2}}{w_t}. \tag{2.3}$$

**6**     Compute $\boldsymbol{\theta}_t$ by minimizing the following convex problem

$$\boldsymbol{\theta}_t := \underset{\boldsymbol{\theta} \in \mathrm{Ball}_d(B)}{\mathrm{argmin}} \ L_t(\boldsymbol{\theta}) \text{ with } L_t(\boldsymbol{\theta}) := \frac{\lambda}{2}\|\boldsymbol{\theta}\|^2 + \sum_{s=1}^{t} \ell_{\tau_s}\left(\frac{y_s - \langle \boldsymbol{\phi}_s, \boldsymbol{\theta} \rangle}{\sigma_s}\right). \tag{2.4}$$

**7**     Define the confidence set radius $\beta_t$ as in equation 2.6 and set $\boldsymbol{H}_t = \boldsymbol{H}_{t-1} + \frac{\boldsymbol{\phi}_t}{\sigma_t}\frac{\boldsymbol{\phi}_t^\top}{\sigma_t}.$

**8 end**

---

to as the robustification parameter by Sun et al. (2020) in the case of the Huber loss. Since the value of the robustification parameter needs to be adaptive to the data for an optimal tradeoff between robustness and unbiasedness, we shall also refer to the pseudo-Huber regression with a data-adaptive $\tau$ as adaptive pseudo-Huber regression or simply adaptive Huber regression, in line with Sun et al. (2020).

To compute the pseudo-Huber estimator $\boldsymbol{\theta}_t$ for $\boldsymbol{\theta}^*$, given the history $\{(\boldsymbol{\phi}_s, y_s, \nu_s)\}_{s \in [t]}$ up to time $t$, we solve the the convex optimization problem in equation 2.4 (Sun, 2021). Recall that $\sigma_t$'s are surrogate conditional variances, and $\tau_t$'s are the robustification parameters, given by equation 2.3, in which $\sigma_{\min}$ is a small positive constant to avoid singularity, $\tau_0$ is a hyper-parameter, $w_t$'s are importance measures, $c_0$ and $c_1$ are specified in Algorithm 1, and $L$ and $B$ are constants defined in Definition 2.1.

**Robustification parameter** As shown in equation 2.3, the robustification parameter $\tau_t$ is set differently for each data point $(\boldsymbol{\phi}_t, y_t, \nu_t)$ in the pseudo-Huber regression. This is a significant departure from the case of i.i.d. data, where all robustification parameters are typically set to the same value $\tau$, as i.i.d. data are naturally weighted equally (Sun et al., 2020). In linear bandits, the data are generated adaptively, where the choice of $\boldsymbol{\phi}_t$ can depend on all past observations. Since observations collected in later rounds are less important as they are based on previous observations and contribute less to the estimation accuracy, we assign greater weight to earlier observations. To measure the importance of the $t$-th observation, we use $w_t = \|\boldsymbol{\phi}_t\|_{\boldsymbol{H}_{t-1}^{-1}} / \sigma_t$ as the importance measure for the $t$-th observation and set $\tau_t = \tau_0 \sqrt{1 + w_t^2}/w_t$ as the corresponding robustification parameter.

When taking $\tau_0 = \infty$, the optimization problem in equation 2.4 reduces to weighted regularized least-squares, which has been proven to achieve worst-case optimality for linear bandits with uniformly bounded or sub-Gaussian rewards (Kirschner & Krause, 2018; Zhou & Gu, 2022). However, an appropriate value of $\tau_0$ is necessary to balance robustness against heavy-tailed rewards and asymptotic unbiasedness. In Corollary 2.1, we will demonstrate that setting $\tau_0 = \tilde{\mathcal{O}}(\sqrt{d})$ is sufficient to achieve the state-of-the-art regret bound.

**Variance estimates**  We choose $\sigma_t \geqslant \sqrt{LB}\|\phi_t\|_{\boldsymbol{H}_{t-1}^{-1}}^{\frac{1}{2}}/(c_1^{\frac{1}{4}}d^{\frac{1}{4}})$, which implies $c_1 d \geqslant L^2 B^2 w_t^2/(\sigma_t^2)$. This condition is used to lower bound the Hessian matrix $\nabla^2 L_T(\boldsymbol{\theta})$. For any $\boldsymbol{\theta} \in \text{Ball}_d(B)$, we expect $\nabla^2 L_T(\boldsymbol{\theta}) \approx \boldsymbol{H}_T$ up to universal constant factors to proceed with theoretical analysis. A direct computation yields $\nabla^2 L_T(\boldsymbol{\theta}) \preceq \boldsymbol{H}_T$, while for the other direction we show $\nabla^2 L_T(\boldsymbol{\theta}) \succeq \left(c - \sup_{t\in[T]}|\langle\phi_t, \boldsymbol{\theta}^* - \boldsymbol{\theta}\rangle/(\tau_t\sigma_t)|^2\right)\boldsymbol{H}_T$ for some universal constant $c > 0$ with high probability. With the last condition on $\sigma_t$, for any feasible solution $\boldsymbol{\theta} \in \text{Ball}_d(B)$, the following quantity

$$\left|\frac{\langle\phi_t, \boldsymbol{\theta}^* - \boldsymbol{\theta}\rangle}{\tau_t\sigma_t}\right|^2 \leqslant \frac{\|\phi_t\|^2\|\boldsymbol{\theta}^* - \boldsymbol{\theta}\|^2}{\tau_t^2\sigma_t^2} \leqslant \frac{4w_t^2 L^2 B^2}{\tau_0^2\sigma_t^2} \leqslant \frac{4c_1 d}{\tau_0^2}$$

can be sufficiently small provided that $\tau_0^2 \geqslant c \cdot d$ for a sufficiently large constant $c > 0$.

## 2.3  Regret Analysis

We first validate that the optimism holds with high probability in Theorem 2.1 and then establish a high probability bound for the regret in Theorem 2.2.

**Theorem 2.1.** Let $\kappa = d \cdot \log\left(1 + TL^2/(d\lambda\sigma_{\min}^2)\right)$. For the heavy-tailed linear bandit in Definition 2.1, if $\tau_0\sqrt{\log(2T^2/\delta)} \geqslant \max\{\sqrt{2\kappa}, 2\sqrt{d}LB\}$, then with probability at least $1 - 4\delta$, it holds that, for all $0 \leqslant t \leqslant T$,

$$\|\boldsymbol{\theta}_t - \boldsymbol{\theta}^*\|_{\boldsymbol{H}_t} \leqslant \beta_t,$$

where

$$\beta_t = 32\left(\frac{\kappa}{\tau_0} + \sqrt{\kappa\log\frac{2t^2}{\delta}} + \tau_0\log\frac{2t^2}{\delta}\right) + 5\sqrt{\lambda}B. \tag{2.6}$$

Theorem 2.1 establishes that $\boldsymbol{\theta}^*$ is contained in the set $\mathcal{C}_t := \{\boldsymbol{\theta} \in \text{Ball}_d(B) : \|\boldsymbol{\theta} - \boldsymbol{\theta}_t\|_{\boldsymbol{H}_t} \leqslant \beta_t\}$ for all $t \geqslant 0$ with high probability. It is proved by using Bernstein-type concentration inequality for self-normalized vector-valued martingales with additional care paid to deal with heavy-tailed rewards. See the next subsection for a proof sketch.

**Theorem 2.2.** Let $\sigma_{\min} = 1/\sqrt{T}$. Then with probability at least $1 - 4\delta$, we have

$$\text{Reg}(T) \leqslant 2\beta_T \cdot \left[\sqrt{2\kappa} \cdot \sqrt{\sum_{t=1}^T \nu_t^2 + 1} + \frac{2L\kappa}{c_0^2\sqrt{\lambda}} + \frac{2LB\kappa}{\sqrt{c_1 d}}\right]$$

where $\beta_T$ is defined in equation 2.6, and $c_0, c_1 = \tilde{\mathcal{O}}(1)$ are positive constants given in Algorithm 1.

Theorem 2.2 provides a regret bound in a general form that depends on $\beta_T$. As shown in equation 2.6, $\beta_t$ is a hyperbolic function of the robustification parameter $\tau_0$. Increasing $\tau_0$ decreases the bias term $\mathcal{O}(\kappa/\tau_0)$ while increasing the range term $\mathcal{O}(\tau_0\log(2t^2/\delta))$. Therefore, choosing $\tau_0$ carefully is essential to achieve the optimal trade-off between unbiasedness and robustness. Setting $\tau_0 = \tilde{\mathcal{O}}(\sqrt{d})$ minimizes the right-hand side of equation 2.6. This, combined with Theorem 2.2, yields the simplified regret bound equation 2.7 in the following corollary.

**Corollary 2.1.** Let $\tau_0 = \max\{\sqrt{2\kappa}, 2\sqrt{d}\}/\sqrt{\log(2T^2/\delta)}$ and $\lambda = d/B^2$, then $\beta_T \leqslant 64\left(2\sqrt{\kappa\log(2T^2/\delta)} + \sqrt{d\log(2T^2/\delta)}\right) + 5\sqrt{d}$. Consequently, the regret bound in Theorem 2.2 becomes

$$\text{Reg}(T) = \tilde{\mathcal{O}}\left(d\sqrt{\sum_{t\in[T]}\nu_t^2} + d \cdot \max\{LB, 1\}\right), \tag{2.7}$$

where $\tilde{\mathcal{O}}(\cdot)$ hides constant factors and logarithmic dependence on $T$.

Corollary 2.1 demonstrates that AdaOFUL achieves state-of-the-art regret bound under heavy-tailed rewards, comparable to the case where rewards are uniformly bounded or sub-Gaussian. The regret upper bound in the noiseless case reduces to $\tilde{\mathcal{O}}(d)$, and in the noisy case, it reduces to $\tilde{\mathcal{O}}(d\sqrt{\sum_{t\in[T]}\nu_t^2})$. In the worst case scenario where $\nu_t = \Theta(1)$ for all $t \geqslant 1$, the regret bound reduces to $\tilde{\mathcal{O}}(d\sqrt{T})$, which matches the worst-case minimax lower bound (Dani et al., 2008). Hence, our variance-aware regret bound equation 2.7 is tighter than the pessimistic worst-case bound $\tilde{\mathcal{O}}(d\sqrt{T})$ when $\sum_{t=1}^{T}\nu_t^2 \ll T$. To the best of our knowledge, such a variance-aware regret bound has only been obtained in the literature for sub-Gaussian rewards (Kirschner & Krause, 2018) or uniformly bounded rewards (Zhou & Gu, 2022). We are the first to provide a variance-aware regret bound for heavy-tailed stochastic linear bandits.

### 2.4 Proof Sketch of Theorem 2.1

**Step one: Hessian approximation**   Let $z_t(\boldsymbol{\theta}) := \frac{y_t - \langle \phi_t, \boldsymbol{\theta} \rangle}{\sigma_t}$ and $\kappa := d\log\left(1 + \frac{TL^2}{d\lambda\sigma_{\min}^2}\right)$ for simplicity.

**Lemma 2.1.** Assume there exists a constant $b > 0$ such that $\mathbb{E}[z_t^2(\boldsymbol{\theta}^*)|\mathcal{F}_{t-1}] \leqslant b^2$ for all $t \geqslant 1$. If $\tau_0\sqrt{\log\frac{2T^2}{\delta}} \geqslant \max\{\sqrt{2\kappa}b, 2\sqrt{d}\}$, then with probability at least $1 - 2\delta$, for all $T \geqslant 0$ and any $\|\boldsymbol{\theta}\| \leqslant B$,

$$\frac{1}{4}\boldsymbol{H}_T \preceq \nabla^2 L_T(\boldsymbol{\theta}) \preceq \boldsymbol{H}_T.$$

Lemma 2.1 shows that with high probability and up to constant factors, $\nabla^2 L_T(\boldsymbol{\theta})$ approximates $\boldsymbol{H}_T$ well uniformly for all $T \geqslant 1$ and $\|\boldsymbol{\theta}\| \leqslant B$. By contrast, in standard ridge regressions, $\nabla^2 L_T(\boldsymbol{\theta})$ equals to $\boldsymbol{H}_T$ because the corresponding loss $L_T$ is quadratic. The proof is deferred to Section C.2.

**Step two: High probability gradient bound**   In the following, we provide a high-probability bound for $\|\nabla L_\tau(\boldsymbol{\theta}^*)\|_{\boldsymbol{H}_T^{-1}}$ in Lemma 2.2.

**Lemma 2.2.** Assume there exists a constant $b > 0$ such that $\mathbb{E}[z_t^2(\boldsymbol{\theta}^*)|\mathcal{F}_{t-1}] \leqslant b^2$ for all $t \geqslant 1$. With probability at least $1 - 2\delta$, for all $T \geqslant 1$, it follows that

$$\|\nabla L_T(\boldsymbol{\theta}^*)\|_{\boldsymbol{H}_T^{-1}} \leqslant 8\bigg[ \underbrace{\frac{\kappa b^2}{\tau_0}}_{\text{bias term}} + \underbrace{\sqrt{\kappa b^2 \log\frac{2T^2}{\delta}}}_{\text{variance term}} + \underbrace{\tau_0 \log\frac{2T^2}{\delta}}_{\text{range term}} \bigg] + \underbrace{\sqrt{\lambda}B}_{\text{ridge term}}.$$

We explain briefly about each term in Lemma 2.2. Following Zhou et al. (2021), a decomposition follows that $\|\nabla L_t(\boldsymbol{\theta}^*)\|_{\boldsymbol{H}_t^{-1}}^2 = \sum_{t=1}^{T}(X_t + Y_t)$ for two sequences of random variables $X_t, Y_t \in \mathcal{F}_t$. To illustrate the proof idea, we explain how to bound $\sum_{t=1}^{T} X_t$, since $\sum_{t=1}^{T} Y_t$ can be bounded similarly. For the adaptive Huber regression, $\{X_t\}_{t\in[T]}$ is not a martingale difference sequence but $\{X_t - \mathbb{E}[X_t|\mathcal{F}_{t-1}]\}_{t\in[T]}$ is. We apply a Bernstein inequality to upper bound $\sum_{t=1}^{T}(X_t - \mathbb{E}[X_t|\mathcal{F}_{t-1}])$ which contributes to the variance and range terms. Thanks to the different robustification parameters $\tau_t$, we can control $\sum_{t=1}^{T}\mathbb{E}[X_t|\mathcal{F}_{t-1}]$ deterministically within $\mathcal{O}\left(\kappa^2/\tau_0^2\right)$, resulting in the bias term. Finally, the last ridge term $\sqrt{\lambda}B$ exists because we use ridge regularization to ensure that the Hessian is always invertible. The detailed proof is in Appendix C.3.

**Step three: Combination through stationary condition**   Notice that the gradient is given by

$$\nabla L_T(\boldsymbol{\theta}) := \lambda\boldsymbol{\theta} - \sum_{t=1}^{T} \frac{\tau_t z_t(\boldsymbol{\theta})}{\sqrt{\tau_t^2 + z_t(\boldsymbol{\theta})^2}} \frac{\phi_t}{\sigma_t},$$

and our estimator $\boldsymbol{\theta}_T$ is the minimizer of a constrained problem in equation 2.4. By Proposition 1.3 in (Bubeck et al., 2015), the first-order stationary condition of the constrained convex optimization equation 2.4 implies that $\langle \nabla L_T(\boldsymbol{\theta}_T), \boldsymbol{\theta}_T - \boldsymbol{\theta} \rangle \leqslant 0$ for all $\boldsymbol{\theta} \in \text{Ball}_d(B)$. More specifically, due to $\|\boldsymbol{\theta}^*\| \leqslant B$, we have

$$\langle \nabla L_T(\boldsymbol{\theta}_T), \boldsymbol{\theta}_T - \boldsymbol{\theta}^* \rangle \leqslant 0. \tag{2.8}$$

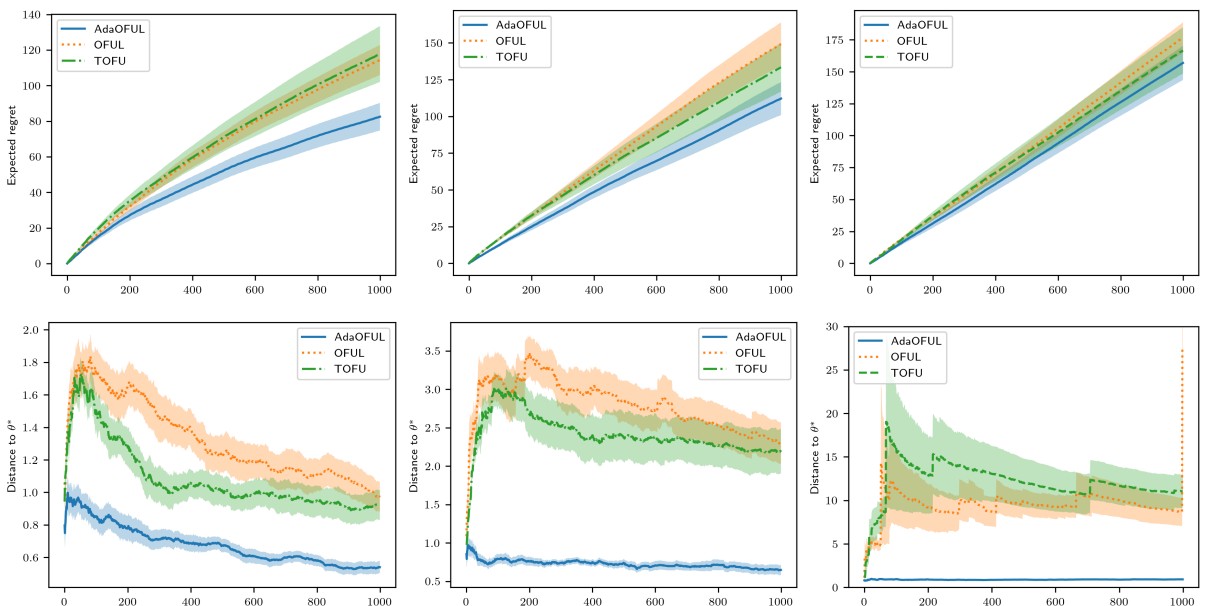

Figure 1: Regret and convergence results across three noise types: Case (a) $\varepsilon_t \sim \mathcal{N}(0,1)$ on the left, Case (b) $\varepsilon_t \sim \mathrm{t}(\mathrm{df})$ with df $\sim \mathcal{U}(2,3)$ in the middle, and Case (c) $\varepsilon_t \sim \mathrm{t}(\mathrm{df})$ with df $\sim \mathcal{U}(1,2)$ on the right.

By the mean value theorem for vector-valued functions, we have

$$\nabla L_T(\boldsymbol{\theta}_T) - \nabla L_T(\boldsymbol{\theta}^*) = \int_0^1 \nabla^2 L_T((1-\eta)\boldsymbol{\theta}^* + \eta\boldsymbol{\theta}_T)\mathrm{d}\eta \cdot (\boldsymbol{\theta}_T - \boldsymbol{\theta}^*).$$

Using Lemma 2.1 and the fact that $\|(1-\eta)\boldsymbol{\theta}^* + \eta\boldsymbol{\theta}_T\| \leqslant B$ for all $\eta \in [0,1]$, we have

$$\frac{1}{4}\|\boldsymbol{\theta}_T - \boldsymbol{\theta}^*\|_{\boldsymbol{H}_T}^2 \leqslant \langle \boldsymbol{\theta}_T - \boldsymbol{\theta}^*, \nabla L_T(\boldsymbol{\theta}_T) - \nabla L_T(\boldsymbol{\theta}^*)\rangle. \tag{2.9}$$

By equation 2.9 and equation 2.8, we have

$$\|\boldsymbol{\theta}_T - \boldsymbol{\theta}^*\|_{\boldsymbol{H}_T} \leqslant 4\|\nabla L_T(\boldsymbol{\theta}^*)\|_{\boldsymbol{H}_T^{-1}}. \tag{2.10}$$

Combining equation 2.10, Lemma 2.1 and Lemma 2.2, we know that if $\tau_0\sqrt{\log\frac{2T^2}{\delta}} = \max\{\sqrt{2\kappa}b, 2\sqrt{d}\}$, with probability at least $1 - 2\delta$, we have that $\|\boldsymbol{\theta}_t - \boldsymbol{\theta}^*\|_{\boldsymbol{H}_t} < \beta_t$ for all $1 \leqslant t \leqslant T$ where $\beta_t$ is given in equation 2.6. It implies that $\boldsymbol{\theta}^*$ indeed locals in all constructed confidence regimes, i.e., for all $1 \leqslant t \leqslant T$, $\boldsymbol{\theta}^* \in \mathcal{C}_t$. Notice that by the choice of $\beta_0$ and $\boldsymbol{H}_0$, we still have $\boldsymbol{\theta}^* \in \mathcal{C}_0$. Finally, $b = 1$ in our case completes the proof.

## 2.5 A Numerical Study

**Considered methods** In this subsection, we conduct a numerical comparison between AdaOFUL and two baseline algorithms: original OFUL (Abbasi-Yadkori et al., 2011) and TOFU (Shao et al., 2018). TOFU is a truncation-based variant of OFUL, designed to address the heavy-tail issue. Because these algorithms do not consider the variance information, for a fair comparison, we abstain from the variance weights and set each $\sigma_t \equiv 1$. Hyperparameters were chosen based on observations from the initial couple of steps so that $\tau_0 = \sqrt{d}$ and $c_0 = c_1 = 1$.

**Experiment setup** We experiment with the following configuration. We set $d = 10$ and $|\mathcal{D}_t| \equiv 20$. The optimal $\boldsymbol{\theta}^*$ is generated by randomly sampling each coordinate from a uniform distribution $\mathcal{U}(0,1)$ and normalizing the resultant vector to unit length so that $B = 1$. To simulate varying action sets, we generate 20 distinct basic action sets, $\{\mathcal{A}_i\}_{i \in [20]}$, and assign $\mathcal{D}_t = \mathcal{A}_i$ if $t = i \mod 20$. For each $\mathcal{A}_i$, each arm vector $\boldsymbol{\phi} \in \mathcal{A}_i$ is formed in the same way as $\boldsymbol{\theta}^*$ so that $L = 1$. Rewards are generated by $y_t = \langle \boldsymbol{\phi}_t, \boldsymbol{\theta}^*\rangle + \varepsilon_t$ with $\varepsilon_t$ being an independent zero-mean noise. We investigate three noise types: Case (a) is Gaussian distribution

$\varepsilon_t \sim \mathcal{N}(0, 1)$, while Cases (b) and (c) correspond to Student $t$-distributions $\varepsilon_t \sim \mathtt{t}(\mathrm{df})$ with df, the degree of freedom, varying. Note that if a random variable $X$ follows a Student's $t$-distribution with a degree of freedom df, its mean is well defined for df $> 1$, its variance exists for df $> 2$, and its variance becomes infinite for $2 \geqslant \mathrm{df} > 1$. Case (b) sets df $\sim \mathcal{U}(2, 3)$, while Case (c) uses df $\sim \mathcal{U}(1, 2)$. As we move from Case (a) to Case (c), the noise becomes increasingly heavy-tailed. To ensure a fair comparison, most parameters are shared, e.g., $\beta_t \equiv 1$ and $\lambda = 1$. The experiment runs for $T = 1000$ steps and is replicated 10 times, with the outcomes averaged. The shadowed area depicts the area within one standard deviation, calculated over ten repetitions.

**Experiment results** Figure 1 shows the regret and convergence results across three noise cases. It is clear that for all considered algorithms, the regrets continue to grow and the $L_2$ convergence errors $\|\theta_t - \theta^*\|$ tend to diminish. The continuous growth in regrets is also observed in previous heavy-tailed experiments (Shao et al., 2018). A key message is that AdaOFUL consistently achieves the lowest regret, smallest convergence errors, and least variability. In the context of light-tailed noise (in the left column), OFUL has a slightly smaller regret than TOFU. The result implies that the truncation technique might hurt the performance under light-tailed noises. As we transition to heavy-tailed noise with finite variance (in the middle column), TOFU outperforms OFUL instead in terms of regret and convergence, implying truncation works better in this case. However, both remain suboptimal compared to AdaOFUL. In the case where the noise is predominantly heavy-tailed with only a bounded expectation (in the right column), TOFU and OFUL's convergence errors and regrets deteriorate, contrasting with the steadfast performance of AdaOFUL. These findings show AdaOFUL's empirical robustness even in the infinite variance noise regime.

## 3 An Extension to Linear MDPs

Ridge regression estimators are widely used in RL to provide confidence guarantees for bounded rewards. However, when dealing with heavy-tailed rewards, these estimators tend to degrade or even fail (as shown in Figure 1). In response, we advocate for the use of the adaptive Huber regression, or AdaOFUL, as a robust alternative to ridge regression. AdaOFUL can seamlessly enhance the original algorithm to accommodate heavy-tailed scenarios with minimal disruption to its core. In this section, we demonstrate this by integrating AdaOFUL as a foundational element to solve linear MDPs and provide variance-aware regrets. This approach can also be extended to other linear problems such as linear mixture MDPs (Zhou & Gu, 2022).

### 3.1 Linear MDPs with Heavy-tailed Rewards

**Preliminaries about linear MDPs** An episodic finite horizon MDP is denoted by a tuple $\mathcal{M} = (\mathcal{S}, \mathcal{A}, H, \{r_h\}_{h \in [H]}, \{\mathbb{P}_h\}_{h \in [H]})$ where $\mathcal{S}$ is the state space with a possibly infinite number of states, $\mathcal{A}$ the action space, $H \in \mathbb{Z}^+$ the length of each episode, $\mathbb{P}_h : \mathcal{S} \times \mathcal{A} \to \Delta(\mathcal{S})$ the transition probability function, and $r_h : \mathcal{S} \times \mathcal{A} \to \mathbb{R}$ the expected reward function. A linear MDP assumes that both the transition probability and the expected reward are linear in a known state-action feature map $\phi(\cdot, \cdot) \in \mathbb{R}^d$ (Bradtke & Barto, 1996; Melo & Ribeiro, 2007; Yang & Wang, 2019; Jin et al., 2020b).

**Definition 3.1** (Linear MDP). $\mathcal{M}$ is called a time-inhomogeneous linear MDP, if there exist some known feature map $\phi(s, a) : \mathcal{S} \times \mathcal{A} \to \mathrm{Ball}_d(1)$, unknown signed measures $\{\boldsymbol{\mu}_h^*\}_{h \in [H]} \subseteq \mathbb{R}^{d \times |\mathcal{S}|}$, and unknown coefficients $\{\boldsymbol{\theta}_h^*\}_{h \in [H]} \subseteq \mathrm{Ball}_d(W)$ such that $r_h(s, a) = \langle \phi(s, a), \boldsymbol{\theta}_h^* \rangle$ and $\mathbb{P}_h(\cdot|s, a) = \langle \phi(s, a), \boldsymbol{\mu}_h^*(\cdot) \rangle$ for any $(s, a) \in \mathcal{S} \times \mathcal{A}$, $h \in [H]$, where $\|\boldsymbol{\mu}_h^*(\mathcal{S})\| := \|\sum_{s \in \mathcal{S}} \boldsymbol{\mu}_h^*(s)\| \leqslant \sqrt{d}$ for all $h \in [H]$.

For a time-inhomogeneous MDP, we denote its deterministic and time-dependent policy by $\pi = \{\pi_h\}_{h \in [H]}$. Let $\{(s_h, a_h)\}_{h \in [H]}$ be state-action pairs such that $a_h = \pi_h(s_h)$ and $s_{h+1} \sim \mathbb{P}_h(\cdot|s_h, a_h)$. Define the occupancy measure for the policy $\pi$ at the $h$-th round by $d_h^\pi(s, a) = \mathbb{P}^\pi(s_h = s, a_h = a|s_1)$ where $(a_1, s_2, a_2, \cdots, s_h, a_h)$ is a trajectory starting from $s_1$ and following the policy $\pi$. The state-action function $Q_h^\pi(\cdot, \cdot)$ and value function $V_h^\pi(\cdot)$ at the $h$-th round are defined as $Q_h^\pi(\cdot, \cdot) = \mathbb{E}[\sum_{i=h}^H r_i(s_i, a_i)|(s_h, a_h) = (\cdot, \cdot)]$ and $V_h^\pi(\cdot) = Q_h^\pi(\cdot, \pi_h(\cdot))$ respectively. The optimal policy is denoted by $\pi^*$ and its value function is denoted by $V_1^*$. One can show that $V_1^*(s) = \sup_\pi V_1^\pi(s)$ for any $s \in \mathcal{S}$. For any value function $V$, write $[\mathbb{P}_h V](s, a) = \mathbb{E}_{s' \sim \mathbb{P}_h(\cdot|s,a)} V(s')$ and $[\mathbb{V}_h V](s, a) = [\mathbb{P}_h V^2](s, a) - [\mathbb{P}_h V]^2(s, a)$. With a slight abuse of notation, let $[\mathbb{P}_h R_h](s, a)$ and $[\mathbb{V}_h R_h](s, a)$

denote the expectation and variance of the random reward $R_h(s,a)$ at the $h$-th round given state-action pair $(s,a)$.

**Linear MDPs with heavy-tailed rewards**  We consider linear MDPs with heavy-tailed random rewards that satisfy the following assumptions.

**Assumption 3.1** (Realizable reward)**.** We assume that the following holds.

1. For all $(s,a) \in \mathcal{S} \times \mathcal{A}$ and $h \in [H]$, the random reward $R_h(s,a)$ is independent of $s_{h+1}(s,a)$, where $s_{h+1}(s,a) \sim \mathbb{P}_h(\cdot|s,a)$ represents the next state transitioned from $(s,a)$ at the $h$-th round.

2. There exists known feature maps $\widetilde{\phi}(s,a) : \mathcal{S} \times \mathcal{A} \to \mathrm{Ball}_d(1)$ and unknown coefficients $\{\psi_h^*\}_{h \in [H]} \subseteq \mathrm{Ball}_d(W)$ so that $[\mathbb{P}_h R_h^2](s,a) = \langle \widetilde{\phi}(s,a), \psi_h^* \rangle$ for all $(s,a) \in \mathcal{S} \times \mathcal{A}$ and $h \in [H]$.

**Assumption 3.2** (Bounded variance)**.** We assume that the following holds.

1. There exist known constants $\sigma_R, \sigma_{R^2} > 0$ such that $[\mathbb{V}_h R_h](s,a) \leqslant \sigma_R^2$ and $[\mathbb{V}_h R_h^2](s,a) \leqslant \sigma_{R^2}^2$ for all $(s,a,h) \in \mathcal{S} \times \mathcal{A}$ and $h \in [H]$.

2. There exist known upper bounds $\mathcal{H}, \mathcal{V} > 0$ such that for any policy $\pi$, we have $0 \leqslant \mathbb{E} R_\pi \leqslant \mathcal{H}$ and $\mathrm{Var}(R_\pi) \leqslant \mathcal{V}^2$ where $R_\pi = \sum_{h=1}^H R_h(s_h, a_h)$ denotes the sum of random rewards along the trajectory following $\pi$.

**Rationale behind the assumptions**  Assumption 3.1 assumes that the random reward at each round is independent of future states and its second moment can be realized using a known feature map. Under this assumption, linear MDPs can recover tabular MDPs by setting the size of the state-action space as $d = |\mathcal{S}||\mathcal{A}|$ and using the canonical basis $\phi(s,a) = \widetilde{\phi}(s,a) = e_{(s,a)}$ in $\mathbb{R}^d$. Assumption 3.2 places upper bounds on the means and variances of every random reward and the cumulative rewards. These upper bounds are available under the classic uniformly bounded reward assumption that $0 \leqslant \sup_{(s,a) \in \mathcal{S} \times \mathcal{A}} \sup_{h \in [H]} R_h(s,a) \leqslant 1$ so that $\sigma_R = \sigma_{R^2} = 1$ and $\mathcal{H} = \mathcal{V} = H$. We emphasize that almost all previous works use these "1" and "$H$" upper bounds implicitly in their algorithm design and regret analysis. In this way, they can't tell the effect of the expectation of cumulative rewards on the final regret from their variance. We are the first to distinguish them by separate $\mathcal{H}$ and $\mathcal{V}$. Since only upper bounds for $\mathcal{H}$ and $\mathcal{V}$ are required, in practice one can guess them using the doubling trick.[2] As we will observe, very large guessing values for $\mathcal{H}$ and $\mathcal{V}$ will not affect the order of the dominant (or variance-aware) term in our regret as long as $T$ is sufficiently large. As far as we know, Assumption 3.2 is the weakest moment condition on random rewards in the variance-aware RL literature.

**Learning protocol**  Let $\mathcal{F}_{h,k}$ denote the $\sigma$-field generated by all random variables up to, and including, the $h$-th round and $k$-th episode. At the beginning of each episode $k$, the environment selects the initial state $s_{1,k}$. The agent proposes a policy $\pi_k = \{\pi_h^k\}_{h \in [H]}$ based on the history up to the end of episode $k-1$, and then executes $\pi_k$ to generate a new trajectory $\{(s_{h,k}, a_{h,k}, r_{h,k})\}_{h \in [H]}$. Here $a_{h,k} = \pi_h^k(s_{h,k}), r_{h,k} \sim R_h(s_{h,k}, a_{h,k})$ and $s_{h+1,k} \sim \mathbb{P}(\cdot|s_{h,k}, a_{h,k})$. Here $R_h(s,a)$ denotes the distribution of the random reward conditioned on the state-action pair $(s,a)$ at horizon $h$, with its expected value being $r_h(s,a)$. The agent aims to minimize the cumulative regret over $K$ episodes, given by

$$\mathrm{Reg}(K) := \sum_{k=1}^K (V_1^* - V_1^{\pi_k})(s_{1,k}).$$

## 3.2  High-level Algorithm Description

In this subsection, we introduce VARA, an algorithm present in Algorithm 2, that extends AdaOFUL to solve linear MDPs with heavy-tailed rewards. At a high level, the VARA algorithm is built on LSVI-UCB++ (He et al., 2022), an algorithm proposed recently to achieve minimax optimality for linear MDPs.

---

[2]For example, we can guess $\mathcal{H}$ as $2, 4, 6, \cdots$. After a logarithmic number of guessing, we can find a true upper bound for $\sup_\pi \mathbb{E} R_\pi$. One can run a similar procedure for other quantities.

---

**Algorithm 2** The VARA algorithm (informal)

---
**Require** : $K, H, \mathcal{H}, \mathcal{V}, W, \sigma_R, \sigma_{R^2}$.

**1 for** *episode* $k = 1$ **to** $K$ **do**

**2**  **for** *horizon* $h = H$ **to** $1$ **do**

**3**   Based on all $\{\boldsymbol{\theta}_{h,k'}, \boldsymbol{\psi}_{h,k'}\}_{k' \leqslant k}$, estimate an optimistic $\overline{Q}_h^k$ and a pessimistic Q-value $\underline{Q}_h^k$ by LSVI-UCB++.

**4**   $\overline{V}_h^k(\cdot) = \max_a \overline{Q}_h^k(\cdot, a), \underline{V}_h^k(\cdot) = \max_a \underline{Q}_h^k(\cdot, a), \pi_h^k(\cdot) \in \operatorname{argmax}_a \overline{Q}_h^k(\cdot, a)$.

**5**  **end**

**6**  **for** *horizon* $h = 1$ **to** $H$ **do**

**7**   Play $a_{h,k} = \pi_h^k(s_{h,k})$ and observe $r_{h,k} \sim R_h(s_{h,k}, a_{h,k}), s_{h+1,k} \sim \mathbb{P}(\cdot|s_{h,k}, a_{h,k})$.

**8**   Observe feature vectors $\boldsymbol{\phi}_{h,k} = \boldsymbol{\phi}(s_{h,k}, a_{h,k})$ and $\widetilde{\boldsymbol{\phi}}_{h,k} = \widetilde{\boldsymbol{\phi}}(s_{h,k}, a_{h,k})$.

**9**   Update the estimated variance $\sigma_{h,k}$ using observed data and estimated values $\overline{Q}_h^k$ and $\underline{Q}_h^k$.

**10**   Update the parameters $w_{h,k}, \tau_{h,k}, \widetilde{w}_{h,k}, \widetilde{\tau}_{h,k}$ following the spirit of AdaOFUL.

**11**   Using $\{\boldsymbol{\phi}_{h,k'}\}_{k' \leqslant k}$ and $\{(\sigma_{h,k'}, w_{h,k'}, \tau_{h,k'})\}_{k' \leqslant k}$, AdaOFUL produces $\boldsymbol{\theta}_{h,k}$ as the estimate for $\boldsymbol{\theta}_h^*$.

**12**   Using $\{\widetilde{\boldsymbol{\phi}}_{h,k'}\}_{k' \leqslant k}$ and $\{(\sigma_{h,k'}, \widetilde{w}_{h,k'}, \widetilde{\tau}_{h,k'})\}_{k' \leqslant k}$, AdaOFUL produces $\boldsymbol{\psi}_{h,k}$ as the estimate for $\boldsymbol{\psi}_h^*$.

**13**  **end**

**14 end**

---

LSVI-UCB++ (He et al., 2022) uses weighted ridge regression, where the weights depend on some proper variance estimators $\sigma_{h,k}$'s. The variance estimation techniques in LSVI-UCB++ are important to obtain variance-aware regrets. These techniques include (i) separate variance estimation, (ii) monotonicity of value functions, and (iii) rare-switching value function update.

Due to limited space, we present the detailed and formal algorithm description in Appendix A and focus on the differences between VARA and LSVI-UCB++ (He et al., 2022) here. To obtain variance-aware regrets under heavy-tailed rewards, we made two improvements to LSVI-UCB++. First, while LSVI-UCB++ assumes a deterministic, uniformly bounded, and known reward function, we use AdaOFUL to estimate the parameters $\boldsymbol{\theta}_h^*$ and $\boldsymbol{\psi}_h^*$ for both the expected reward functions and their second-order moments. This complicates the construction of the variance estimators $\sigma_{h,k}$ and requires a more detailed analysis of their impacts on the final regrets (see Lemma D.10). Second, previous works use the Azuma-Hoeffding inequality to analyze the concentration effect in the suboptimality gap, which leads to the regret of $\widetilde{\mathcal{O}}(\sqrt{K})$. Instead, we use a variance-aware Bernstein inequality and produce a much tighter upper bound of $\widetilde{\mathcal{O}}(1)$ for the concentration effect (see Lemma D.8). We explain the analytical novelty in detail in Appendix D.2.

### 3.3 Regret Analysis

This section presents the statistical, space, and computational complexities of Algorithm 3.

**Theorem 3.1.** Consider a linear MDP satisfying Asumption 3.1 and 3.2. For any $\delta \in (0,1)$, with probability at least $1 - 21\delta$, Algorithm 3 achieves the following regret

$$\operatorname{Reg}(K) = \widetilde{\mathcal{O}}\left(d\sqrt{HK\mathcal{G}^*} + Hd\sqrt{K}\sigma_{\min} + \frac{H^{2.5}d^6\mathcal{H}^2 + Hd^2\sigma_{R^2}}{\sigma_{\min}} + H^3d^5\mathcal{H} + Hd\sigma_R + Hd^2\right), \quad (3.1)$$

where $\sigma_{\min}$ is a manually set arbitrary lower bound for all variance estimators $\sigma_{h,k}$'s,

$$\mathcal{G}^* = \min\left\{\sum_{h=1}^H \mathbb{E}_{(s,a)\sim\widetilde{d}_h^K}\left[\mathbb{V}_h R_h + \mathbb{V}_h V_{h+1}^*\right](s,a), \mathcal{V}^2\right\}, \quad (3.2)$$

and $\widetilde{d}_h^K(s,a) = \frac{1}{K}\sum_{k=1}^K d_h^{\pi_k}(s,a)$ with $d_h^{\pi_k}(s,a) = \mathbb{P}^{\pi_k}(s_h = s, a_h = a|s_0 = s_{1,k})$ the probability of reaching $(s_{h,k}, a_{h,k}) = (s, a)$ at the $h$-th step when the agent starts from $s_{1,k}$ and follows the policy $\pi_k$.

**Trade-off by $\sigma_{\min}$** Theorem 3.1 reveals a trade-off arising from the choice of $\sigma_{\min}$. The second term in equation 3.1, stemming from the imposed lower bound on variance estimates for stability purposes, is

---

positively dependent on $\sigma_{\min}$. Consequently, if $\sigma_{\min} = 0$, this term vanishes. The third term is negatively dependent on $\sigma_{\min}$ due to its effect on $\boldsymbol{H}_T$. Consider an extreme case. If $\sigma_{\min} = \infty$, $\boldsymbol{H}_T$ is reduced to $\lambda \boldsymbol{I}$, implying the shape of the confidence region $\mathcal{C}_t$ changes. This, then, would slightly decrease the confidence radii $\beta_T$ and the regret. The choice of $\sigma_{\min}$ must balance these opposing effects. Corollary 3.1 implies that choosing the optimal $\sigma_{\min}^* = \sqrt{H^{1.5} d^5 \mathcal{H}^2 + d\sigma_{R^2}} \cdot K^{-\frac{1}{4}}$ yields a regret barrier of $\widetilde{\mathcal{O}}\left(Hd \cdot \sqrt{d^5 \mathcal{H}^2 + d\sigma_{R^2}} \cdot \sqrt[4]{K}\right)$. When $K$ is sufficiently large, the regret bound in equation 3.3 can be further simplified to $\widetilde{\mathcal{O}}\left(d\sqrt{H\mathcal{G}^* K}\right)$. To the best of our knowledge, Theorem 3.1 is the first to derive the variance-aware regret for linear MDPs, especially with heavy-tailed rewards.

**Corollary 3.1.** Under the same setting of Theorem 3.1, if we set $\sigma_{\min} = \sqrt{H^{1.5} d^5 \mathcal{H}^2 + d\sigma_{R^2}} \cdot K^{-\frac{1}{4}}$, the regret of VARA is bounded by

$$\text{Reg}(K) = \widetilde{\mathcal{O}}\left(d\sqrt{H\mathcal{G}^* K} + Hd\sqrt{d^5 \mathcal{H}^2 + d\sigma_{R^2}} \cdot \sqrt[4]{K} + H^3 d^5 \mathcal{H} + Hd\sigma_R + Hd^2\right). \tag{3.3}$$

**Instance-dependent quantity $\mathcal{G}^*$** The quantity $\mathcal{G}^*$ is given by equation 3.2. Firstly, it is bounded above by $\mathcal{V}^2$ in Assumption 3.2, which sets an upper bound on the variance of the cumulative random rewards received when following any policy. Other upper bounds such as $\mathcal{H}, \sigma_R$ and $\sigma_{R^2}$ don't involve in $\mathcal{G}^*$ and thus the regret when $K$ is sufficiently large. Secondly, even $\mathcal{V}$ is set to be extremely large, $\mathcal{G}^*$ is no greater than the sum of per-round conditional variances $[\mathbb{V}_h R_h + \mathbb{V}_h V_{h+1}^*](s, a)$, weighted by an averaged occupancy measure $\widetilde{d}_h^K(s, a) := \frac{1}{K} \sum_{k=1}^K d_h^{\pi_k}(s, a)$. The function $\widetilde{d}_h^K(\cdot, \cdot)$ introduces a probability measure on $\mathcal{S} \times \mathcal{A}$ for any fixed $h \in [H]$, by the definition of $d_h^\pi$, which records the history of the policies taken.

Our variance-aware regret has two key features. Firstly, we do not require any prior knowledge of $\mathcal{G}^*$ to achieve variance awareness, which is the same as (Zanette & Brunskill, 2019). Secondly, the additional conditions imposed on the MDP structure lead to other instance-dependent regrets. In the following, we also impose Assumption 3.3 for a fair comparison with related work. However, we would like to emphasize that all of our results are obtained in the presence of heavy-tailed rewards.

**Assumption 3.3.** We assume that $0 \leqslant R_h(s, a) \leqslant 1$ for all $h \in [H]$ and $(s, a) \in \mathcal{S} \times \mathcal{A}$.

### 3.4 Other Instance-dependent Regrets

**Worst-case regret** Under Assumption 3.3, $\mathcal{V}^2 = H^2$ according to the law of total variance (Azar et al., 2013). Consequently, we can infer that $\mathcal{G}^* \leqslant H^2$, and the regret reduces to the minimax optimal $\widetilde{\mathcal{O}}(dH\sqrt{HK})$ (He et al., 2022). The authors achieved this regret by directly setting $\sigma_{\min} = 1/H$, without taking into account the trade-off introduced by $\sigma_{\min}$. Although this was sufficient for their worst-case scenario, it was not suitable for our goal of achieving variance awareness. If we also set $\sigma_{\min} = 1/H$, the second term in equation 3.1 becomes $d\sqrt{K}$, and we cannot determine the dominant term between $d\sqrt{HK\mathcal{G}^*}$ and $d\sqrt{K}$. Once we balance the trade-off of $\sigma_{\min}$, the second term becomes much smaller, making $\widetilde{\mathcal{O}}(d\sqrt{H\mathcal{G}^* K})$ the dominant term.

**Range-dependent regret** Let $\mathcal{S}_{s,a}$ be the set of immediate successor states after one transition from state $s$ upon taking action $a$, which is also the support set of $\mathbb{P}(\cdot|s, a)$. Define $\Phi_{\text{succ}}$ as the maximum value function range when restricted to the immediate successor states:

$$\Phi_{\text{succ}} := \sup_{h \in [H]} \sup_{(s,a)} \left[ \sup_{s' \in \mathcal{S}_{s,a}} V_{h+1}^*(s') - \inf_{s' \in \mathcal{S}_{s,a}} V_{h+1}^*(s') \right].$$

Since the variance is upper bounded by one-fourth of the square range of a random variable, we have $\sup_{h \in [H]} \sup_{(s,a)} [\mathbb{V}_h V_{h+1}^*](s, a) \leqslant \frac{1}{4} \Phi_{\text{succ}}^2$ and thus $\mathcal{G}^* \leqslant H(\sigma_R^2 + \Phi_{\text{succ}}^2)$. Therefore, our regret reduces to $\widetilde{\mathcal{O}}\left(dH\sqrt{(\sigma_R^2 + \Phi_{\text{succ}}^2)K}\right)$. It is worth noting that similar range-dependent regrets have been derived for tabular MDPs with bounded rewards (Bartlett & Tewari, 2009; Fruit et al., 2018; Zanette & Brunskill, 2019), but to the best of our knowledge, we obtain the first such result for linear MDPs with heavy-tailed rewards.

**First-order regret** The first-order regret that scales proportionally to $V_1^*$, where $V_1^* := V_1^*(s_1)$ is the value of the optimal value policy at the initial state $s_1$, has been studied for tabular MDPs (Jin et al.,

2020a) and linear MDPs (Wagenmaker et al., 2022a).[3] However, under Assumption 3.3, the corresponding instance-dependent quantity $H^2 V_1^*$ can be much larger than $\mathcal{G}^*$. This is because

$$\mathcal{G}^* \overset{(a)}{\leqslant} H \sum_{h=1}^{H} \mathbb{E}_{(s,a) \sim \widetilde{d}_h^K}[r_h + \mathbb{P}_h V_{h+1}^*](s,a) \overset{(b)}{\leqslant} H \sum_{h=1}^{H} \mathbb{E}_{(s,a) \sim \widetilde{d}_h^K} V_h^*(s,a) \overset{(c)}{\leqslant} H^2 V_1^*,$$

where $(a)$ uses $0 \leqslant R_h \leqslant 1$ and $0 \leqslant V_{h+1}^* \leqslant H-1$ under Assumption 3.3, $(b)$ uses the optimality condition $V_h^*(s) = [r_h + \mathbb{P}_h V_{h+1}^*](s,a)$ for any $(s,a) \in \mathcal{S} \times \mathcal{A}$, and $(c)$ uses $V_{h+1}^*(s) \leqslant V_h^*(s)$ for any $h \in [H]$ and $s \in \mathcal{S}$ and $\sum_{a \in \mathcal{S}} \widetilde{d}_h^k(s_1, a) = 1$ since each episode starts at a fixed state $s_1$. Moreover, even replacing $\mathcal{G}^*$ with the coarse upper bound $H^2 V_1^*$, our regret bound becomes $\widetilde{\mathcal{O}}(\sqrt{d^2 H^3 V_1^* K})$, which has a better dependence on $d$ than $\widetilde{\mathcal{O}}(\sqrt{d^3 H^3 V_1^* K})$ in (Wagenmaker et al., 2022a).

**Concentrability-dependent regret** Let $R_{\pi*}$ denote the sum of random rewards collected in a trajectory following the optimal policy $\pi^*$. It is straightforward to see that $\mathrm{Var}(R_{\pi*}) = \sum_{h=1}^{H} \mathbb{E}_{(s,a) \sim d_h^{\pi*}}[\mathbb{V}_h R_h + \mathbb{V}_h V_{h+1}^*](s,a)$. Since $\mathcal{G}^* \leqslant \sup_\pi \sum_{h=1}^{H} \mathbb{E}_{(s,a) \sim d_h^\pi}[\mathbb{V}_h R_h + \mathbb{V}_h V_{h+1}^*](s,a)$, we can show that $\mathcal{G}^* \leqslant C^\dagger \cdot \mathrm{Var}(R_{\pi*})$ where $C^\dagger$ is a data coverage measure defined as

$$C^\dagger := \sup_\pi \frac{\sum_{h=1}^{H} \mathbb{E}_{(s,a) \sim d_h^\pi}[\mathbb{V}_h R_h + \mathbb{V}_h V_{h+1}^*](s,a)}{\sum_{h=1}^{H} \mathbb{E}_{(s,a) \sim d_h^{\pi*}}[\mathbb{V}_h R_h + \mathbb{V}_h V_{h+1}^*](s,a)}.$$

Therefore, our regret reduces to $\widetilde{\mathcal{O}}(d\sqrt{C^\dagger \mathrm{Var}(R_{\pi*}) H K})$ given $C^\dagger < \infty$. The $C^\dagger$ is a counterpart of the generalized concentrability coefficient which quantifies the effect of the distribution shift in offline RL (Chen & Jiang, 2019; Xie et al., 2021; Cheng et al., 2022).

### 3.5 Space and Computational Complexities

**Theorem 3.2** (Space and computational complexity). *Assume the Nesterov accelerated method is used as a solver to solve the adaptive Huber regression. Solving a $H$-horizon finite MDP in $K$ episodes, VARA takes $\mathcal{O}(d^3 H^2 + d|\mathcal{A}|HK)$ space and has a running time of $\widetilde{\mathcal{O}}(d^4|\mathcal{A}|H^3 K + HK(d + H^{-3/4}d^{-3/2}K^{3/4}))$.*

On one hand, VARA achieves the same space complexity as LSVI-UCB++ but is slightly worse than the original LSVI-UCB (Jin et al., 2020b) that needs $\mathcal{O}(d^2 H + d|\mathcal{A}|HK)$ space. This is because the technique of monotone value function update requires remembering at most $\widetilde{\mathcal{O}}(dH)$ latest value functions, incurring a slightly worse dependence on $d$ and $H$. On the other hand, the computational complexity of VARA $\widetilde{\mathcal{O}}(d^4|\mathcal{A}|H^3 K + HK(d + H^{-3/4}d^{-3/2}K^{3/4}))$ is slightly worse than LSVI-UCB++'s $\widetilde{\mathcal{O}}(d^4|\mathcal{A}|H^3 K)$ in terms of the dependence on $K$. This is because the adaptive Huber regression estimator does not have a closed-form solution. Even though the Nesterov accelerated method is used, a slightly larger computational complexity is still incurred due to the possibly large conditional number. However, VARA's computational complexity is better than LSVI-UCB's $\widetilde{\mathcal{O}}(d^2|\mathcal{A}|HK^2)$ thanks to the rare-switching mechanism in LSVI-UCB++.

## 4 Related Work

**Heavy-tailed rewards in online decision making** The standard heavy-tailed setting assumes rewards with $(1 + \varepsilon)$-moments where $\varepsilon > 0$. There exists a large body of work considering this setting in multi-arm bandits, including deterministic (Vakili et al., 2013) and non-deterministic settings (Bubeck et al., 2013; Carpentier & Valko, 2014; Lattimore, 2017; Bhatt et al., 2022). To handle heavy-tailed rewards, robust mean estimation methods such as median of means and truncation have been applied to linear bandits (Medina & Yang, 2016; Shao et al., 2018; Lu et al., 2019; Xue et al., 2021). Given that our objective is to provide variance-aware regrets for general linear bandits, the minimal requirement is to have bounded second moments, which is the primary focus of this study. Under the assumption of rewards with bounded second

---

[3]They assume all episodes start from the same initial state so that $s_{1,k} \equiv s_1$. However, our regret can be easily extended to the setting where initial states are different. In this case one should replace $V_1^* K$ with $\sum_{k=1}^{K} V_1^*(s_{1,k})$ in the regret bound.

moments, the minimax optimal regret is $\widetilde{\mathcal{O}}(d\sqrt{T})$. Recently, Kang & Kim (2023) introduced the use of Huber regression to address heavy-tailed linear contextual bandits where the action set is fixed (such that $\mathcal{D}_t \equiv \mathcal{D}$) and the arm $\boldsymbol{\phi}_t$ is independently and identically distributed, sampled from a fixed distribution over $\mathcal{D}$. In contrast, our proposed AdaOFUL is simpler and more versatile, capable of being applied to more complex scenarios where the arm $\boldsymbol{\phi}_t$ is selected adaptively based on historical observations.

On the other hand, there are a few RL algorithms designed to handle heavy-tailed rewards for MDPs. One is (Zhuang & Sui, 2021), which modifies UCRL2 and Q-Learning by using truncated rewards and achieves minimax optimal regret in tabular MDPs. However, none of these methods for linear bandits or MDPs provide variance-aware regrets, even if variance information is available. Moreover, simple truncation methods are not optimal in the noiseless setting. Recently, Huang et al. (2023) extends the Huber regression to the more general $(1 + \varepsilon)$-moment setting and provides instance-dependent regret bounds for both linear bandits and linear MDPs.

**Variance-aware regrets for linear bandits**   A weighted ridge regression-based algorithm proposed by Kirschner & Krause (2018) achieves the same regret in equation 2.7 by assuming each $\varepsilon_t$ is $\nu_t$-sub-Gaussian. More recently, Zhou & Gu (2022) obtained the same regret assuming each $\varepsilon_t$ is uniformly bounded and has finite conditional variance $\nu_t^2$. In the case where the information of conditional variances $\{\nu_t\}_{t \geqslant 0}$ is unknown, Zhang et al. (2021) and Kim et al. (2021) achieved regret bounds that involve sub-optimal dependence on $d$. The currently tightest variance-aware regret is achieved by Zhao et al. (2023) with an optimal dependence on $d$. Recently, Dai et al. (2022) explored variance-aware regrets in the context of high-dimensional and sparse linear bandits, a topic that extends beyond the scope of our paper.   All of the above works consider light-tailed noises, which are either sub-Gaussian or uniformly bounded.

**Robust approach to instance-dependent bounds**   Recent research explores the robust mean estimation approach to obtain instance-dependent regrets, leveraging the observation that robust estimators can achieve estimation errors that only depend on the noise scale. Such estimators often have better theoretical guarantees than non-robust ones, whose estimation errors additionally depend on the range of the problem noise. For instance, Pananjady & Wainwright (2020) use the median-of-means technique (Lecué & Lerasle, 2020) to achieve local minimax optimality that depends on the standard deviations of the optimal value function and random rewards for synchronous tabular MDPs. In linear bandits, Wagenmaker et al. (2022a) use Catoni's estimator (Catoni, 2012) to estimate the mean of $\boldsymbol{v}^\top \boldsymbol{H}_T^{-1} \boldsymbol{\phi}_t y_t / \sigma_t^2$ for a fixed unit-norm vector $\boldsymbol{v}$. In contrast, we modify the adaptive Huber regression to estimate $\boldsymbol{\theta}^*$ directly. This difference makes their bounds depend on the second moments of $y_t$'s, while ours only relies on their variances. Moreover, all of these works, except ours, still assume light-tailed rewards.

**Variance-aware regrets for tabular and linear MDPs**   In the context of online episodic MDPs, Zanette & Brunskill (2019) first derived a variance-aware regret bound in the tabular setting with uniformly bounded rewards. Their model-based algorithm, Euler, achieves a regret that can be bounded by either $\widetilde{\mathcal{O}}(\sqrt{\mathbb{Q}^S AHK})$ or $\widetilde{\mathcal{O}}(\sqrt{\mathcal{G}^2 SAK})$, where $\mathbb{Q}^* = \max_{(s,a,h)}(\mathbb{V}_h R_h + \mathbb{V}_h V_{h+1}^*)(s,a)$ is the maximum per-round conditional variance and $\mathcal{G}$ is a deterministic upper bound on the maximum attainable reward on a single trajectory for any policy $\pi$, such that $\sum_{h=1}^H R_h(s_h, \pi(s_h)) \leqslant \mathcal{G}$. One can show that our instance-dependent quantity $\mathcal{G}^*$ is smaller than $\min\{H\mathbb{Q}^*, \mathcal{G}^2\}$ therein. Later, Jin et al. (2020a) adopted a modified analysis of Euler to obtain the regret bound $\widetilde{\mathcal{O}}(\sqrt{SAH^3 V_1^* K})$ with $V_1^* = V_1^*(s_1)$.[4] In linear MDPs, there are several recent works on obtaining regret bounds for model-free algorithms. For example, Wagenmaker et al. (2022a) proposed an optimistic algorithm with a regret bound that scales as $\widetilde{\mathcal{O}}(\sqrt{d^3 H^3 V_1^* K})$. However, this algorithm is computationally inefficient. A computationally efficient alternative suffers from a slightly worse regret $\widetilde{\mathcal{O}}(\sqrt{d^4 H^3 V_1^* K})$. All of these works utilize the instance-dependent quantity $H^2 V_1^*$ (assuming $\mathcal{H} = H$). However, as we argued, our proposed quantity $\mathcal{G}^*$ is smaller than $H^2 V_1^*$, which implies that our algorithm may achieve better performance than these previous works.   Another research direction

---

[4]Unlike our setting, they assume all initial states are the same, denoted as $s_1$. Furthermore, the original regret $\widetilde{\mathcal{O}}(\sqrt{SAH \cdot V_1^* K})$ by Jin et al. (2020a) was derived for an MDP where the reward function equals to one deterministically only at a single $(h, s)$ pair. In this way, they have $0 \leqslant V_1^* \leqslant 1$. To convert it in the considered setting where $0 \leqslant V_1^* \leqslant H$, an additional factor of $H$ should be multiplied to their regret.

explores variance-adaptive algorithms for linear mixture MDPs, as initially explored by (Zhou et al., 2021). Subsequent developments in this area were made by (Zhang et al., 2021; Zhou & Gu, 2022), culminating in the state-of-the-art advancements by (Zhao et al., 2023). While our study does not consider this particular setting, it is straightforward to extend our techniques and analysis to it, as linear mixture MDPs are generally considered simpler than linear MDPs.

**Other instance-dependent bounds**   In the infinite-horizon setting, Pananjady & Wainwright (2020); Khamaru et al. (2021); Li et al. (2023) provided variance-aware sample complexities for Q-Learning and its variants in tabular MDPs, given a generative model that produces independent samples for all state-action pairs in every round. Variance-aware performance guarantees have also been established for offline RL optimization (Yin & Wang, 2021; Nguyen-Tang et al., 2023), off-policy evaluation (Min et al., 2021), stochastic approximation (Mou et al., 2020; 2022). Another approach to instance-dependence bounds focuses on the minimum suboptimality gap, which is the minimum gap between the best and second-best actions over all states (He et al., 2021; Wagenmaker et al., 2022c; Wagenmaker & Jamieson, 2022; Dong & Ma, 2022). However, due to the differences in the settings, we cannot make a meaningful comparison between these bounds and ours.

## 5   Conclusion

This paper introduces two new algorithms, AdaOFUL for linear bandits and VARA for linear MDPs, both of which use modifications of the original adaptive Huber regression and are designed to handle online sequential decision-making. With only the assumption of bounded reward variances, our algorithms achieve either state-of-the-art or finer variance-aware regrets. Additionally, in linear MDPs, the instance-dependent quantity $\mathcal{G}^*$ can be bounded by other instance-dependent quantities when additional structure assumptions are available. Our modified adaptive Huber regression can be a useful building block for algorithm design in online problems with heavy-tailed rewards.

### Acknowledgments

The work was done when Xiang Li was a visiting student in the IVGS program at the Department of Statistical Sciences, University of Toronto. Qiang Sun was partially supported in part by the Natural Sciences and Engineering Research Council of Canada under Grant RGPIN-2018-06484 and a Data Sciences Institute Catalyst Grant.

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

# Appendix

## Overview

We describe VARA detailedly in Appendix A and explain the rationale behind its variance estimator in Appendix B. Appendix C contains proofs for Theorem 2.2 and related lemmas specifically for linear bandits. The theoretical analysis for VARA is presented in Appendix D, where we offer a proof sketch for Theorem 3.1, while all related technical lemmas are deferred to Appendices F and G. We also highlight the differences between our analysis and previous work. In Appendix E, we provide a proof for Theorem 3.2 that analyzes the space and computational complexity of VARA.

## Table of Contents

# A  Detailed Algorithm Description for VARA

For each episode $k$, we perform optimistic value iterations (Lines 3-11), compute the greedy policy $\pi_h^k$ with respective to the pessimistic value function $\overline{Q}_h^k$ (Line 12), and then execute it to collect a new trajectory of data (Lines 16-17). The rest of Algorithm 3 updates maintained estimators, including the conditional variances $\sigma_{h,k}^2$ (Line 19), the transition parameters $\boldsymbol{\mu}_{h,k}$ (Line 20), the reward parameters $\boldsymbol{\theta}_{h,k}, \boldsymbol{\psi}_{h,k}$ (Lines 21-22), and the Hessian matrices $\boldsymbol{H}_{h,k}, \widetilde{\boldsymbol{H}}_{h,k}$ (Line 23). In what follows, we discuss in detail the key steps of Algorithm 3 in more detail.

**Reward estimation**  Since rewards are collected adaptively and have only finite second moments, we use the same strategy adopted in AdaOFUL to estimate $\boldsymbol{\theta}_h^*$:

$$\boldsymbol{\theta}_{h,k} := \underset{\boldsymbol{\theta} \in \mathrm{Ball}_d(W)}{\mathrm{argmin}} \left\{ L_{h,k}^{(R)}(\boldsymbol{\theta}) := \frac{\lambda}{2} \|\boldsymbol{\theta}\|^2 + \sum_{j=1}^k \ell_{\tau_{h,j}} \left( \frac{r_{h,j} - \langle \boldsymbol{\phi}_{h,j}, \boldsymbol{\theta} \rangle}{\sigma_{h,j}} \right) \right\}. \tag{A.1}$$

Following the spirit of Theorem 2.1, we set $\tau_0 = \widetilde{\mathcal{O}}(\sqrt{d})$ with its detailed expression provided in equation D.5 of the online supplement.

**Transition estimation**  Let $\boldsymbol{\delta}(s) \in \mathbb{R}^{|\mathcal{S}|}$ be a one-hot vector that is zero everywhere except for the entry corresponding to the state $s$, which is one. We define $\boldsymbol{\varepsilon}_{h,k} = \mathbb{P}_h(\cdot|s_{h,k}, a_{h,k}) - \boldsymbol{\delta}(s_{h+1,k})$. As $\mathbb{E}[\boldsymbol{\varepsilon}_{h,k}|\mathcal{F}_{h,k}] = \mathbf{0}$, $\boldsymbol{\delta}(s_{h+1,k})$ is an unbiased estimator of $\mathbb{P}_h(\cdot|s_{h,k}, a_{h,k}) = \boldsymbol{\mu}_h^\top \boldsymbol{\phi}(s_{h,k}, a_{h,k}) = \boldsymbol{\mu}_h^\top \boldsymbol{\phi}_{h,k}$. Thus, we can learn $\boldsymbol{\mu}_h$ by regressing $\boldsymbol{\delta}(s_{h+1,k})$ on $\boldsymbol{\phi}_{h,k} := \boldsymbol{\phi}(s_{h,k}, a_{h,k})$:

$$\boldsymbol{\mu}_{h,k} := \underset{\boldsymbol{\mu} \in \mathbb{R}^{d \times |\mathcal{S}|}}{\mathrm{argmin}} \left\{ L_{h,k}^{(P)}(\boldsymbol{\mu}) := \frac{\lambda}{2} \|\boldsymbol{\mu}\|_F^2 + \sum_{j=1}^k \left\| \frac{\boldsymbol{\mu}^\top \boldsymbol{\phi}_{h,k} - \boldsymbol{\delta}(s_{h+1,j})}{\sigma_{h,j}} \right\|^2 \right\} \tag{A.2}$$

where $\|\cdot\|_F$ denotes the Frobenius norm. This problem admits a closed-form solution given by $\boldsymbol{\mu}_{h,k} = \boldsymbol{H}_{h,k}^{-1} \sum_{j=1}^k \sigma_{h,j}^{-2} \boldsymbol{\phi}_{h,j} \boldsymbol{\delta}(s_{h+1,j})^\top$. We emphasize that VARA doesn't need to compute $\boldsymbol{\mu}_{h,k}$ exactly out. VARA relies on only the matrix product of $\boldsymbol{\mu}_{h,k}$ and a vectoerzied value function $\boldsymbol{V}$ that is $\boldsymbol{\mu}_{h,k} \boldsymbol{V} = \boldsymbol{H}_{h,k}^{-1} \sum_{j=1}^k \sigma_{h,j}^{-2} \boldsymbol{\phi}_{h,j} V(s_{h+1,k})$ for any value function $V(\cdot)$. As Theorem 3.2 shows, both the computation and space complexity do not depend on the finite value of $|\mathcal{S}|$.

**Variance estimation for rewards**  In linear MDPs, estimating the variance of the reward $R_h(s, a)$ is straightforward. Since $\mathbb{P}_h R_h^2(s, a) = \langle \widetilde{\boldsymbol{\phi}}(s, a), \boldsymbol{\psi}_h^* \rangle$, we estimate $\boldsymbol{\psi}_h^*$ by

$$\boldsymbol{\psi}_{h,k} := \underset{\boldsymbol{\psi} \in \mathrm{Ball}_d(W)}{\mathrm{argmin}} \left\{ L_{h,k}^{(R^2)}(\boldsymbol{\psi}) := \frac{\lambda}{2} \|\boldsymbol{\psi}\|^2 + \sum_{j=1}^k \ell_{\widetilde{\tau}_{h,j}} \left( \frac{r_{h,j}^2 - \langle \widetilde{\boldsymbol{\phi}}_{h,j}, \boldsymbol{\psi} \rangle}{\sigma_{h,j}} \right) \right\} \tag{A.3}$$

where $\widetilde{\tau}_{h,k} = \widetilde{\tau}_0 \sqrt{1 + \widetilde{w}_{h,k}^2}/\widetilde{w}_{h,k}$ is the corresponding robustification parameter and $\widetilde{w}_{h,k} = \|\widetilde{\boldsymbol{\phi}}_{h,k}\|_{\boldsymbol{H}_{h,k-1}^{-1}}$ is the importance weight. We then estimate $[\mathbb{V}_h R_h](s_h, a_h)$ by

$$[\widehat{\mathbb{V}}_h R_h](s_{h,k}, a_{h,k}) = \langle \widetilde{\boldsymbol{\phi}}_{h,k}, \boldsymbol{\psi}_{h,k-1} \rangle - \left[ \langle \boldsymbol{\phi}_{h,k}, \boldsymbol{\theta}_{h,k-1} \rangle_{[0,\mathcal{H}]} \right]^2. \tag{A.4}$$

**Variance estimation**  Inspired by Hu et al. (2022), we set the variance estimator $\sigma_{h,k}$ to be

$$\sigma_{h,k}^2 = \max \left\{ \sigma_{\min}^2, \, d^3 H \cdot E_{h,k}, \, J_{h,k}, \, c_0^{-2} b_{h,k}^2, \, \left( \frac{W}{\sqrt{c_1 d}} + \mathcal{H} d^{2.5} H \right) b_{h,k} \right\} \tag{A.5}$$

where $\sigma_{\min}$ is a small positive constant to avoid singularity, $b_{h,k} = \max\{\|\boldsymbol{\phi}_{h,k}\|_{\boldsymbol{H}_{h,k-1}^{-1}}, \|\widetilde{\boldsymbol{\phi}}_{h,k}\|_{\widetilde{\boldsymbol{H}}_{h,k-1}^{-1}}\}$ is the bonus term, $E_{h,k}$ and $J_{h,k}$ are defined as

$$J_{h,k} = [\widehat{\mathbb{V}}_{h,k} R_h + \widehat{\mathbb{V}}_{h,k} \overline{V}_{h+1}^k](s_{h,k}, a_{h,k}) + R_{h,k} + U_{h,k}, \tag{A.6}$$

---

**Algorithm 3** The VARA algorithm (formal)

**Require** : $K, H, \mathcal{H}, \mathcal{V}, W, \sigma_R, \sigma_{R^2}, \tau_0, \tilde{\tau}_0$.

**Initialization:** $\boldsymbol{H}_{h,0} = \widetilde{\boldsymbol{H}}_{h,0} = \lambda \boldsymbol{I}, c_0 = \frac{1}{6\sqrt{3\log\frac{2HK^2}{\delta}}}, c_1 = \frac{1}{42\cdot\log\frac{2HK^2}{\delta}}, \lambda = \frac{1}{\mathcal{H}^2+W^2}, k_{\text{last}} = 1$.

**1** **for** *episode* $k = 1$ **to** $K$ **do**
**2** $\quad \overline{V}_{H+1}^k(\cdot) = \underline{V}_{H+1}^k(\cdot) = 0$  **for** *round* $h = H$ **to** $1$ **do**
**3** $\quad\quad$ **if** *there exists a stage* $h' \in [H]$ *such that* $\det(\boldsymbol{H}_{h',k-1}) \geqslant 2\det(\boldsymbol{H}_{h',k_{\text{last}}-1})$ **then**
**4** $\quad\quad\quad$ Compute the products $\boldsymbol{\mu}_{h,k-1}\overline{\boldsymbol{V}}_{h+1}^k$ and $\boldsymbol{\mu}_{h,k-1}\underline{\boldsymbol{V}}_{h+1}^k$ with $\boldsymbol{\mu}_{h,k}$ given in equation A.2.
**5** $\quad\quad\quad$ $\widehat{Q}_h^k(\cdot,\cdot) = \langle\boldsymbol{\phi}(\cdot,\cdot), \boldsymbol{\theta}_{h,k-1} + \boldsymbol{\mu}_{h,k-1}\overline{\boldsymbol{V}}_{h+1}^k\rangle + \beta\|\boldsymbol{\phi}(\cdot,\cdot)\|_{\boldsymbol{H}_{h,k-1}^{-1}}$.
**6** $\quad\quad\quad$ $\breve{Q}_h^k(\cdot,\cdot) = \langle\boldsymbol{\phi}(\cdot,\cdot), \boldsymbol{\theta}_{h,k-1} + \boldsymbol{\mu}_{h,k-1}\underline{\boldsymbol{V}}_{h+1}^k\rangle - \beta\|\boldsymbol{\phi}(\cdot,\cdot)\|_{\boldsymbol{H}_{h,k-1}^{-1}}$.
**7** $\quad\quad\quad$ $\overline{Q}_h^k(\cdot,\cdot) = \min\left\{\widehat{Q}_h^k(\cdot,\cdot), \overline{Q}_h^{k-1}(\cdot,\cdot), \mathcal{H}\right\}, \underline{Q}_h^k(\cdot,\cdot) = \max\left\{\breve{Q}_h^k(\cdot,\cdot), \underline{Q}_h^{k-1}(\cdot,\cdot), 0\right\}$.
**8** $\quad\quad\quad$ Record the last updating episode $k_{\text{last}} = k$.
**9** $\quad\quad$ **else**
**10** $\quad\quad\quad$ $\overline{Q}_h^k(\cdot,\cdot) = \overline{Q}_h^{k-1}(\cdot,\cdot), \underline{Q}_h^k(\cdot,\cdot) = \underline{Q}_h^{k-1}(\cdot,\cdot)$.
**11** $\quad\quad$ **end**
**12** $\quad\quad$ $\overline{V}_h^k(\cdot) = \max_a \overline{Q}_h^k(\cdot,a), \underline{V}_h^k(\cdot) = \max_a \underline{Q}_h^k(\cdot,a)$.
**13** $\quad\quad$ $\pi_h^k(\cdot) \in \operatorname{argmax}_a \overline{Q}_h^k(\cdot,a)$.
**14** $\quad$ **end**
**15** $\quad$ Receive the initial state $s_{1,k}$.
**16** $\quad$ **for** *round* $h = 1$ **to** $H$ **do**
**17** $\quad\quad$ Play $a_{h,k} = \pi_h^k(s_{h,k})$ and observe $r_{h,k} \sim R_h(s_{h,k}, a_{h,k}), s_{h+1,k} \sim \mathbb{P}(\cdot|s_{h,k}, a_{h,k})$.
**18** $\quad\quad$ Observe feature vectors $\boldsymbol{\phi}_{h,k} = \boldsymbol{\phi}(s_{h,k}, a_{h,k})$ and $\widetilde{\boldsymbol{\phi}}_{h,k} = \widetilde{\boldsymbol{\phi}}(s_{h,k}, a_{h,k})$.
**19** $\quad\quad$ Set the bonus as $b_{h,k} = \max\{\|\boldsymbol{\phi}_{h,k}\|_{\boldsymbol{H}_{h,k-1}^{-1}}, \|\widetilde{\boldsymbol{\phi}}_{h,k}\|_{\widetilde{\boldsymbol{H}}_{h,k-1}^{-1}}\}$.
**20** $\quad\quad$ Set the estimated variance $\sigma_{h,k}$ as in equation A.5.
**21** $\quad\quad$ Compute $\boldsymbol{\theta}_{h,k}$ via equation A.1 with $\tau_{h,k} = \tau_0\sqrt{1 + w_{h,k}^2}/w_{h,k}$ and $w_{h,k} = \sigma_{h,k}^{-1}\|\boldsymbol{\phi}_{h,k}\|_{\boldsymbol{H}_{h,k-1}^{-1}}$.
**22** $\quad\quad$ Compute $\boldsymbol{\psi}_{h,k}$ via equation A.3 with $\widetilde{\tau}_{h,k} = \widetilde{\tau}_0\sqrt{1 + \widetilde{w}_{h,k}^2}/\widetilde{w}_{h,k}$ and $\widetilde{w}_{h,k} = \sigma_{h,k}^{-1}\|\widetilde{\boldsymbol{\phi}}_{h,k}\|_{\widetilde{\boldsymbol{H}}_{h,k-1}^{-1}}$.
**23** $\quad\quad$ Update $\boldsymbol{H}_{h,k} = \boldsymbol{H}_{h,k-1} + \sigma_{h,k}^{-2}\boldsymbol{\phi}_{h,k}\boldsymbol{\phi}_{h,k}^\top$ and $\widetilde{\boldsymbol{H}}_{h,k} = \widetilde{\boldsymbol{H}}_{h,k-1} + \sigma_{h,k}^{-2}\widetilde{\boldsymbol{\phi}}_{h,k}\widetilde{\boldsymbol{\phi}}_{h,k}^\top$.
**24** $\quad$ **end**
**25** **end**

---

$$E_{h,k} = \min\left\{\mathcal{H}^2, 2\mathcal{H}\beta_0 \cdot \|\boldsymbol{\phi}_{h,k}\|_{\boldsymbol{H}_{h,k-1}^{-1}} + \mathcal{H} \cdot \left[\widehat{\mathbb{P}}_{h,k}(\overline{V}_{h+1}^k - \underline{V}_{h+1}^k)\right](s_{h,k}, a_{h,k})\right\}, \tag{A.7}$$

in which $\beta_0 = \widetilde{\mathcal{O}}\left(\sigma_{\min}^{-1}\mathcal{H}\sqrt{d^3H}\right)$ is an initial exploration radius, $\widehat{\mathbb{P}}_{h,k}(\cdot|s,a) = \boldsymbol{\mu}_{h,k-1}^\top\boldsymbol{\phi}(s,a)$ is the empirical transition kernel at the $h$-th round and $k$-the episode, $\widehat{\mathbb{V}}_h(\cdot)$ the empirical variance operator defined in equation A.4, and $R_{h,k}, U_{h,k}$ are defined as

$$R_{h,k} := \beta_{R^2}\|\widetilde{\boldsymbol{\phi}}_{h,k}\|_{\widetilde{\boldsymbol{H}}_{h,k-1}^{-1}} + 2\mathcal{H}\beta_R\|\boldsymbol{\phi}_{h,k}\|_{\boldsymbol{H}_{h,k-1}^{-1}}, \tag{A.8}$$

$$U_{h,k} = \min\left\{\mathcal{V}^2, 11\mathcal{H}\beta_0 \cdot \|\boldsymbol{\phi}_{h,k}\|_{\boldsymbol{H}_{h,k-1}^{-1}} + 4\mathcal{H} \cdot \widehat{\mathbb{P}}_{h,k}(\overline{V}_{h+1}^k - \underline{V}_{h+1}^k)(s_{h,k}, a_{h,k})\right\}, \tag{A.9}$$

with $\beta_R = \widetilde{\mathcal{O}}(\sqrt{d}), \beta_{R^2} = \widetilde{\mathcal{O}}\left(\sqrt{d} + \sqrt{d}\frac{\sigma_{R^2}}{\sigma_{\min}}\right)$ being two initial exploration radiuses. In Appendix B, we explain in detail why $\sigma_{h,k}$'s are taken in the above way.

## B  Variance Estimation for Value Functions

To achieve worst-case optimality, He et al. (2022) proposes two important techniques we adopt in Algorithm 3.

The first is the monotonicity of value functions. Specifically, we aim to enforce a decrease in $k$ for the actual optimistic value function $\overline{Q}_h^k(\cdot, \cdot)$ and an increase in $k$ for the actual pessimistic value function $\underline{Q}_h^k(\cdot, \cdot)$. This concept is explained in detail below. In linear MDPs, we have $[\mathbb{P}_h V_{h+1}](s, a) = \langle \phi(s, a), \mu_h^* V_{h+1} \rangle$ for any value function $V = \{V_h\}_{h \in [H]}$ and $[\mathbb{P}_h R_h](s, a) = \langle \phi(s, a), \theta_h^* \rangle$ for all $h \in [H]$. One crucial aspect of typical analysis (including ours) is demonstrating the high probability of the following event outlined in Appendix D.1. Specifically, for all $h \in [H]$ and $k \in [K]$, we need to establish that the following equations hold simultaneously with high probability:

$$\|\theta_{h,k} - \theta_h^*\|_{H_{h,k}} \leqslant \beta_R = \tilde{\mathcal{O}}(\sqrt{d}),$$
$$\|(\mu_{h,k-1} - \mu_h^*)\overline{V}_{h+1}^k\|_{H_{h,k-1}} \leqslant \beta_V = \tilde{\mathcal{O}}(\sqrt{d}), \tag{B.1}$$
$$\|(\mu_{h,k-1} - \mu_h^*)\underline{V}_{h+1}^k\|_{H_{h,k-1}} \leqslant \beta_V = \tilde{\mathcal{O}}(\sqrt{d}).$$

Conditional on the event that all inequalities in equation B.1 hold, we can easily verify that

$$|\langle \phi(\cdot, \cdot), \theta_{h,k-1} \rangle - [\mathbb{P}_h R_h](\cdot, \cdot)| \leqslant \beta_R \|\phi(\cdot, \cdot)\|_{H_{h,k-1}^{-1}},$$
$$|\langle \phi(\cdot, \cdot), \mu_{h,k-1} V_{h+1} \rangle - [\mathbb{P}_h V_{h+1}](\cdot, \cdot)| \leqslant \beta_V \|\phi(\cdot, \cdot)\|_{H_{h,k-1}^{-1}}$$

for both $V_{h+1} \in \{\underline{V}_{h+1}^k, \overline{V}_{h+1}^k\}$. Therefore, we define the temporary optimistic value function by

$$\widehat{Q}_h^k(\cdot, \cdot) = \langle \phi(\cdot, \cdot), \theta_{h,k-1} + \mu_{h,k-1} \overline{V}_{h+1}^k \rangle + \beta \|\phi(\cdot, \cdot)\|_{H_{h,k-1}^{-1}},$$

and the temporary pessimistic value function by

$$\check{Q}_h^k(\cdot, \cdot) = \langle \phi(\cdot, \cdot), \theta_{h,k-1} + \mu_{h,k-1} \overline{V}_{h+1}^k \rangle - \beta \|\phi(\cdot, \cdot)\|_{H_{h,k-1}^{-1}}$$

where $\beta := \beta_R + \beta_V = \tilde{\mathcal{O}}(\sqrt{d})$. The actual optimistic value function $\overline{Q}_h^k(\cdot, \cdot)$ is the minimum function of history temporary optimistic value functions $\widehat{Q}_h^k(\cdot, \cdot)$, and the actual pessimistic value function $\underline{Q}_h^k(\cdot, \cdot)$ is the maximum function of history temporary pessimistic value functions $\check{Q}_h^k(\cdot, \cdot)$ (Line 7 in Algorithm 3). In this way, $\overline{Q}_h^k(\cdot, \cdot)$ is always non-increasing in $k$ and $\underline{Q}_h^k(\cdot, \cdot)$ is always non-decreasing in $k$.

The second is the rare-switching value function update, which updates the value function only when the determinant of the covariance matrix significantly exceeds the previous value (Line 6 in Algorithm 3). This approach allows the complexity, as measured by the metric entropy, of the function class to which $\overline{V}_h^k(\cdot)$ or $\underline{V}_h^k(\cdot)$ belongs to be independent of $K$. Notably, the metric entropy is linearly dependent on $\tilde{\mathcal{O}}(dH)$. Moreover, on the event equation B.1, we can establish optimism and pessimism in Lemma D.4, i.e., for all $k \in [K]$ and $h \in [H]$,

$$\underline{V}_{h+1}^k(\cdot) \leqslant V_{h+1}^*(\cdot) \leqslant \overline{V}_{h+1}^k(\cdot). \tag{B.2}$$

Directly estimating the variance of the optimistic value function $\overline{V}_{h+1}^k(\cdot)$ will encounter the dependence issue, which is discussed in (Jin et al., 2020b) and will introduce an additional $\sqrt{d}$ factor in the regret due to the covering-based decoupling argument. To eliminate this factor, after noting the inequality

$$[\mathbb{V}_h \overline{V}_{h+1}^k](\cdot, \cdot) \leqslant 2[\mathbb{V}_h V_{h+1}^*](\cdot, \cdot) + 2[\mathbb{V}_h(\overline{V}_{h+1}^k - V_{h+1}^*)](\cdot, \cdot),$$

Hu et al. (2022) decompose the optimistic value function $\overline{V}_{h+1}^k(\cdot)$ into the optimal value function $V_{h+1}^*(\cdot)$ and the sub-optimality gap $[\overline{V}_{h+1}^k - V_{h+1}^*](\cdot)$ and estimate their variances $[\mathbb{V}_h V_{h+1}^*](\cdot, \cdot)$ and $[\mathbb{V}_h(\overline{V}_{h+1}^k - V_{h+1}^*)](\cdot, \cdot)$ separately. The key insight is that: (i) as $V_{h+1}^*$ is deterministic, there is no additional $\sqrt{d}$ dependence in estimating $[\mathbb{V}_h V_{h+1}^*](\cdot, \cdot)$, and (ii) as $\overline{V}_{h+1}^k$ gradually converges to $V_{h+1}^*$, though a uniform argument is still used, the incurred $\sqrt{d}$ factor in the estimation of $[\mathbb{V}_h(\overline{V}_{h+1}^k - V_{h+1}^*)]$ has ignorable effects on the final regret. We now describe the way we estimate these two variances.

- For $[\mathbb{V}_h V_{h+1}^*](\cdot, \cdot)$, since $V_{h+1}^*$ is unknown, a natural choice is to estimate it by the optimistic value function $\overline{V}_{h+1}^k$. Hence, we estimate $[\mathbb{V}_h V_{h+1}^*](s_{h,k}, a_{h,k})$ via

$$[\widehat{\mathbb{V}}_{h,k} \overline{V}_{h+1}^k](s_{h,k}, a_{h,k}) := [\widehat{\mathbb{P}}_{h,k}(\overline{V}_{h+1}^k)^2](s_{h,k}, a_{h,k})_{[0, \mathcal{H}^2]} - \left[[\widehat{\mathbb{P}}_{h,k}\overline{V}_{h+1}^k](s_{h,k}, a_{h,k})_{[0, \mathcal{H}^2]}\right]^2. \tag{B.3}$$

To measure estimation accuracy, we introduce an error term $U_{h,k}$ to guarantee that with high probability, $\left|[\widehat{\mathbb{V}}_{h,k}\overline{V}_{h+1}^k](s_{h,k}, a_{h,k}) - [\mathbb{V}_h V_{h+1}^*](s_{h,k}, a_{h,k})\right| \leqslant U_{h,k}$ holds uniformly over all $h, k$ where

$$U_{h,k} = \min\left\{\mathcal{V}^2, 11\mathcal{H}\beta_0 \cdot \|\phi_{h,k}\|_{\boldsymbol{H}_{h,k-1}^{-1}} + 4\mathcal{H} \cdot \widehat{\mathbb{P}}_{h,k}(\overline{V}_{h+1}^k - \underline{V}_{h+1}^k)(s_{h,k}, a_{h,k})\right\} \tag{A.9}$$

and $\beta_0 = \widetilde{\mathcal{O}}\left(\frac{\mathcal{H}}{\sigma_{\min}}\sqrt{d^3 H}\right)$ is an exploration radius.

- For $[\mathbb{V}_h(\overline{V}_{h+1}^k - V_{h+1}^*)](\cdot, \cdot)$, to meet the measurability condition of a concentration inequality (Lemma G.3), we require

$$\sigma_{h,k}^2 \geqslant d^3 H \cdot \sup_{k \leqslant j \leqslant K}[\mathbb{V}_h(\overline{V}_{h+1}^j - V_{h+1}^*)](s_{h,k}, a_{h,k}). \tag{B.4}$$

Note that $\sigma_{h,k}$ is $\mathcal{F}_{h,k}$-measurable while $\overline{V}_{h+1}^k(\cdot)$ is $\mathcal{F}_{H,k-1}$-measurable. The condition equation B.4 essentially requires a $\mathcal{F}_{h,k}$-measurable upper bound for $[\mathbb{V}_h(\overline{V}_{h+1}^j - V_{h+1}^*)](s_{h,k}, a_{h,k})$ even if $j \geqslant k$. Fortunately, we have for any $k \leqslant j \leqslant K$,

$$\begin{aligned}
[\mathbb{V}_h(\overline{V}_{h+1}^j - V_{h+1}^*)](s_{h,k}, a_{h,k}) &\leqslant [\mathbb{P}_h(\overline{V}_{h+1}^j - V_{h+1}^*)^2](s_{h,k}, a_{h,k}) \\
&\overset{(a)}{\leqslant} \mathcal{H}[\mathbb{P}_h(\overline{V}_{h+1}^j - V_{h+1}^*)](s_{h,k}, a_{h,k}) \\
&\overset{(b)}{\leqslant} \mathcal{H}[\mathbb{P}_h(\overline{V}_{h+1}^j - \underline{V}_{h+1}^j)](s_{h,k}, a_{h,k}) \\
&\overset{(c)}{\leqslant} \mathcal{H}[\mathbb{P}_h(\overline{V}_{h+1}^k - \underline{V}_{h+1}^k)](s_{h,k}, a_{h,k})
\end{aligned} \tag{B.5}$$

where $(a)$ uses $|\overline{V}_{h+1}^j - V_{h+1}^*|(\cdot) \leqslant \mathcal{H}$ and the optimism of $\overline{V}_{h+1}^j(\cdot)$, $(b)$ follows from the pessimism in equation B.2, and $(c)$ uses the monotonicity of value functions. The RHS of equation B.5 is $\mathcal{F}_{h,k}$-measurable but intractable due to the population expectation $\mathbb{P}_h(\cdot)$. By replacing $\mathbb{P}_h(\cdot)$ with the tractable $\widehat{\mathbb{P}}_{h,k}(\cdot)$, we introduce $E_{h,k}$ to overestimate the RHS of equation B.5 where

$$E_{h,k} = \min\left\{\mathcal{H}^2, 2\mathcal{H}\beta_0 \cdot \|\phi_{h,k}\|_{\boldsymbol{H}_{h,k-1}^{-1}} + \mathcal{H} \cdot \left[\widehat{\mathbb{P}}_{h,k}(\overline{V}_{h+1}^k - \underline{V}_{h+1}^k)\right](s_{h,k}, a_{h,k})\right\}. \tag{A.7}$$

Hence, equation B.4 is guaranteed by $\sigma_{h,k}^2 \geqslant d^3 H \cdot E_{h,k}$. The extra $d^3 H$ factor is introduced to offset the error caused by the covering number argument.

## C  Proof for Section 2.3

### C.1  Proof of Theorem 2.2

Now, we turn to the regret equation 2.1. Recall that at iteration $t$, we set

$$(\phi_t, *) = \underset{\phi \in \mathcal{D}_t, \boldsymbol{\theta} \in \mathcal{C}_{t-1}}{\operatorname{argmax}} \langle \phi, \boldsymbol{\theta} \rangle.$$

Due to $\sup_{\phi \in \bigcup_{t \geqslant 0} \mathcal{D}_t} |\langle \phi, \boldsymbol{\theta}^* \rangle| \leqslant R := LB$, it follows that

$$\operatorname{Reg}(T) := \sum_{t=1}^{T}\left[\sup_{\phi \in \mathcal{D}_t}\langle \phi, \boldsymbol{\theta}^* \rangle - \langle \phi_t, \boldsymbol{\theta}^* \rangle\right]$$

$$\leqslant \sum_{t=1}^{T} \left[ \sup_{\phi \in \mathcal{D}_t, \boldsymbol{\theta} \in \mathcal{C}_{t-1}} \langle \phi, \boldsymbol{\theta} \rangle - \langle \phi_t, \boldsymbol{\theta}^* \rangle \right]$$

$$= \sum_{t=1}^{T} \left[ \sup_{\boldsymbol{\theta} \in \mathcal{C}_{t-1}} \langle \phi_t, \boldsymbol{\theta} \rangle - \langle \phi_t, \boldsymbol{\theta}^* \rangle \right]$$

$$\leqslant \sum_{t=1}^{T} \|\phi_t\|_{\boldsymbol{H}_{t-1}^{-1}(\boldsymbol{\theta}_{t-1})} \cdot \sup_{\boldsymbol{\theta} \in \mathcal{C}_{t-1}} \|\boldsymbol{\theta} - \boldsymbol{\theta}^*\|_{\boldsymbol{H}_{t-1}}$$

Notice that with probability $1 - \delta$, $\boldsymbol{\theta}^* \in \mathcal{C}_t$ for all $t \geqslant 1$, i.e., $\|\boldsymbol{\theta}_t - \boldsymbol{\theta}^*\|_{\boldsymbol{H}_t} \leqslant \beta_t$. Hence,

$$\sup_{\boldsymbol{\theta} \in \mathcal{C}_t} \|\boldsymbol{\theta} - \boldsymbol{\theta}^*\|_{\boldsymbol{H}_t} \leqslant \sup_{\boldsymbol{\theta} \in \mathcal{C}_t} \|\boldsymbol{\theta} - \boldsymbol{\theta}_t\|_{\boldsymbol{H}_t} + \|\boldsymbol{\theta}_t - \boldsymbol{\theta}^*\|_{\boldsymbol{H}_t} \leqslant 2\beta_t.$$

Notice that $\beta_t$ is increasing in $t$ and $w_t = \left\| \frac{\phi_t}{\sigma_t} \right\|_{\boldsymbol{H}_{t-1}^{-1}}$. Therefore,

$$\text{Reg}(T) \leqslant 2\beta_T \sum_{t=1}^{T} \|\phi_t\|_{\boldsymbol{H}_{t-1}^{-1}} = 2\beta_T \sum_{t=1}^{T} \sigma_t w_t = 2\beta_T \sum_{t=1}^{T} \sigma_t \min\{1, w_t\}. \tag{C.1}$$

The last equality uses $w_t \leqslant 1$ (which is due to $\sigma_t \geqslant \|\phi_t\|_{\boldsymbol{H}_{t-1}^{-1}}/c_0$ and $c_0 \leqslant 1$). Notice that $\|\phi_t\|/\sigma_t \leqslant \|\phi_t\|/\sigma_{\min} \leqslant L/\sigma_{\min}$. Then by Lemma G.5,

$$\sum_{t=1}^{T} \min \left\{ 1, \left\| \frac{\phi_t}{\sigma_t} \right\|_{\boldsymbol{H}_{t-1}^{-1}}^2 \right\} = \sum_{t=1}^{T} \min \left\{ 1, w_t^2 \right\} \leqslant 2d \log \left( 1 + \frac{TL^2}{d\lambda \sigma_{\min}^2} \right) = 2\kappa. \tag{C.2}$$

Recall that

$$\sigma_t = \max \left\{ \nu_t, \sigma_{\min}, \frac{\|\phi_t\|_{\boldsymbol{H}_{t-1}^{-1}}}{c_0}, \frac{\sqrt{LB}\|\phi_t\|_{\boldsymbol{H}_{t-1}^{-1}}^{\frac{1}{2}}}{c_1^{\frac{1}{4}} d^{\frac{1}{4}}} \right\}.$$

According to what value $\sigma_t$ takes, we decompose $[T]$ into three sets $[T] \subseteq \bigcup_{i=1}^{3} \mathcal{J}_i$ where

$$\mathcal{J}_1 = \{t \in [T] : \sigma_t \in \{\nu_t, \sigma_{\min}\}\},$$

$$\mathcal{J}_2 = \left\{ t \in [T] : \sigma_t = \frac{\|\phi_t\|_{\boldsymbol{H}_{t-1}^{-1}}}{c_0} \right\},$$

$$\mathcal{J}_3 = \left\{ t \in [T] : \sigma_t = \sqrt{LB} \frac{\|\phi_t\|_{\boldsymbol{H}_{t-1}^{-1}}^{\frac{1}{2}}}{c_1^{\frac{1}{4}} d^{\frac{1}{4}}} \right\}.$$

First, it follows that

$$\begin{aligned}
\sum_{t \in \mathcal{J}_1} \sigma_t \min \{1, w_t\} &\leqslant \sum_{t \in \mathcal{J}_1} \max\{\nu_t, \sigma_{\min}\} \min \{1, w_t\} \\
&\leqslant \sum_{t \in [T]} \max\{\nu_t, \sigma_{\min}\} \min \{1, w_t\} \\
&\overset{(a)}{\leqslant} \sqrt{\sum_{t \in [T]} (\nu_t^2 + \sigma_{\min}^2)} \sqrt{\sum_{t \in [T]} \min \{1, w_t^2\}} \\
&\overset{(b)}{\leqslant} \sqrt{2\kappa} \cdot \sqrt{\sum_{t \in [T]} \nu_t^2 + 1}. \tag{C.3}
\end{aligned}$$

Here $(a)$ holds due to Cauchy-Schwarz inequality and $(b)$ uses equation C.2 and $\sigma_{\min} = \frac{1}{\sqrt{T}}$.

Second, for any $t \in \mathcal{J}_2$, we have $w_t = \left\| \frac{\phi_t}{\sigma_t} \right\|_{\boldsymbol{H}_{t-1}^{-1}} = c_0 \leqslant 1$. Therefore,

$$
\begin{aligned}
\sum_{t \in \mathcal{J}_2} \sigma_t \min\{1, w_t\} = \sum_{t \in \mathcal{J}_2} \sigma_t w_t &= \frac{1}{c_0} \sum_{t \in \mathcal{J}_2} \sigma_t w_t^2 \leqslant \frac{\sup_{t \in \mathcal{J}_2} \sigma_t}{c_0} \sum_{t \in \mathcal{J}_2} w_t^2 \\
&\leqslant \frac{\sup_{t \in [T]} \|\phi_t\|_{\boldsymbol{H}_{t-1}^{-1}}}{c_0^2} \cdot \sum_{t \in \mathcal{J}_2} \min\{1, w_t^2\} \\
&\leqslant \frac{\sup_{t \in [T]} \|\phi_t\|_{\boldsymbol{H}_{t-1}^{-1}}}{c_0^2} \cdot \sum_{t \in [T]} \min\{1, w_t^2\} \leqslant \frac{2L\kappa}{c_0^2 \sqrt{\lambda}}
\end{aligned}
\tag{C.4}
$$

where the last inequality uses $\|\phi_t\|_{\boldsymbol{H}_{t-1}^{-1}} \leqslant \frac{1}{\sqrt{\lambda}} \|\phi_t\| \leqslant \frac{L}{\sqrt{\lambda}}$ for all $t \geqslant 1$ and equation C.2.

Finally, for any $t \in \mathcal{J}_3$, we have $L^2 B^2 w_t^2 = c_1 d \sigma_t^2$ due to $w_t^2 = \left\| \frac{\phi_t}{\sigma_t} \right\|_{\boldsymbol{H}_{t-1}^{-1}}^2$. It implies $\sigma_t = L B w_t / \sqrt{c_1 d} = L B \min\{1, w_t\}/\sqrt{c_1 d}$ with the fact that $w_t \leqslant 1$. Therefore,

$$
\sum_{t \in \mathcal{J}_3} \sigma_t \min\{1, w_t\} = \frac{LB}{\sqrt{c_1 d}} \cdot \sum_{t \in \mathcal{J}_3} \min\{1, w_t^2\} \leqslant \frac{LB}{\sqrt{c_1 d}} \cdot \sum_{t \in [T]} \min\{1, w_t^2\} \leqslant \frac{2LB\kappa}{\sqrt{c_1 d}}.
\tag{C.5}
$$

Plugging equation C.3, equation C.4 and equation C.5 into equation C.1, we have

$$
\text{Reg}(T) \leqslant 2\beta_T \left[ \sqrt{2\kappa} \cdot \sqrt{\sum_{t \in [T]} \nu_t^2 + 1} + \frac{2L\kappa}{c_0^2 \sqrt{\lambda}} + \frac{2LB\kappa}{\sqrt{c_1 d}} \right].
$$

## C.2   Proof of Lemma 2.1

Recall that $z_t(\boldsymbol{\theta}) = \frac{y_t - \langle \phi_t, \boldsymbol{\theta} \rangle}{\sigma_t}$. Direct computation yields that

$$
\nabla^2 L_T(\boldsymbol{\theta}) = \lambda \boldsymbol{I} + \sum_{t=1}^{T} \left( \frac{\tau_t}{\sqrt{\tau_t^2 + z_t^2(\boldsymbol{\theta})}} \right)^3 \frac{\phi_t \phi_t^\top}{\sigma_t^2}.
$$

Clearly, for any $\boldsymbol{\theta} \in \mathbb{R}^d$,

$$
\nabla^2 L_T(\boldsymbol{\theta}) \preceq \lambda \boldsymbol{I} + \sum_{t=1}^{T} \frac{\phi_t \phi_t^\top}{\sigma_t^2} = \boldsymbol{H}_T.
$$

For the other direction, we decompose it into four terms and analyze them respectively.

$$
\begin{aligned}
\nabla^2 L_T(\boldsymbol{\theta}) = \boldsymbol{H}_T &- \underbrace{\sum_{t=1}^{T} \left[ 1 - \left( \frac{\tau_t}{\sqrt{\tau_t^2 + z_t^2(\boldsymbol{\theta}^*)}} \right)^3 \right] \frac{\phi_t \phi_t^\top}{\sigma_t^2}}_{\boldsymbol{H}_{1,T}} \\
&+ \underbrace{\sum_{t=1}^{T} \left[ \left( \frac{\tau_t}{\sqrt{\tau_t^2 + z_t^2(\boldsymbol{\theta})}} \right)^3 - \left( \frac{\tau_t}{\sqrt{\tau_t^2 + z_t^2(\boldsymbol{\theta}^*)}} \right)^3 \right] \frac{\phi_t \phi_t^\top}{\sigma_t^2}}_{\boldsymbol{H}_{2,T}}.
\end{aligned}
\tag{C.6}
$$

where $\mathbb{E}_t[\cdot] = \mathbb{E}[\cdot | \mathcal{F}_{t-1}]$ for simplicity.

Since $\nu_t, \boldsymbol{\theta}_{t-1} \in \mathcal{F}_{t-1}$, from Algorithm 1, we have $\sigma_t, w_t, \tau_t \in \mathcal{F}_{t-1}$.

**Analysis of $\boldsymbol{H}_{1,T}$** Notice that for any unit norm $\boldsymbol{v} \in \mathbb{R}^d$, it follows that

$$
\begin{aligned}
\boldsymbol{v}^\top \boldsymbol{H}_{1,T} \boldsymbol{v} &= \sum_{t=1}^{T} \left[ 1 - \left( \frac{\tau_t}{\sqrt{\tau_t^2 + z_t^2(\boldsymbol{\theta}^*)}} \right)^3 \right] \left\langle \frac{\boldsymbol{\phi}_t}{\sigma_t}, \boldsymbol{v} \right\rangle^2 \\
&\leqslant 3 \sum_{t=1}^{T} \left[ 1 - \frac{\tau_t}{\sqrt{\tau_t^2 + z_t^2(\boldsymbol{\theta}^*)}} \right] \left\langle \frac{\boldsymbol{\phi}_t}{\sigma_t}, \boldsymbol{v} \right\rangle^2 \\
&\leqslant 3 \sum_{t=1}^{T} \left[ 1 - \frac{\tau_t}{\sqrt{\tau_t^2 + z_t^2(\boldsymbol{\theta}^*)}} \right] \cdot \sup_{t \in [T]} \left\langle \frac{\boldsymbol{\phi}_t}{\sigma_t}, \boldsymbol{v} \right\rangle^2 \\
&\leqslant 3 \sum_{t=1}^{T} \left[ 1 - \frac{\tau_t}{\sqrt{\tau_t^2 + z_t^2(\boldsymbol{\theta}^*)}} \right] \cdot \sup_{t \in [T]} \left\| \frac{\boldsymbol{\phi}_t}{\sigma_t} \right\|_{\boldsymbol{H}_T^{-1}}^2 \cdot \boldsymbol{v}^\top \boldsymbol{H}_T \boldsymbol{v} \\
&\overset{(a)}{\leqslant} 3 \sum_{t=1}^{T} \left[ 1 - \frac{\tau_t}{\sqrt{\tau_t^2 + z_t^2(\boldsymbol{\theta}^*)}} \right] \cdot \sup_{t \in [T]} \left\| \frac{\boldsymbol{\phi}_t}{\sigma_t} \right\|_{\boldsymbol{H}_t^{-1}}^2 \cdot \boldsymbol{v}^\top \boldsymbol{H}_T \boldsymbol{v} \\
&\overset{(b)}{=} 3 \sum_{t=1}^{T} \left[ 1 - \frac{\tau_t}{\sqrt{\tau_t^2 + z_t^2(\boldsymbol{\theta}^*)}} \right] \cdot \sup_{t \in [T]} \frac{w_t^2}{1 + w_t^2} \cdot \boldsymbol{v}^\top \boldsymbol{H}_T \boldsymbol{v},
\end{aligned}
$$

where $(a)$ uses $\boldsymbol{H}_T^{-1} \preceq \boldsymbol{H}_t^{-1}$ for all $t \in [T]$ and $(b)$ follows from

$$
\left\| \frac{\boldsymbol{\phi}_t}{\sigma_t} \right\|_{\boldsymbol{H}_t^{-1}}^2 = \frac{\boldsymbol{\phi}_t^\top}{\sigma_t} \left( \boldsymbol{H}_{t-1}^{-1} - \frac{\boldsymbol{H}_{t-1}^{-1} \frac{\boldsymbol{\phi}_t}{\sigma_t} \frac{\boldsymbol{\phi}_t^\top}{\sigma_t} \boldsymbol{H}_{t-1}^{-1}}{1 + \frac{\boldsymbol{\phi}_t^\top}{\sigma_t} \boldsymbol{H}_{t-1}^{-1} \frac{\boldsymbol{\phi}_t}{\sigma_t}} \right) \frac{\boldsymbol{\phi}_t}{\sigma_t} = w_t^2 - \frac{w_t^4}{1 + w_t^2} = \frac{w_t^2}{1 + w_t^2}.
$$

By the arbitrariness of $\boldsymbol{v}$, we know that

$$
\boldsymbol{H}_{1,T} \preceq 3 \sum_{t=1}^{T} \left[ 1 - \frac{\tau_t}{\sqrt{\tau_t^2 + z_t^2(\boldsymbol{\theta}^*)}} \right] \cdot \sup_{t \in [T]} \frac{w_t^2}{1 + w_t^2} \cdot \boldsymbol{H}_T. \tag{C.7}
$$

Let $X_t = 1 - \frac{\tau_t}{\sqrt{\tau_t^2 + z_t^2(\boldsymbol{\theta}^*)}}$. It is obvious that $0 \leqslant X_t \leqslant 1$. We then focus on the concentration of $\sum_{t=1}^{T} X_t$. To that end, we need a variance-aware Bernstein's inequality Lemma G.2 for martingales. Lemma G.2 implies that with probability at least $1 - \frac{\delta}{T^2}$, we have

$$
\sum_{t=1}^{T} X_t \leqslant \sum_{t=1}^{T} \mathbb{E}_t X_t + 3 \sqrt{\sum_{t=1}^{T} \mathrm{Var}[X_t | \mathcal{F}_{t-1}] \cdot \log \frac{2KT^2}{\delta}} + 5 \log \frac{2KT^2}{\delta}
$$

where $K := 1 + \lceil 2 \log_2 V \rceil$ and $V^2$ is an upper bound satisfying $\sum_{t=1}^{T} \mathbb{E}[X_t^2 | \mathcal{F}_{t-1}] \leqslant V^2$.

First notice that for any $t \geqslant 1$, we have

$$
\begin{aligned}
\mathbb{E}_t X_t = 1 - \mathbb{E}_t \frac{\tau_t}{\sqrt{\tau_t^2 + z_t^2(\boldsymbol{\theta}^*)}} &= \mathbb{E}_t \frac{z_t^2(\boldsymbol{\theta}^*)}{\sqrt{\tau_t^2 + z_t^2(\boldsymbol{\theta}^*)} (\sqrt{\tau_t^2 + z_t^2(\boldsymbol{\theta}^*)} + \tau_t)} \\
&\leqslant \frac{1}{2\tau_t^2} \mathbb{E}_t z_t^2(\boldsymbol{\theta}^*) \leqslant \frac{b^2}{2\tau_t^2} \leqslant \frac{b^2}{2\tau_0^2} \frac{w_t^2}{1 + w_t^2}
\end{aligned} \tag{C.8}
$$

which implies that

$$
\sum_{t=1}^{T} \mathbb{E}_t X_t \leqslant \frac{b^2}{2\tau_0^2} \frac{w_t^2}{1 + w_t^2} \leqslant \frac{b^2}{2\tau_0^2} \sum_{t=1}^{T} \min\{1, w_t^2\} \leqslant \frac{\kappa b^2}{\tau_0^2}
$$

where the last inequality uses Lemma G.5 and thus

$$
\sum_{t=1}^{T} \min\{1, w_t^2\} = \sum_{t=1}^{T} \min \left\{ 1, \left\| \frac{\boldsymbol{\phi}_t}{\sigma_t} \right\|_{\boldsymbol{H}_{t-1}^{-1}}^2 \right\} \leqslant 2d \log \left( 1 + \frac{TL^2}{d\lambda \sigma_{\min}^2} \right) = 2\kappa.
$$

Secondly, we have

$$\mathrm{Var}[X_t|\mathcal{F}_{t-1}] \leqslant \mathbb{E}[X_t^2|\mathcal{F}_{t-1}] \leqslant \mathbb{E}_t\left(1 - \frac{\tau_t}{\sqrt{\tau_t^2 + z_t^2(\boldsymbol{\theta}^*)}}\right)^2 \overset{(*)}{\leqslant} \frac{1}{4}\frac{\mathbb{E}_t z_t^2(\boldsymbol{\theta}^*)}{\tau_t^2} \leqslant \frac{b^2}{4\tau_t^2}$$

where $(*)$ uses $1 - \frac{\tau_t}{\sqrt{\tau_t^2 + z_t^2(\boldsymbol{\theta}^*)}} \leqslant \frac{z_t^2(\boldsymbol{\theta}^*)}{2\tau_t\sqrt{\tau_t^2 + z_t^2(\boldsymbol{\theta}^*)}}$ which is also used in equation C.8. As a result, we have

$$\sum_{t=1}^{T} \mathrm{Var}[X_t|\mathcal{F}_{t-1}] \leqslant \sum_{t=1}^{T} \frac{b^2}{4\tau_t^2} \leqslant \frac{b^2}{4\tau_0^2}\sum_{t=1}^{T}\frac{w_t^2}{1+w_t^2} \leqslant \frac{\kappa b^2}{2\tau_0^2}.$$

Once requiring $\tau_0^2 \geqslant 2\kappa b^2$, we have $\sum_{t=1}^{T}\mathrm{Var}[X_t|\mathcal{F}_{t-1}] \leqslant 1$ and thus we can set $V = 1$ and obtain $K = 1$. Putting them together, if $\tau_0^2 \geqslant \frac{2\kappa b^2}{\log\frac{2T^2}{\delta}}$, with probability at least $1 - \delta$, we have

$$\begin{aligned}
\sum_{t=1}^{T} X_t &\leqslant \frac{\kappa b^2}{\tau_0^2} + \frac{3b}{\tau_0}\sqrt{\frac{\kappa\log\frac{2T^2}{\delta}}{2}} + 5\log\frac{2T^2}{\delta} \\
&\leqslant \frac{1}{2}\log\frac{2T^2}{\delta} + \frac{3}{2}\log\frac{2T^2}{\delta} + 5\log\frac{2T^2}{\delta} \\
&\leqslant 9\log\frac{2T^2}{\delta} = \frac{1}{12c_0^2}
\end{aligned} \tag{C.9}$$

where the last equation is due to the definition of $c_0$. Finally, taking a union bound for the last inequality from $T = 1$ to $\infty$ and using the fact that $\sum_{t=1}^{\infty} t^{-2} < 2$, we have $\sum_{t=1}^{T} X_t \leqslant \frac{1}{12c_0^2}$ for all $T \geqslant 1$ with probability at least $1 - 2\delta$.

On the other hand, by the choice of $\sigma_t$, we have $\sigma_t^2 \geqslant \frac{1}{c_0^2}\cdot\|\boldsymbol{\phi}_t\|^2_{\boldsymbol{H}_{t-1}^{-1}}$, which implies

$$\sup_{t\in[T]}\frac{w_t^2}{1+w_t^2} \leqslant \sup_{t\in[T]} w_t^2 \leqslant c_0^2. \tag{C.10}$$

Plugging equation C.9 and equation C.10 into equation C.7, we have

$$\boldsymbol{H}_{1,T} \preceq \frac{1}{4}\boldsymbol{H}_T. \tag{C.11}$$

**Analysis of $\boldsymbol{H}_{2,T}$** We first notice that

$$\begin{aligned}
\left|\left(\frac{\tau_t}{\sqrt{\tau_t^2 + z_t^2(\boldsymbol{\theta})}}\right)^3 - \left(\frac{\tau_t}{\sqrt{\tau_t^2 + z_t^2(\boldsymbol{\theta}^*)}}\right)^3\right| &\leqslant 3\left|\frac{\tau_t}{\sqrt{\tau_t^2 + z_t^2(\boldsymbol{\theta})}} - \frac{\tau_t}{\sqrt{\tau_t^2 + z_t^2(\boldsymbol{\theta}^*)}}\right| \\
&\leqslant \frac{3\tau_t}{\sqrt{\tau_t^2 + z_t^2(\boldsymbol{\theta})}\sqrt{\tau_t^2 + z_t^2(\boldsymbol{\theta}^*)}}\frac{|z_t^2(\boldsymbol{\theta}) - z_t^2(\boldsymbol{\theta}^*)|}{\sqrt{\tau_t^2 + z_t^2(\boldsymbol{\theta})} + \sqrt{\tau_t^2 + z_t^2(\boldsymbol{\theta}^*)}}.
\end{aligned} \tag{C.12}$$

Notice that $z_t(\boldsymbol{\theta}) = z_t(\boldsymbol{\theta}^*) + \langle\frac{\boldsymbol{\phi}_t}{\sigma_t}, \boldsymbol{\theta} - \boldsymbol{\theta}^*\rangle$. It then follows that for any $c > 0$

$$z_t^2(\boldsymbol{\theta}) \leqslant \left(1 + \frac{1}{c}\right) z_t^2(\boldsymbol{\theta}^*) + (1+c)\left\langle\frac{\boldsymbol{\phi}_t}{\sigma_t}, \boldsymbol{\theta} - \boldsymbol{\theta}^*\right\rangle^2;$$

$$z_t^2(\boldsymbol{\theta}^*) \leqslant \left(1 + \frac{1}{c}\right) z_t^2(\boldsymbol{\theta}) + (1+c)\left\langle\frac{\boldsymbol{\phi}_t}{\sigma_t}, \boldsymbol{\theta} - \boldsymbol{\theta}^*\right\rangle^2,$$

By discussing which is larger between $z_t^2(\boldsymbol{\theta})$ and $z_t^2(\boldsymbol{\theta}^*)$, we have

$$|z_t^2(\boldsymbol{\theta}) - z_t^2(\boldsymbol{\theta}^*)| \leqslant \frac{1}{c}\min\left\{z_t^2(\boldsymbol{\theta}), z_t^2(\boldsymbol{\theta}^*)\right\} + (1+c)\left\langle\frac{\boldsymbol{\phi}_t}{\sigma_t}, \boldsymbol{\theta} - \boldsymbol{\theta}^*\right\rangle^2. \tag{C.13}$$

Plugging equation C.13 into equation C.12, we have that

$$\left| \left( \frac{\tau_t}{\sqrt{\tau_t^2 + z_t^2(\boldsymbol{\theta})}} \right)^3 - \left( \frac{\tau_t}{\sqrt{\tau_t^2 + z_t^2(\boldsymbol{\theta}^*)}} \right)^3 \right|$$

$$\leq \frac{3\tau_t}{\tau_t^2 + \min\left\{z_t^2(\boldsymbol{\theta}), z_t^2(\boldsymbol{\theta}^*)\right\}} \frac{\frac{1}{c}\min\left\{z_t^2(\boldsymbol{\theta}), z_t^2(\boldsymbol{\theta}^*)\right\}}{2\sqrt{\tau_t^2 + \min\left\{z_t^2(\boldsymbol{\theta}), z_t^2(\boldsymbol{\theta}^*)\right\}}} + \frac{3(1+c)}{2\tau_t^2} \left\langle \frac{\boldsymbol{\phi}_t}{\sigma_t}, \boldsymbol{\theta} - \boldsymbol{\theta}^* \right\rangle^2$$

$$\leq \frac{3}{2c} + \frac{3(1+c)}{2\tau_t^2} \left\langle \frac{\boldsymbol{\phi}_t}{\sigma_t}, \boldsymbol{\theta} - \boldsymbol{\theta}^* \right\rangle^2 \overset{(a)}{\leq} \frac{3}{2c} + \frac{6(1+c)}{\tau_t^2} \frac{L^2 B^2}{\sigma_t^2}$$

$$\leq \frac{3}{2c} + \frac{6(1+c)}{\tau_0^2} \frac{w_t^2 L^2 B^2}{\sigma_t^2} \overset{(b)}{\leq} \frac{3}{2c} + \frac{6(1+c)c_1 d}{\tau_0^2} \tag{C.14}$$

where $(a)$ uses $\left\langle \frac{\boldsymbol{\phi}_t}{\sigma_t}, \boldsymbol{\theta} - \boldsymbol{\theta}^* \right\rangle \leq \left\| \frac{\boldsymbol{\phi}_t}{\sigma_t} \right\| (\|\boldsymbol{\theta}\| + \|\boldsymbol{\theta}^*\|) \leq \frac{2LB}{\sigma_t}$ due to $\|\boldsymbol{\phi}_t\| \leq L$ and $\boldsymbol{\theta}, \boldsymbol{\theta}^* \in \text{Ball}_d(B)$ and $(b)$ uses the following result. By the definition of $\sigma_t$, we have $\sigma_t \geq \sqrt{LB}\|\boldsymbol{\phi}_t\|_{\boldsymbol{H}_{t-1}^{-1}}^{\frac{1}{2}} / c_1^{\frac{1}{4}} d^{\frac{1}{4}}$ which implies $\sigma_t^2 \geq \frac{w_t^2 L^2 B^2}{c_1 d}$. As a result of equation C.14, by definition of $\boldsymbol{H}_{3,T}$, we have

$$-\left( \frac{3}{2c} + \frac{6(1+c)c_1 d}{\tau_0^2} \right) \sum_{t=1}^T \frac{\boldsymbol{\phi}_t}{\sigma_t} \frac{\boldsymbol{\phi}_t^\top}{\sigma_t} \preceq \boldsymbol{H}_{2,T} \tag{C.15}$$

**Putting pieces together**   Plugging equation C.11 and equation C.15 into equation C.6, with probability at least $1 - \delta$, for any $T \geq 1$ and for all $\boldsymbol{\theta} \in \text{Ball}_d(B)$, we have

$$\nabla^2 L_T(\boldsymbol{\theta}) \succeq \boldsymbol{H}_T - \frac{1}{4}\boldsymbol{H}_T - \left( \frac{3}{2c} + \frac{6(1+c)c_1 d}{\tau_0^2} \right) \sum_{t=1}^T \frac{\boldsymbol{\phi}_t}{\sigma_t} \frac{\boldsymbol{\phi}_t^\top}{\sigma_t}$$

$$\succeq \frac{3\lambda}{4}\boldsymbol{I} + \left( 1 - \frac{1}{4} - \frac{3}{2c} - \frac{6(1+c)c_1 d}{\tau_0^2} \right) \sum_{t=1}^T \frac{\boldsymbol{\phi}_t}{\sigma_t} \frac{\boldsymbol{\phi}_t^\top}{\sigma_t}.$$

Notice that $c_1 = \frac{1}{42 \cdot \log \frac{2T^2}{\delta}}$. If we set $c = 6$ and $\tau_0 \sqrt{\frac{2T^2}{\delta}} \geq \max\{\sqrt{2\kappa}b, 2\sqrt{d}\}$, we have

$$\max\left\{ \frac{3}{2c}, \frac{6(1+c)c_1 d}{\tau_0^2} \right\} \leq \frac{1}{4}.$$

As a result, we have

$$\nabla^2 L_T(\boldsymbol{\theta}) \succeq \frac{3\lambda}{4}\boldsymbol{I} + \frac{1}{4} \sum_{t=1}^T \frac{\boldsymbol{\phi}}{\sigma_t} \frac{\boldsymbol{\phi}^\top}{\sigma_t} \succeq \frac{1}{4}\boldsymbol{H}_T.$$

## C.3   Proof of Lemma 2.2

For simplicity, we denote $z_t^* = z_t(\boldsymbol{\theta}^*)$ for short. By triangle inequality, we have

$$\|\nabla L_T(\boldsymbol{\theta}^*)\|_{\boldsymbol{H}_T^{-1}} \leq \|\lambda\boldsymbol{\theta}^*\|_{\boldsymbol{H}_T^{-1}} + \left\| \sum_{t=1}^T \frac{\tau_t z_t^*}{\sqrt{\tau_t^2 + (z_t^*)^2}} \frac{\boldsymbol{\phi}_t}{\sigma_t} \right\|_{\boldsymbol{H}_T^{-1}}$$

$$\leq \|\lambda\boldsymbol{\theta}^*\|_{\boldsymbol{H}_T^{-1}} + \underbrace{\left\| \sum_{t=1}^T \frac{\tau_t z_t^*}{\sqrt{\tau_t^2 + (z_t^*)^2}} \frac{\boldsymbol{\phi}_t}{\sigma_t} \right\|_{\boldsymbol{H}_T^{-1}}}_{:=\boldsymbol{d}_T}. \tag{C.16}$$

**For the residual term $\|\lambda\boldsymbol{\theta}^*\|_{\boldsymbol{H}_T^{-1}}$**   Notice that $\boldsymbol{H}_T \succeq \lambda\boldsymbol{I}$ and thus $\boldsymbol{H}_T^{-1} \preceq \lambda^{-1}\boldsymbol{I}_d$. Therefore, $\|\lambda\boldsymbol{\theta}^*\|_{\boldsymbol{H}_T^{-1}} \leq \sqrt{\lambda}B$.

**For the self-normalized term** $\|d_T\|_{H_T^{-1}}$   The fact that $\boldsymbol{H}_T = \boldsymbol{H}_{T-1} + \frac{\phi_T \phi_T^\top}{\sigma_T^2}$ together with the Woodbury matrix identity implies that

$$\boldsymbol{H}_T^{-1} = \boldsymbol{H}_{T-1}^{-1} - \frac{\boldsymbol{H}_{T-1}^{-1}\phi_T\phi_T^\top \boldsymbol{H}_{T-1}^{-1}}{\sigma_T^2(1+w_T^2)} \quad \text{where} \quad w_T^2 := \frac{\phi_T^\top \boldsymbol{H}_{T-1}^{-1}\phi_T}{\sigma_T^2} = \left\|\frac{\phi_T}{\sigma_T}\right\|_{\boldsymbol{H}_{T-1}}^2. \tag{C.17}$$

Clearly, $w_T$ is $\mathcal{F}_{T-1}$-measurable and thus is predictable. By definition of $d_T$ and equation C.17,

$$\|d_T\|_{\boldsymbol{H}_T^{-1}}^2 = \left(d_{T-1} + \frac{\tau_T z_T^*}{\sqrt{\tau_T^2 + (z_T^*)^2}}\frac{\phi_T}{\sigma_T}\right)^\top \boldsymbol{H}_T^{-1}\left(d_{T-1} + \frac{\tau_T z_T^*}{\sqrt{\tau_T^2 + (z_T^*)^2}}\frac{\phi_T}{\sigma_T}\right)$$

$$= \|d_{T-1}\|_{\boldsymbol{H}_{T-1}^{-1}}^2 - \frac{1}{1+w_T^2}\left(\frac{d_{T-1}^\top \boldsymbol{H}_{T-1}^{-1}\phi_T}{\sigma_T}\right)^2$$

$$+ \frac{2\tau_T z_T^*}{\sqrt{\tau_T^2 + (z_T^*)^2}}\frac{d_{T-1}^\top \boldsymbol{H}_T^{-1}\phi_T}{\sigma_T} + \frac{\tau_T^2 (z_T^*)^2}{\tau_T^2 + (z_T^*)^2}\frac{\phi_T^\top \boldsymbol{H}_T^{-1}\phi_T}{\sigma_T^2}$$

$$\leqslant \|d_{T-1}\|_{\boldsymbol{H}_{T-1}^{-1}}^2 + \underbrace{\frac{2\tau_T z_T^*}{\sqrt{\tau_T^2 + (z_T^*)^2}}\frac{d_{T-1}^\top \boldsymbol{H}_T^{-1}\phi_T}{\sigma_T}}_{I_1} + \underbrace{\frac{\tau_T^2 (z_T^*)^2}{\tau_T^2 + (z_T^*)^2}\frac{\phi_T^\top \boldsymbol{H}_T^{-1}\phi_T}{\sigma_T^2}}_{I_2}. \tag{C.18}$$

For $I_1$, by equation C.17, we have

$$I_1 = \frac{2\tau_T z_T^*}{\sqrt{\tau_T^2 + (z_T^*)^2}}\frac{1}{\sigma_T}d_{T-1}^\top\left(\boldsymbol{H}_{T-1}^{-1} - \frac{\boldsymbol{H}_{T-1}^{-1}\phi_T\phi_T^\top \boldsymbol{H}_{T-1}^{-1}}{\sigma_T^2(1+w_T^2)}\right)\phi_T$$

$$= \frac{2\tau_T z_T^*}{\sqrt{\tau_T^2 + (z_T^*)^2}}\frac{1}{1+w_T^2}\frac{d_{T-1}^\top \boldsymbol{H}_{T-1}^{-1}\phi_T}{\sigma_T}.$$

For $I_2$, we have

$$I_2 = \frac{\tau_T^2 (z_T^*)^2}{\tau_T^2 + (z_T^*)^2}\frac{\phi_T^\top \boldsymbol{H}_T^{-1}\phi_T}{\sigma_T^2}$$

$$= \frac{\tau_T^2 (z_T^*)^2}{\tau_T^2 + (z_T^*)^2}\frac{1}{\sigma_T^2}\phi_T^\top\left(\boldsymbol{H}_{T-1}^{-1} - \frac{\boldsymbol{H}_{T-1}^{-1}\phi_T\phi_T^\top \boldsymbol{H}_{T-1}^{-1}}{\sigma_T^2(1+w_T^2)}\right)\phi_T$$

$$= \frac{\tau_T^2 (z_T^*)^2}{\tau_T^2 + (z_T^*)^2}\left(w_T^2 - \frac{w_T^4}{1+w_T^2}\right)$$

$$= \frac{\tau_T^2 (z_T^*)^2}{\tau_T^2 + (z_T^*)^2}\frac{w_T^2}{1+w_T^2}.$$

Using the equations for $I_1, I_2$ and iterating equation C.18, we have

$$\|d_T\|_{\boldsymbol{H}_T^{-1}}^2 \leqslant \sum_{t=1}^T \frac{\tau_t z_t^*}{\sqrt{\tau_t^2 + (z_t^*)^2}}\frac{2}{1+w_t^2}\frac{d_{t-1}^\top \boldsymbol{H}_{t-1}^{-1}\phi_t}{\sigma_t} + \sum_{t=1}^T \frac{\tau_t^2 (z_t^*)^2}{\tau_t^2 + (z_t^*)^2}\frac{w_t^2}{1+w_t^2}. \tag{C.19}$$

Recall that

$$\kappa = d\log\left(1 + \frac{TL^2}{d\lambda\sigma_{\min}^2}\right).$$

**Lemma C.1.** Assume $\mathbb{E}[(z_t^*)^2|\mathcal{F}_{t-1}] \leqslant b^2$ for all $t \geqslant 1$. Let $A_t$ denotes the event where $\|d_n\|_{\boldsymbol{H}_n^{-1}} \leqslant \alpha_n$ for all $n \in [t]$. With probability at least $1 - \delta$, we have for all $T \geqslant 1$,

$$\sum_{t=1}^T \frac{2\tau_t z_t^* \mathbf{1}_{A_{t-1}}}{(\tau_t^2 + (z_t^*)^2)^{1/2}}\frac{1}{1+w_t^2}\frac{d_{t-1}^\top \boldsymbol{H}_{t-1}^{-1}\phi_t}{\sigma_t} \leqslant 4\max_{t\in[T]}\alpha_t \cdot \left[\frac{\kappa b^2}{4\tau_0} + b\sqrt{\kappa\log\frac{2T^2}{\delta}} + \frac{2\tau_0}{3}\log\frac{2T^2}{\delta}\right].$$

**Lemma C.2.** Assume $\mathbb{E}[(z_t^*)^2|\mathcal{F}_{t-1}] \leqslant b^2$ for all $t \geqslant 1$. For a fixed $\tau \geqslant 0$, with probability at least $1 - \delta$, the follow inequality uniformly holds for all $T \geqslant 1$,

$$\sum_{t=1}^{T} \frac{\tau_t^2(z_t^*)^2}{\tau_t^2 + (z_t^*)^2} \frac{w_t^2}{1 + w_t^2} \leqslant \left[ \sqrt{2\kappa}b + \tau_0 \sqrt{\log \frac{2T^2}{\delta}} \right]^2.$$

For any $T \geqslant 1$, we define

$$\alpha_T = 8 \left[ \frac{\kappa b^2}{\tau_0} + b\sqrt{\kappa \log \frac{2T^2}{\delta}} + \tau_0 \log \frac{2T^2}{\delta} \right]. \tag{C.20}$$

As a result of Lemma C.1 and Lemma C.2, with probability at least $1 - 2\delta$, for all $T \geqslant 0$,[5]

$$\sum_{t=1}^{T} \frac{2\tau_t z_t^* \mathbf{1}_{A_{t-1}}}{\sqrt{\tau_t^2 + (z_t^*)^2}} \frac{1}{1 + w_t^2} \frac{\boldsymbol{d}_{t-1}^\top \boldsymbol{H}_{t-1}^{-1} \boldsymbol{\phi}_t}{\sigma_t} \leqslant \frac{\alpha_T^2}{2} \quad \text{and} \quad \sum_{t=1}^{T} \frac{\tau_t^2(z_t^*)^2}{\tau_t^2 + (z_t^*)^2} \frac{w_t^2}{1 + w_t^2} \leqslant \frac{\alpha_T^2}{2}. \tag{C.21}$$

Let $B$ denote the event that the conditions in equation C.21 hold for $T \geqslant 0$. By Lemma C.1 and Lemma C.2, we know that $\mathbb{P}(B) \geqslant 1 - 2\delta$. We now introduce a new event $C$ that is defined by

$$C := \left\{ \|\boldsymbol{d}_T\|_{\boldsymbol{H}_T^{-1}} \leqslant \alpha_T, \text{ for all } T \geqslant 0 \right\} = \bigcap_{t=0}^{\infty} A_t.$$

In the following, we will show that $B \subseteq C$ by mathematical induction. As a result, it follows that

$$\mathbb{P}(C) \geqslant \mathbb{P}(B) \geqslant 1 - \delta.$$

Finally, we use mathematical induction to show that if $B$ is true, then $C$ must be true, i.e., all $A_t$ is true for all $t \geqslant 0$ on the condition that the last inequalities equation C.21 are valid for all $T \geqslant 0$. When $t = 0$, $A_0$ is true by definition. Suppose that at iteration $T - 1$, for all $0 \leqslant t \leqslant T - 1$, the event $A_t$ is true, then we are going to show that $A_T$ is also true. By comparing the definition of $A_T$ and $A_{T-1}$, we only need to show that $\|d_T\|_{H_T^{-1}} \leqslant \alpha_T$ which is equivalent to $\|d_T\|_{H_T^{-1}}^2 \leqslant \alpha_T^2$. It follows due to the following inequality

$$\|\boldsymbol{d}_T\|_{\boldsymbol{H}_T^{-1}}^2 \overset{(C.19)}{\leqslant} \sum_{t=1}^{T} \frac{\tau_t z_t^*}{\sqrt{\tau_t^2 + (z_t^*)^2}} \frac{2}{1 + w_t^2} \frac{\boldsymbol{d}_{t-1}^\top \boldsymbol{H}_{t-1}^{-1} \boldsymbol{\phi}_t}{\sigma_t} + \sum_{t=1}^{T} \frac{\tau_t^2(z_t^*)^2}{\tau_t^2 + (z_t^*)^2} \frac{w_t^2}{1 + w_t^2}$$

$$\overset{(a)}{=} \sum_{t=1}^{T} \frac{\tau_t z_t^* \mathbf{1}_{A_{t-1}}}{\sqrt{\tau_t^2 + (z_t^*)^2}} \frac{2}{1 + w_t^2} \frac{\boldsymbol{d}_{t-1}^\top \boldsymbol{H}_{t-1}^{-1} \boldsymbol{\phi}_t}{\sigma_t} + \sum_{t=1}^{T} \frac{\tau_t^2(z_t^*)^2}{\tau_t^2 + (z_t^*)^2} \frac{w_t^2}{1 + w_t^2}$$

$$\overset{(b)}{\leqslant} \frac{\alpha_T^2}{2} + \frac{\alpha_T^2}{2} = \alpha_T^2,$$

where $(a)$ uses the condition that all $A_t$ is true for all $0 \leqslant t \leqslant T - 1$ and $(b)$ uses the conditions equation C.21. As a result, we can conclude that all $\{A_t\}_{t \geqslant 0}$ is true and thus $\|\boldsymbol{d}_T\|_{\boldsymbol{H}_T^{-1}} \leqslant \alpha_T$ for all $T \geqslant 1$.

## C.4 Proof of Lemma C.1

*Proof of Lemma C.1.* We will make use of the Freedman inequality Lemma G.1 to prove our result. Recall that $\tau_t = \tau_0 \frac{\sqrt{1 + w_t^2}}{w_t}$. Set $Y_t = \frac{\tau_t z_t^*}{\sqrt{\tau_t^2 + (z_t^*)^2}} \frac{2}{1 + w_t^2} \frac{\boldsymbol{d}_{t-1}^\top \boldsymbol{H}_{t-1}^{-1} \boldsymbol{\phi}_t \mathbf{1}_{A_{t-1}}}{\sigma_t}$ with the event $A_{t-1}$ defined in the lemma. For simplicity, we denote $X_t = Y_t - \mathbb{E}[Y_t|\mathcal{F}_{t-1}]$. Notice that

$$\left| \frac{\boldsymbol{d}_{t-1}^\top \boldsymbol{H}_{t-1}^{-1} \boldsymbol{\phi}_t}{\sigma_t} \cdot \mathbf{1}_{A_{t-1}} \right| \leqslant \|\boldsymbol{d}_{t-1} \mathbf{1}_{A_{t-1}}\|_{\boldsymbol{H}_{t-1}^{-1}} \cdot \left\| \frac{\boldsymbol{\phi}_t}{\sigma_t} \right\|_{\boldsymbol{H}_{t-1}^{-1}} \leqslant \alpha_{t-1} w_t.$$

---

[5]Note that it's easy to verify that the following inequalities are true when $t = 0$.

As a result, we have

$$|Y_t| \leqslant \tau_t \alpha_{t-1} \cdot \frac{2w_t}{1 + w_t^2} \leqslant 2\tau_0 \alpha_{t-1} \quad \text{and thus} \quad |X_t| \leqslant |Y_t| + |\mathbb{E}[Y_t|\mathcal{F}_{t-1}]| \leqslant 4\tau_0 \alpha_{t-1}.$$

We also find that

$$\mathbb{E}[X_t^2|\mathcal{F}_{t-1}] \overset{(a)}{\leqslant} \mathbb{E}[Y_t^2|\mathcal{F}_{t-1}] = \mathbb{E}\left[\left(\frac{2w_t}{1 + w_t^2}\right)^2 \|d_{t-1}\|_{H_{t-1}^{-1}}^2 \mathbf{1}_{A_{t-1}} \frac{\tau_t^2(z_t^*)^2}{\tau_t^2 + (z_t^*)^2}\bigg|\mathcal{F}_{t-1}\right]$$

$$\overset{(b)}{\leqslant} \left(\frac{2w_t}{1 + w_t^2}\right)^2 \alpha_{t-1}^2 b^2 \leqslant \min\{1, 2w_t\}^2 \alpha_{t-1}^2 b^2 \leqslant 4\min\{1, w_t^2\}\alpha_{t-1}^2 b^2$$

where $(a)$ uses $\mathbb{E}(X - \mathbb{E}X)^2 \leqslant \mathbb{E}X^2$ for any random variable $X$ and $(b)$ uses $\mathbb{E}[\varepsilon_t^2|\mathcal{F}_{t-1}] \leqslant b^2\sigma_t^2$ due to $\mathbb{E}[(z_t^*)^2|\mathcal{F}_{t-1}] \leqslant b^2$.

Notice that $\|\phi_t\|/\sigma_t \leqslant \|\phi_t\|/\sigma_{\min} \leqslant L/\sigma_{\min}$. Then by Lemma G.5, we have

$$\sum_{t=1}^{T} \min\{1, w_t^2\} \leqslant 2d \log\left(1 + \frac{TL^2}{d\lambda\sigma_{\min}^2}\right) := 2\kappa. \tag{C.22}$$

Hence, by equation C.22,

$$\sum_{t=1}^{T} \mathbb{E}[X_t^2|\mathcal{F}_{t-1}] \leqslant 4\sum_{t=1}^{T} \min\{1, w_t^2\}\alpha_{t-1}^2 b^2 \leqslant 4\max_{t\in[T]} \alpha_t^2 \cdot \sum_{t=1}^{T} \min\{1, w_t^2\}b^2$$

$$\leqslant \max_{t\in[T]} \alpha_t^2 \cdot 8db^2 \log\left(1 + \frac{TL^2}{d\lambda\sigma_{\min}^2}\right) \leqslant 8\kappa b^2 \cdot \max_{t\in[T]} \alpha_t^2.$$

On the other hand, using $\mathbb{E}[z_t^*|\mathcal{F}_{t-1}] = 0$ we have

$$\left|\mathbb{E}\left[\frac{\tau_t z_t^*}{\sqrt{\tau_t^2 + (z_t^*)^2}}\bigg|\mathcal{F}_{t-1}\right]\right| = \left|\mathbb{E}\left[\left(\frac{\tau_t}{\sqrt{\tau_t^2 + (z_t^*)^2}} - 1\right)z_t^*\bigg|\mathcal{F}_{t-1}\right]\right| \leqslant \mathbb{E}\left[\frac{(z_t^*)^2}{2\tau_t}\bigg|\mathcal{F}_{t-1}\right] \leqslant \frac{b^2}{2\tau_t}$$

which implies

$$\left|\sum_{t=1}^{T} \mathbb{E}[Y_t|\mathcal{F}_{t-1}]\right| \leqslant \sum_{t=1}^{T} \frac{b^2}{2\tau_t} \frac{w_t}{1 + w_t^2} \alpha_{t-1} \leqslant \frac{b^2}{2\tau_0} \sum_{t=1}^{T} \frac{w_t^2}{1 + w_t^2} \alpha_{t-1}$$

$$\leqslant \sup_{t\in[T]} \alpha_t \cdot \frac{b^2}{2\tau} \sum_{t=1}^{T} \min\{1, w_t^2\} \leqslant \sup_{t\in[T]} \alpha_t \cdot \frac{\kappa b^2}{\tau_0}.$$

By Freedman inequality in Lemma G.1, it follows that for a given $T$ and $\tau_0$, with probability $1 - \frac{\delta}{2T^2}$,

$$\sum_{t=1}^{T} Y_t \leqslant \left|\sum_{t=1}^{T} \mathbb{E}[Y_t|\mathcal{F}_{t-1}]\right| + 4\max_{t\in[T]} \alpha_t \cdot \left[b\sqrt{\kappa \log \frac{2T^2}{\delta}} + \frac{2\tau_0}{3} \log \frac{2T^2}{\delta}\right]$$

$$\leqslant 4\max_{t\in[T]} \alpha_t \cdot \left[\frac{\kappa b^2}{4\tau_0} + b\sqrt{\kappa \log \frac{2T^2}{\delta}} + \frac{2\tau_0}{3} \log \frac{2T^2}{\delta}\right].$$

Finally, taking a union bound for the last inequality from $T = 1$ to $\infty$ and using the fact that $\sum_{t=1}^{\infty} t^{-2} < 2$ completes the proof. $\qquad\square$

## C.5 Proof of Lemma C.2

*Proof of Lemma C.2.* Set $Y_t = \frac{\tau_t^2 (z_t^*)^2}{\tau_t^2 + (z_t^*)^2} \frac{w_t^2}{1 + w_t^2}$ and $X_t = Y_t - \mathbb{E}[Y_t | \mathcal{F}_{t-1}]$. Recall that $\tau_t = \tau_0 \frac{\sqrt{1 + w_t^2}}{w_t}$. Clearly, we have $|Y_t| \leqslant \tau_t^2 \frac{w_t^2}{1 + w_t^2} \leqslant \tau_0^2$ and thus $|X_t| = |Y_t - \mathbb{E}[Y_t | \mathcal{F}_{t-1}]| \leqslant \max\{|Y_t|, |\mathbb{E}[Y_t | \mathcal{F}_{t-1}]|\} \leqslant \tau_0^2$. We also find that

$$\mathbb{E}[X_t^2 | \mathcal{F}_{t-1}] \overset{(a)}{\leqslant} \mathbb{E}[Y_t^2 | \mathcal{F}_{t-1}] \leqslant \left( \frac{w_t^2}{1 + w_t^2} \right)^2 \mathbb{E} \left[ \left( \frac{\tau_t^2 (z_t^*)^2}{\tau_t^2 + (z_t^*)^2} \right)^2 \Big| \mathcal{F}_{t-1} \right]$$

$$\leqslant \tau_t^2 \left( \frac{w_t^2}{1 + w_t^2} \right)^2 \mathbb{E} \left[ (z_t^*)^2 | \mathcal{F}_{t-1} \right] \overset{(b)}{\leqslant} \tau_0^2 b^2 \frac{w_t^2}{1 + w_t^2}$$

where $(a)$ uses $\mathbb{E}(X - \mathbb{E}X)^2 \leqslant \mathbb{E}X^2$ for any random variable $X$ and $(b)$ uses $\mathbb{E}[(z_t^*)^2 | \mathcal{F}_{t-1}] \leqslant b^2$ due to $\mathbb{E}[\varepsilon_t^2 | \mathcal{F}_{t-1}] \leqslant b^2 \nu_t^2$. Hence, by equation C.22, we have

$$\sum_{t=1}^{T} \mathbb{E}[X_t^2 | \mathcal{F}_{t-1}] \leqslant \tau_0^2 b^2 \sum_{t=1}^{T} \frac{w_t^2}{1 + w_t^2} \leqslant \tau_0^2 b^2 \sum_{t=1}^{T} \min\{1, w_t^2\} \leqslant 2\kappa \tau_0^2 b^2.$$

On the other hand,

$$\sum_{t=1}^{T} \mathbb{E}[Y_t | \mathcal{F}_{t-1}] = \sum_{t=1}^{T} \frac{w_t^2}{1 + w_t^2} \mathbb{E} \left[ \frac{\tau_t^2 (z_t^*)^2}{\tau_t^2 + (z_t^*)^2} \Big| \mathcal{F}_{t-1} \right]$$

$$\leqslant \sum_{t=1}^{T} \frac{w_t^2}{1 + w_t^2} \mathbb{E} \left[ (z_t^*)^2 | \mathcal{F}_{t-1} \right] \leqslant \sum_{t=1}^{T} \min\{1, w_t^2\} b^2 \leqslant 2\kappa b^2.$$

By Lemma G.1, it follows that with probability $1 - \frac{\delta}{2T^2}$,

$$\sum_{t=1}^{T} Y_t \leqslant \sum_{t=1}^{T} \mathbb{E}[Y_t | \mathcal{F}_{t-1}] + 2\tau_0 b \sqrt{\kappa \log \frac{2T^2}{\delta}} + \frac{2\tau_0^2}{3} \log \frac{2T^2}{\delta}$$

for a given $T$ and $\tau_0$. Putting all pieces together, it follows that with probability $1 - \frac{\delta}{2T^2}$,

$$\sum_{t=1}^{T} Y_t \leqslant \left[ \sqrt{2\kappa} b + \tau_0 \sqrt{\log \frac{2T^2}{\delta}} \right]^2.$$

Finally, taking a union bound for the last inequality from $T = 1$ to $\infty$ and using the fact that $\sum_{t=1}^{\infty} t^{-2} < 2$ completes the proof.

$\square$

# D Proof of Theorem 3.1

**Measurability** Let $\mathcal{F}_{h,k}$ denote the $\sigma$-field generated by all random variables up to and including the $h$-th step and $k$-th episode. More specifically, let $I_{h,k} = \{(i,j) : i \in [H], j \in [k-1] \text{ or } i \in [h], j = k\}$ denote the set of index pairs up to and including the $h$-th step and $k$-th episode and then $\mathcal{F}_{h,k} = \sigma \left( \bigcup_{(i,j) \in I_{h,k}} \{s_{i,j}, a_{i,j}, r_{i,j}\} \right)$. We make a convention that $\mathcal{F}_{0,k} = \mathcal{F}_{H,k-1}$. From our algorithm, we know that (i) $Q_h^k, V_h^k, \pi_h^k \in \mathcal{F}_{H,k-1}$ for any $Q \in \{\overline{Q}, \widehat{Q}, \check{Q}, \underline{Q}\}$ and $V \in \{\overline{V}, \widehat{V}, \check{V}, \underline{V}\}$, and (ii)

$$\boldsymbol{\mu}_{h-1,k}, \boldsymbol{\theta}_{h,k}, \boldsymbol{\psi}_{h,k}, \sigma_{h,k}, U_{h,k}, J_{h,k}, E_{h,k}, \boldsymbol{\phi}_{h,k}, \widetilde{\boldsymbol{\phi}}_{h,k}, w_{h,k}, \widetilde{w}_{h,k}, \tau_{h,k}, \widetilde{\tau}_{h,k}, \boldsymbol{H}_{h,k}, \widetilde{\boldsymbol{H}}_{h,k} \in \mathcal{F}_{h,k}.$$

### D.1 High-Probability Events

Let $\kappa = d \log \left( 1 + \frac{K}{d\lambda\sigma_{\min}^2} \right)$. We first introduce the following high-probability events.

1. We define $\mathcal{B}_{R^2}$ as the event that the following inequalities hold for all $h \in [H]$ and $k \in [K] \bigcup \{0\}$,

$$\boldsymbol{\psi}_h^* \in \widetilde{\mathcal{R}}_{h,k} := \left\{ \|\boldsymbol{\psi}\| \leqslant W : \|\boldsymbol{\psi}_{h,k} - \boldsymbol{\psi}\|_{\widetilde{\boldsymbol{H}}_{h,k}^{-1}} \leqslant \beta_{R^2} \right\}$$

where

$$\beta_{R^2} = 128 \left( \frac{\sqrt{\kappa}\sigma_{R^2}}{\sigma_{\min}} + \sqrt{d} \right) \sqrt{\log \frac{2HK^2}{\delta}} + 5\sqrt{\lambda}W.$$

2. We define $\mathcal{B}_0$ as the event that the following inequalities hold for all $h \in [H]$ and $k \in [K]$,

$$\max \left\{ \left\| (\boldsymbol{\mu}_h^* - \boldsymbol{\mu}_{h,k-1}) \overline{\boldsymbol{V}}_{h+1}^k \right\|_{\boldsymbol{H}_{h,k-1}}, \left\| (\boldsymbol{\mu}_h^* - \boldsymbol{\mu}_{h,k-1}) \underline{\boldsymbol{V}}_{h+1}^k \right\|_{\boldsymbol{H}_{h,k-1}} \right\} \leqslant \beta_0,$$

$$\left\| (\boldsymbol{\mu}_h^* - \boldsymbol{\mu}_{h,k-1}) [\overline{\boldsymbol{V}}_{h+1}^k]^2 \right\|_{\boldsymbol{H}_{h,k-1}} \leqslant \mathcal{H}\beta_0,$$

where

$$\beta_0 = \frac{4\mathcal{H}}{\sigma_{\min}} \sqrt{d^3 H \iota_0^2 + \log \frac{2H}{\delta}} + 3\sqrt{d\lambda}\mathcal{H}$$

$$\iota_0 = \max \left\{ \log \left( 1 + \frac{8LK}{\lambda\mathcal{H}\sqrt{d}\sigma_{\min}^2} \right), \log \left( 1 + \frac{32B^2K^2}{\sqrt{d}\lambda^3\mathcal{H}^2\sigma_{\min}^4} \right), \log \left( 1 + \frac{K}{\lambda\sigma_{\min}^2} \right) \right\}. \tag{D.1}$$

Here we choose $B \geqslant 3(\beta_R + \beta_V)$ and $L = W + \mathcal{H}\sqrt{\frac{dK}{\lambda}}$.

3. We define $\mathcal{B}_R$ as the event that the following inequalities hold for all $h \in [H]$ and $k \in [K] \bigcup \{0\}$,

$$\boldsymbol{\theta}_h^* \in \mathcal{R}_{h,k} := \left\{ \|\boldsymbol{\theta}\| \leqslant W : \|\boldsymbol{\theta}_{h,k} - \boldsymbol{\theta}_h^*\|_{\boldsymbol{H}_{h,k}} \leqslant \beta_R \right\},$$

$$\left| [\widehat{\mathbb{V}}_h \widehat{R}_h - \mathbb{V}_h R_h](s_{h,k}, a_{h,k}) \right| \leqslant R_{h,k} := \beta_{R^2} \|\widetilde{\phi}_{h,k}\|_{\widetilde{\boldsymbol{H}}_{h,k-1}^{k-1}} + 2\mathcal{H}\beta_R \|\phi_{h,k}\|_{\boldsymbol{H}_{h,k-1}^{-1}}, \tag{A.8}$$

where

$$\beta_R = 128(\sqrt{\kappa} + \sqrt{d}) \sqrt{\log \frac{2HK^2}{\delta}} + 5\sqrt{\lambda}W.$$

4. We define $\mathcal{B}_h$ as the event such that for all episode $k \in [K]$, all stages $h \leqslant h' \leqslant H$,

$$\max \left\{ \left\| (\boldsymbol{\mu}_{h'}^* - \boldsymbol{\mu}_{h',k-1}) \overline{\boldsymbol{V}}_{h'+1}^k \right\|_{\boldsymbol{H}_{h',k-1}}, \left\| (\boldsymbol{\mu}_{h'}^* - \boldsymbol{\mu}_{h',k-1}) \underline{\boldsymbol{V}}_{h'+1}^k \right\|_{\boldsymbol{H}_{h',k-1}} \right\} \leqslant \beta_V, \tag{D.2}$$

where

$$\beta_V = \mathcal{O} \left( \sqrt{d}\iota_1^2 + \sqrt{d\lambda}\mathcal{H} \right)$$

$$\iota_1 = \max \left\{ \iota_0, \log \frac{4HK^2}{\delta}, \log \left( 1 + \frac{4L\sqrt{d^3H}}{\sigma_{\min}} \right), \log \left( 1 + \frac{8\sqrt{d^7}HB^2}{\lambda\sigma_{\min}^2} \right) \right\}. \tag{D.3}$$

For simplicity, we further define $\mathcal{B}_V := \mathcal{B}_1$.

Our ultimate goal is to show $\mathcal{B}_V$ holds with high probability, a target used in previous work (Hu et al., 2022; He et al., 2022). More specifically, we first obtain coarse confidence sets for all parameters in the sense that the confidence radius (that is $\beta_{R^2}$ and $\beta_0$) is loose. In our analysis, $\mathcal{B}_{R^2} \bigcap \mathcal{B}_0$ serves as the 'coarse' event where the concentration results hold with a larger confidence radius, and $\mathcal{B}_R \bigcap \mathcal{B}_V$ serves as a 'refined' event where the confidence radius (that is $\beta_R$ and $\beta_V$) is much is tighter. Our first result is that $\mathcal{B}_{R^2} \bigcap \mathcal{B}_0$ holds with high probability as shown in Lemma D.1 and D.2. Their proofs are collected in Appendix F.1 and F.2.

**Lemma D.1.** If we set

$$\widetilde{\tau}_0 = \max\left\{\frac{\sqrt{2\kappa}\sigma_{R^2}}{\sigma_{\min}}, 2\sqrt{d}\right\} \Big/ \sqrt{\log\frac{2HK^2}{\delta}}, \tag{D.4}$$

the event $\mathcal{B}_{R^2}$ holds with probability at least $1 - 4\delta$.

**Lemma D.2.** The event $\mathcal{B}_0$ holds with probability at least $1 - 3\delta$.

These coarse confidence sets are then used to estimate variance for the reward functions and value functions. A key step is to show the adapted variance $\sigma_{h,k}$'s are indeed upper bounds of these variances (that is $[\mathbb{V}_h R_h](s_{h,k}, a_{h,k}) + [\mathbb{V}_h V_{h+1}^*](s_{h,k}, a_{h,k}))$ for all $h \in [H]$. A frequently used argument is backward induction. That is given the estimation is optimistic at the stage $h + 1$, we then show the optimistic estimation is maintained at the stage $h$. Induction over the stage $h$ would complete the proof. The following lemma provides estimation error bounds for $[\mathbb{V}_h R_h](s_{h,k}, a_{h,k})$ and shows that the event $\mathcal{B}_R$ holds with high probability. Its proof is deferred in Appendix F.4.

**Lemma D.3.** If we set

$$\tau_0 = \max\{\sqrt{2\kappa}, 2\sqrt{d}\} \Big/ \sqrt{\log\frac{2HK^2}{\delta}}, \tag{D.5}$$

the event $\mathcal{B}_R$ holds with probability at least $1 - 8\delta$.

In Lemma D.4, we show that our constructed value functions $\overline{V}$ and $\underline{V}$ are indeed optimistic and pessimistic estimators of the true value functions under the event defined before. Its proof is deferred in Appendix F.5.

**Lemma D.4** (Optimism and pessimism). For any $h \in [H]$, if $\mathcal{B}_R \bigcap \mathcal{B}_h$ holds, for any $k \in [K] \bigcup\{0\}$,

$$\underline{V}_h^k(\cdot) \leqslant V_h^*(\cdot) \leqslant \overline{V}_h^k(\cdot).$$

With the established optimism and pessimism, we can establish upper bounds for the estimation errors of the three terms, namely $[\mathbb{V}_h V_{h+1}^*](s_{h,k}, a_{h,k})$, $\left[\mathbb{V}_h(\overline{V}_{h+1}^k - V_{h+1}^*)\right](s_{h,k}, a_{h,k})$, and $\left[\mathbb{V}_h(\underline{V}_{h+1}^k - V_{h+1}^*)\right](s_{h,k}, a_{h,k})$ in the following lemmas. Their proofs are deferred in Appendix F.6 and F.7.

**Lemma D.5.** On the event $\mathcal{B}_0 \bigcap \mathcal{B}_{h+1}$, it follows that for all $k \in [K]$

$$\left|\left[\mathbb{V}_h V_{h+1}^* - \widehat{\mathbb{V}}_h \overline{V}_{h+1}^k\right](s_{h,k}, a_{h,k})\right| \leqslant U_{h,k}$$

where

$$U_{h,k} = \min\left\{\mathcal{V}^2, 11\mathcal{H}\beta_0 \cdot \|\phi_{h,k}\|_{\boldsymbol{H}_{h,k-1}^{-1}} + 4\mathcal{H} \cdot \widehat{\mathbb{P}}_{h,k}(\overline{V}_{h+1}^k - \underline{V}_{h+1}^k)(s_{h,k}, a_{h,k})\right\} \tag{A.9}$$

with $\widehat{\mathbb{P}}_{h,k}(\cdot|s,a) = \boldsymbol{\mu}_{h,k-1}^\top \phi(s,a)$.

**Lemma D.6.** On the event $\mathcal{B}_0 \bigcap \mathcal{B}_R \bigcap \mathcal{B}_{h+1}$, it follows that for all $j \leqslant k \leqslant K$

$$\max\left\{\left[\mathbb{V}_h(\overline{V}_{h+1}^k - V_{h+1}^*)\right](s_{h,j}, a_{h,j}), \left[\mathbb{V}_h(\underline{V}_{h+1}^k - V_{h+1}^*)\right](s_{h,j}, a_{h,j})\right\} \leqslant E_{h,j}$$

where

$$E_{h,j} = \min\left\{\mathcal{H}^2, 2\mathcal{H}\beta_0\|\phi_{h,j}\|_{\boldsymbol{H}_{h,j-1}^{-1}} + \mathcal{H} \cdot \left[\widehat{\mathbb{P}}_{h,j}(\overline{V}_{h+1}^j - \underline{V}_{h+1}^j)\right](s_{h,j}, a_{h,j})\right\} \tag{A.7}$$

with $\widehat{\mathbb{P}}_{h,j}(\cdot|s,a) = \boldsymbol{\mu}_{h,j-1}^\top \phi(s,a)$.

With the last four lemmas, one can easily prove $\sigma_{h,k}$ indeed serves as an upper bound of the true variance of $V_{h+1}^*$ at stage $h$. Therefore, by the backward induction, we can prove the following lemma whose proof is in Appendix F.8.

**Lemma D.7.** On the event $\mathcal{B}_0 \bigcap \mathcal{B}_R$, the event $\mathcal{B}_V$ holds with probability at least $1 - 2\delta$.

## D.2 Regret Analysis

In the previous subsection, we know that with probability at least $1 - 17\delta$, the event $\mathcal{B}_V \bigcap \mathcal{B}_R$ holds. Based on Lemma D.4, the optimism implies that

$$\text{Reg}(K) := \sum_{k=1}^{K} (V_1^* - V_1^{\pi_k})(s_{1,k}) \leqslant \sum_{k=1}^{K} (\overline{V}_1^k - V_1^{\pi_k})(s_{1,k}).$$

We then relate the suboptimality gap $\sum_{k=1}^{k} (\overline{V}_1^k - V_1^{\pi_k})(s_{1,k})$ to the term $\sum_{k=1}^{K} \sum_{h=1}^{H} \|\phi_{h,k}\|_{\boldsymbol{H}_{h,k-1}^{-1}}$ in Lemma D.8. We emphasize that the bound in Lemma D.8 is much finer than previous bounds (e.g., Lemma B.1 in (He et al., 2022)) in the sense that the rest term is $\tilde{\mathcal{O}}(H\mathcal{H})$ instead of previous $\tilde{\mathcal{O}}(\sqrt{HK\mathcal{H}})$. This is because

- We first adopt a variance-aware Bernstein's inequality to relate $\sum_{k=1}^{k} (\overline{V}_1^k - V_1^{\pi_k})(s_{1,k})$ with a sum of martingale differences. In particular, we show that with high probability,

$$\sum_{k=1}^{K} (\overline{V}_1^k - V_1^{\pi_k})(s_{1,k}) \leqslant \sum_{k=1}^{K} \sum_{h=1}^{H} \left[ X_{h,k} + 4\beta \|\phi_{h,k}\|_{\boldsymbol{H}_{h,k-1}^{-1}} \right].$$

  where $\{X_{h,k}\}_{h \in [H]}$ is a martingale difference sequence define by

$$X_{h,k} := \mathbb{P}_h (\overline{V}_{h+1}^k - V_{h+1}^{\pi_k})(s_{h,k}, a_{h,k}) - (\overline{V}_{h+1}^k - V_{h+1}^{\pi_k})(s_{h+1,k}).$$

  The variance-aware Bernstein's inequality implies that with high probability,

$$\left| \sum_{k=1}^{K} \sum_{h=1}^{H} X_{h,k} \right| \leqslant \tilde{\mathcal{O}}(1) \cdot \sqrt{\mathcal{H} \cdot \sum_{k=1}^{K} \sum_{h=1}^{H} \mathbb{P}_h (\overline{V}_{h+1}^k - V_{h+1}^{\pi_k})(s_{h,k}, a_{h,k})} + \tilde{\mathcal{O}}(\mathcal{H}).$$

- We then use a recursion argument to simplify the variance term above. More specifically, we show that with high probability,

$$\sum_{k=1}^{K} \sum_{h=1}^{H} \mathbb{P}_h (\overline{V}_{h+1}^k - V_{h+1}^{\pi_k})(s_{h,k}, a_{h,k}) \leqslant \mathcal{O}(H) \cdot \sqrt{\mathcal{H} \cdot \sum_{k=1}^{K} \sum_{h=1}^{H} \mathbb{P}_h (\overline{V}_{h+1}^k - V_{h+1}^{\pi_k})(s_{h,k}, a_{h,k})}$$

$$+ \mathcal{O}(H\beta) \sum_{k=1}^{K} \sum_{h=2}^{H} \|\phi_{h,k}\|_{\boldsymbol{H}_{h,k-1}^{-1}} + \tilde{\mathcal{O}}(H\mathcal{H}).$$

  We decouple the self-dependence on $\sum_{k=1}^{K} \sum_{h=1}^{H} \mathbb{P}_h (\overline{V}_{h+1}^k - V_{h+1}^{\pi_k})(s_{h,k}, a_{h,k})$ using this inequality that $x \leqslant 2(a^2 + b^2)$ for any $x \leqslant |a|\sqrt{x} + b^2$.

- Combining the two steps, we then complete the proof of Lemma D.8. The detailed proof is deferred to Appendix F.9.

In contrast, previous work directly applies Azuma-Hoeffding inequality to analyze the concentration of $\sum_{k=1}^{k} (\overline{V}_1^k - V_1^{\pi_k})(s_{1,k})$ so that $\left| \sum_{k=1}^{K} \sum_{h=1}^{H} X_{h,k} \right| \leqslant \tilde{\mathcal{O}}(\sqrt{HK\mathcal{H}})$, which inevitably introduces the additional $\tilde{\mathcal{O}}(\sqrt{K})$ dependence.

**Lemma D.8** (Suboptimality gap). With probability at least $1 - \delta$, on the event $\mathcal{B}_R \bigcap \mathcal{B}_V$, it follows that

$$\sum_{k=1}^{K} (\overline{V}_1^k - V_1^{\pi_k})(s_{1,k}) \leqslant 6\beta \sum_{k=1}^{K} \sum_{h=1}^{H} \|\phi_{h,k}\|_{\boldsymbol{H}_{h,k-1}^{-1}} + 38H\mathcal{H} \log \frac{4\lceil \log_2 HK \rceil}{\delta} \text{ and}$$

$$\sum_{k=1}^{K} \sum_{h=1}^{H} \mathbb{P}_h (\overline{V}_{h+1}^k - V_{h+1}^{\pi_k})(s_{h,k}, a_{h,k}) \leqslant 8H\beta \sum_{k=1}^{K} \sum_{h=1}^{H} \|\phi_{h,k}\|_{\boldsymbol{H}_{h,k-1}^{-1}} + 38H^2\mathcal{H} \log \frac{4\lceil \log_2 HK \rceil}{\delta}.$$

Using a similar argument, we provide a finer bound for the gap between optimistic and pessimistic value functions $\sum_{k=1}^{K}\sum_{h=1}^{H}\mathbb{P}_h(\overline{V}_{h+1}^k - \underline{V}_{h+1}^k)(s_{h,k}, a_{h,k})$ in Lemma D.9. Its proof is provided in Appendix F.10.

**Lemma D.9** (Gap between optimistic and pessimistic value functions). With probability at least $1 - \delta$, on the event $\mathcal{B}_V \bigcap \mathcal{B}_R$, it follows that

$$\sum_{k=1}^{K}\sum_{h=1}^{H}\mathbb{P}_h(\overline{V}_{h+1}^k - \underline{V}_{h+1}^k)(s_{h,k}, a_{h,k}) \leqslant 12H\beta \sum_{k=1}^{K}\sum_{h=1}^{H}\|\phi_{h,k}\|_{\boldsymbol{H}_{h,k-1}^{-1}} + 38H^2\mathcal{H}\log\frac{4\lceil\log_2 HK\rceil}{\delta}.$$

The following issue is to upper bound the term $\sum_{k=1}^{K}\sum_{h=1}^{H}\|\phi_{h,k}\|_{\boldsymbol{H}_{h,k-1}^{-1}}$. Since the estimation of reward variance concerns the other term $\sum_{k=1}^{K}\sum_{h=1}^{H}\|\widetilde{\phi}_{h,k}\|_{\widetilde{\boldsymbol{H}}_{h,k-1}^{-1}}$, we are motivated to analyze them simultaneously via $\sum_{k=1}^{K}\sum_{h=1}^{H} b_{h,k}$ where $b_{h,k} = \max\left\{\|\phi_{h,k}\|_{\boldsymbol{H}_{h,k-1}^{-1}}, \|\widetilde{\phi}_{h,k}\|_{\widetilde{\boldsymbol{H}}_{h,k-1}^{-1}}\right\}$. Previous works (Hu et al., 2022; He et al., 2022) mainly use Cauchy–Schwarz inequality to analyze it and obtain

$$\sum_{k=1}^{K}\sum_{h=1}^{H} b_{h,k} \leqslant \sqrt{\left(\sum_{k=1}^{K}\sum_{h=1}^{H}\sigma_{h,k}^2\right)\left(\sum_{k=1}^{K}\sum_{h=1}^{H}\max\{w_{h,k}^2, \widetilde{w}_{h,k}^2\}\right)} = \widetilde{\mathcal{O}}\left(\sqrt{dH} \cdot \sqrt{\sum_{k=1}^{K}\sum_{h=1}^{H}\sigma_{h,k}^2}\right).$$

where the last equality uses the elliptical potential lemmas in Lemma G.5. A standard analysis of the law of total variation would imply $\sqrt{\sum_{k=1}^{K}\sum_{h=1}^{H}\sigma_{h,k}^2} = \widetilde{\mathcal{O}}(\sqrt{H^2 K})$. However, this result doesn't satisfy our target for two reasons. First, due to the use of adaptive Huber regression, our definition of $\sigma_{h,k}$ is more complicated than previous algorithms. We need a more elaborate analysis to handle the additional terms in the definition of $\sigma_{h,k}$'s. Second, the previous result considers the worst-case scenario, while our target is to provide a finer variance-aware regret. Therefore, it is imperative to provide a finer bound for the sum of bonuses $\sum_{k=1}^{K}\sum_{h=1}^{H} b_{h,k}$. We did it in Lemma D.10.

**Lemma D.10** (Sum of bonuses). Set $\lambda = \frac{1}{\mathcal{H}^2 + W^2}$. Let $\mathcal{A}_0$ denote the intersection event of Lemma D.8 and D.9. With probability at least $1 - 2\delta$, on the event $\mathcal{B}_R \bigcap \mathcal{B}_V \bigcap \mathcal{B}_0 \bigcap \mathcal{B}_{R^2} \bigcap \mathcal{A}_0$, we have

$$\sum_{k=1}^{K}\sum_{h=1}^{H} b_{h,k} = \widetilde{\mathcal{O}}\left(\sqrt{dHK\mathcal{G}^*} + Hd^{0.5}K^{0.5}\sigma_{\min} + \frac{H^{2.5}d^{5.5}\mathcal{H}^2 + Hd^{1.5}\sigma_{R^2}}{\sigma_{\min}}\right)$$
$$+ \widetilde{\mathcal{O}}\left(H^3 d^{4.5}\mathcal{H} + Hd^{0.5}\sigma_R + Hd^{1.5}\right).$$

where $\widetilde{\mathcal{O}}(\cdot)$ ignores constant factors and logarithmic dependence.

We emphasize that Lemma D.10 is perhaps the most technical lemma in our paper. To address the difficulty mentioned earlier, we divide the full index set $\mathcal{I} := [H] \times [K]$ into three disjoint subsets $\mathcal{I} = \bigcup_{i=1,2,3}\mathcal{J}_i$ according to which value $\sigma_{h,k}$ takes (given $\sigma_{h,k}$ is the maximum value among five quantities). For those indexes in $\mathcal{J}_1$ where the bonuses are small enough, we still use the Cauchy–Schwarz inequality to bound $\sum_{(h,k)\in\mathcal{J}_1} b_{h,k} \leqslant \widetilde{\mathcal{O}}\left(\sqrt{dH} \cdot \sqrt{\sum_{(h,k)\in\mathcal{I}}\sigma_{h,k}^2}\right)$. This sum-of-squared-bonus quantity involves $\sum_{(h,k)\in\mathcal{I}} E_{h,k}$ and $\sum_{(h,k)\in\mathcal{I}} J_{h,k}$ which we then pay additional efforts to analyze. For those indexes in $\mathcal{J}_2$ or $\mathcal{J}_3$ where the bonuses are relatively large, we directly analyze $\sum_{(h,k)\in\mathcal{J}_2\bigcup\mathcal{J}_3} b_{h,k}$. Thanks to the particular structure, $\sum_{(h,k)\in\mathcal{J}_2\bigcup\mathcal{J}_3} b_{h,k}$ contributes to the non-leading term in the final bound. Putting pieces together, we complete the proof. A formal proof can be found in Appendix F.11.

At the end of the subsection, we summarize the proof in a few lines.

$$\begin{aligned}
\mathrm{Reg}(K) = \sum_{k=1}^{K}(V_1^* - V_1^{\pi_k})(s_{1,k}) &\overset{(a)}{\leqslant} \sum_{k=1}^{K}(\overline{V}_1^k - V_1^{\pi_k})(s_{1,k}) \\
&\overset{(b)}{\leqslant} 3\beta\sum_{k=1}^{K}\sum_{h=1}^{H}\|\phi_{h,k}\|_{\boldsymbol{H}_{h,k-1}^{-1}} + 38H\mathcal{H}\log\frac{4\lceil\log_2 HK\rceil}{\delta}
\end{aligned}$$

$$\overset{(c)}{\leqslant} 3\beta \sum_{k=1}^{K} \sum_{h=1}^{H} b_{h,k} + 38H\mathcal{H} \log \frac{4\lceil \log_2 HK \rceil}{\delta}$$

$$\overset{(d)}{=} \widetilde{\mathcal{O}} \left( d\sqrt{HK\mathcal{G}^*} + HdK^{0.5}\sigma_{\min} + \frac{H^{2.5}d^6\mathcal{H}^2 + Hd^2\sigma_{R^2}}{\sigma_{\min}} + H^3 d^5 \mathcal{H} + Hd\sigma_R + Hd^2 \right)$$

where $(a)$ follows from the optimism result in Lemma D.4, $(b)$ follows from the suboptimality gap result in Lemma D.8, $(c)$ uses $b_{h,k} = \max\left\{ \|\phi_{h,k}\|_{\boldsymbol{H}_{h,k-1}^{-1}}, \|\widetilde{\phi}_{h,k}\|_{\widetilde{\boldsymbol{H}}_{h,k-1}^{-1}} \right\}$, and $(d)$ follows from sum-of-bonus result in Lemma D.10 and $\beta = \beta_R + \beta_V = \widetilde{\mathcal{O}}(\sqrt{d})$.

# E  Proof of Theorem 3.2

*Proof of Theorem 3.2.* We consider the two complexities respectively.

**Space Complexity**  First, to perform AdaOFUL, VARA needs to store all seen rewards and feature vectors (i.e., $\phi_{h,k}, \widetilde{\phi}_{h,k}$), which is required by all RL/bandit algorithms robust to heavy-tailed rewards (Shao et al., 2018; Xue et al., 2021; Zhuang & Sui, 2021). AdaOFUL also keeps all robustification parameters $\tau_{h,k}, \widetilde{\tau}_{h,k}$. It then incurs $\mathcal{O}(HKd)$ space storage in total.

Second, due to the rare-switching technique, one can show that $\overline{Q}_h^k$ (or $\underline{Q}_h^k$) is the minimum (or maximum) of at most $\widetilde{\mathcal{O}}(dH)$ temporary optimistic (or pessimistic) functions (see Lemma G.7). It means that we need to store at most $\widetilde{\mathcal{O}}(dH)$ different versions of $\boldsymbol{\theta}_{h,k-1}, \boldsymbol{\mu}_{h,k-1}\boldsymbol{V}_{h+1}^k, \boldsymbol{H}_{h,k-1}$'s. This incurs $\mathcal{O}(d^3 H^2)$ space cost.

Last, for all $(h,k) \in [H] \times [K]$, we need to trace $\{\phi(s_{h,k}, a)\}_{a \in \mathcal{A}}$ to evaluate each $\boldsymbol{\mu}_{h,k}\boldsymbol{V} = \boldsymbol{H}_{h,k}^{-1} \sum_{j=1}^{k} \sigma_{h,j}^{-2} \phi_{h,j} V(s_{h+1,k})$ for $\boldsymbol{V} \in \{\overline{\boldsymbol{V}}_{h+1}^k, [\overline{\boldsymbol{V}}_{h+1}^k]^2, \underline{\boldsymbol{V}}_{h+1}^k\}$, which takes $\mathcal{O}(d|\mathcal{A}|HK)$ space.

To sum up, VARA takes $\mathcal{O}(d^3 H^2 + d|\mathcal{A}|HK)$ space.

**Computational Complexity**  First, we use the Nesterov accelerated method to compute each $\boldsymbol{\theta}_{h,k}$. Since the loss function in equation A.1 is $\lambda$-strongly convex and $\left(\lambda + \frac{K}{\sigma_{\min}^2}\right)$-smooth, the computational cost for each $\boldsymbol{\theta}_{h,k}$ is $\widetilde{\mathcal{O}}\left(d\sqrt{1 + \frac{K}{\lambda(\sigma_{\min}^*)^2}}\right) = \widetilde{\mathcal{O}}(\max\{d, H^{-3/4}d^{-3/2}K^{3/4}\})$ and the total cost is $\widetilde{\mathcal{O}}(HK(d + H^{-3/4}d^{-3/2}K^{3/4}))$. We emphasize that we don't need to compute $\boldsymbol{\theta}_{h,k}$ exactly. It suffices to terminate at a solution $\widehat{\boldsymbol{\theta}}_{h,k}$ once its accuracy satisfies $\|\widehat{\boldsymbol{\theta}}_{h,k} - \boldsymbol{\theta}_{h,k}\|_{\boldsymbol{H}_{h,k}} \leqslant \sqrt{d}$. The iteration complexity is proportional to the root of the conditional number, i.e., $\widetilde{\mathcal{O}}(\max\{1, d^{-7/4}K^{3/4}\})$. Since each iteration takes $\mathcal{O}(d)$ operation, the computation complexity is $\widetilde{\mathcal{O}}(\max\{d, d^{-3/4}K^{3/4}\})$.

Second, each time when updating the value function, we take the minimum over at most $\widetilde{\mathcal{O}}(dH)$ quadratic functions. Moreover, the Sherman-Morrison formula computes $\boldsymbol{H}_{h,k}^{-1}$ and its products with any vectors, which takes $\mathcal{O}(d^2)$ operations. As a result, it needs $\widetilde{\mathcal{O}}(d^3 H)$ to evaluate the updated $Q_{h,k}(s, a)$ for a given pair $(s, a)$. Hence, computing $Q_{h,k}(s_{h,k}, \cdot)$, choosing $a_{h,k} = \arg\max_{a \in \mathcal{A}} Q_{h,k}(s_{h,k}, a)$, and estimating the variance $\sigma_{h,k}$ lead to $\widetilde{\mathcal{O}}(d^3 H^2 |\mathcal{A}|)$ computational complexity for each episode.

Last, note $\boldsymbol{\mu}_{h,k}\boldsymbol{V} = \boldsymbol{H}_{h,k}^{-1} \sum_{j=1}^{k} \sigma_{h,j}^{-2} \phi_{h,j} V(s_{h+1,k})$ for any value function $V(\cdot)$. If $V$ remains unchanged, we only need to compute the new term $\sigma_{h,k}^{-2} \phi_{h,k} V(s_{h+1,k})$, which has an $\widetilde{\mathcal{O}}(d^3 H|\mathcal{A}|)$ complexity each time. If $V$ changes to $V'$, we need to recalculate $\boldsymbol{\mu}_{h,k}\boldsymbol{V}'$, which has an $\widetilde{\mathcal{O}}(d^3 H|\mathcal{A}|K)$ complexity each time. Combining the computational complexity for all horizons and noticing that the number of episodes that trigger the updating criterion is at most $\widetilde{\mathcal{O}}(dH)$, VARA has a running time of $\widetilde{\mathcal{O}}(d^4|\mathcal{A}|H^3 K + HK(d + H^{-3/4}d^{-3/2}K^{3/4}))$. In terms of the dependence on $K$, it is slightly worse than LSVI-UCB++'s $\widetilde{\mathcal{O}}(d^4|\mathcal{A}|H^3 K)$ since the adaptive Huber regression doesn't have a closed-form solution, but is better than LSVI-UCB's $\widetilde{\mathcal{O}}(d^2|\mathcal{A}|HK^2)$ due to the rare-switching mechanism. $\square$

# F    Omitted lemmas in Section D

## F.1    Proof of Lemma D.1

*Proof of Lemma D.1.* The proof idea of Lemma D.1 is similar to that of Theorem 2.1 except for the following changes. First, $\widetilde{\phi}_{h,k} = \widetilde{\phi}(s_{h,k}, a_{h,k}) \in \mathbb{R}^d$ is instead the feature vector. Second, in the particular setting, we should respectively replace $L, B, T, \delta$ therein with $1, W, K, \delta/H$ defined here and redefine $c_0, c_1$ as $c_0 = \frac{1}{6\sqrt{3\log\frac{2HK^2}{\delta}}}, c_1 = \frac{1}{42 \cdot \frac{2HK^2}{\delta}}$ respectively. Third, by the choice of $\sigma_{h,k}$, we have $\sigma_{h,k}^2 \geqslant \left(\frac{W}{\sqrt{c_1 d}} + \mathcal{H}d^{2.5}H\right)b_{h,k} \geqslant \frac{W}{\sqrt{c_1 d}}\|\widetilde{\phi}_{h,k}\|_{\widetilde{H}_{h,k-1}^{-1}}$, which implies that $\frac{W^2\widetilde{w}_{h,k}^2}{\sigma_{h,k}^2} \leqslant c_1 d$. Similarly, due to $\sigma_{h,k}^2 \geqslant c_0^{-2}\|\widetilde{\phi}_{h,k}\|_{\widetilde{H}_{h-1,k}^{-1}}^2$, we have $\widetilde{w}_{h,k}^2 \leqslant c_0^2$. Last, for simplicity, we define $\varepsilon_{h,k} = \frac{r_{h,k}^2 - \langle\widetilde{\phi}_{h,k}, \psi_h^*\rangle}{\sigma_{h,k}}$ and $\mathcal{G}_{h,k} = \sigma(\mathcal{F}_{h-1,k}\bigcup\{s_{h,k}, a_{h,k}\})$. Then, we have $\varepsilon_{h,k} \in \mathcal{F}_{h,k}$, $\mathbb{E}[\varepsilon_{h,k}|\mathcal{G}_{h,k}] = 0$ and $\mathrm{Var}[\varepsilon_{h,k}|\mathcal{G}_{h,k}] \leqslant \left(\frac{\sigma_{R^2}}{\sigma_{\min}}\right)^2 := b^2$. Theorem 2.1 concerns the case where $b = 1$, however, its proof considers the general case where $b$ can be arbitrary. As a result, by a similar argument in Appendix C (which is doable due to the four conditions mentioned above), once setting $\widetilde{\tau}_0\sqrt{\log\frac{2HK^2}{\delta}} = \max\left\{\sqrt{2\kappa}b, 2\sqrt{d}\right\}$, with probability at least $1 - 3\delta$, we have for all $h \in [H]$ and $k \in [K]$, $\|\psi_{h,k} - \psi_h^*\|_{\widetilde{H}_{h,k}} \leqslant \beta_{R^2}$, that is the event $\mathcal{B}_{R^2}$ holds. $\qquad\square$

## F.2    Proof of Lemma D.2

We will make use of the following general results frequently. The proof is quite standard (Jin et al., 2020b; Wagenmaker et al., 2022a; Hu et al., 2022). We provide proof in Appendix F.3 for completeness.

**Lemma F.1.** Fix any $h \in [H]$. Consider a specific value function $f(\cdot)$ which satisfies

(i)  $\sup_{s \in \mathcal{S}} |f(s)| \leqslant C_0$;

(ii)  $f \in \mathcal{V}$ where $\mathcal{V}$ is a class of functions with $\mathcal{N}(\mathcal{V}, \varepsilon)$ the $\varepsilon$-covering number of $\mathcal{V}$ with respective to the distance $\mathrm{dist}(f, f') := \sup_{s \in \mathcal{S}} |f(s) - f'(s)|$.

We assume there exists a deterministic $C_\sigma > 0$ and $\mathcal{A}_{h,k}$ (which is $\mathcal{F}_{h,k}$-measurable) such that $\mathcal{A}_{h,k} \subseteq \left\{\sigma_{h,k}^2 \geqslant (\mathbb{V}_h f)(s_{h,k}, a_{h,k})/C_\sigma^2\right\}$ for all $k \in [K]$. Let $\mu_{h,k}$ be defined equation A.2 and $\sigma_{h,k}, H_{h,k}$ be defined in our algorithm. Under any of the following conditions, with probability at least $1 - \delta/H$, it follows for all $k \in [K]\bigcup\{0\}$,

$$\mu_h^* \in \left\{\mu : \|(\mu - \mu_{h,k})f\|_{H_{h,k}} \leqslant \beta\right\}. \tag{F.1}$$

(i)  If $f(\cdot)$ is a deterministic function and $\bigcap_{k \in [K]} \mathcal{A}_{h,k}$ is true, equation F.1 holds with

$$\beta = 8C_\sigma\sqrt{d\log\left(1 + \frac{K}{\sigma_{\min}^2 d\lambda}\right)\log\frac{4HK^2}{\delta}} + \frac{8C_0}{d^{2.5}H}\log\frac{4HK^2}{\delta} + \sqrt{d\lambda}C_0.$$

(ii)  If $f(\cdot)$ is a random function and $\bigcap_{k \in [K]} \mathcal{A}_{h,k}$ is true, equation F.1 holds with

$$\beta = 8C_\sigma\sqrt{d\log\left(1 + \frac{K}{\sigma_{\min}^2 d\lambda}\right)\log\frac{4HK^2N_0}{\delta}} + \frac{8C_0}{d^{2.5}H\mathcal{H}}\log\frac{4HK^2N_0}{\delta} + 3\sqrt{d\lambda}C_0$$

where $N_0 = |\mathcal{N}(\mathcal{V}, \varepsilon_0)|$ and $\varepsilon_0 = \min\left\{C_\sigma\sigma_{\min}, \frac{\lambda C_0\sqrt{d}}{K}\sigma_{\min}^2\right\}$.

(iii)  If $f(\cdot)$ is a random function, equation F.1 holds with

$$\beta = \frac{2C_0}{\sigma_{\min}}\sqrt{d\log\left(1 + \frac{K}{\sigma_{\min}^2 d\lambda}\right) + \log\frac{N_1}{\delta}} + 3\sqrt{d\lambda}C_0.$$

where $N_1 = |\mathcal{N}(\mathcal{V}, \varepsilon_1)|$ and $\varepsilon_1 = \frac{\lambda C_0 \sqrt{d}}{K} \sigma_{\min}^2$.

Using the last item suffices to prove Lemma D.2.

*Proof of Lemma D.2.* Let $\mathcal{V}^+$ denote the class of optimistic value functions mapping from $\mathcal{S}$ to $\mathbb{R}$ with the parametric form given in equation G.1 and $\mathcal{V}^-$ the class of pessimistic value functions with the parametric form given in equation G.2. By Lemma G.8 and Lemma G.7,

$$\log \mathcal{N}(\mathcal{V}^{\pm}, \varepsilon) \leqslant \left[ d \log \left( 1 + \frac{4L}{\varepsilon} \right) + d^2 \log \left( 1 + \frac{8d^{1/2} B^2}{\lambda \varepsilon^2} \right) \right] \tag{F.2}$$

where $B \geqslant \beta_0$ and $L = W + \mathcal{H}\sqrt{\frac{dK}{\lambda}}$.

(i) Let $\boldsymbol{f} = \overline{\boldsymbol{V}}_{h+1}^k$. One can find that $\boldsymbol{f} \in \mathcal{V}_f^+$ with parameter $L = W + \frac{K\mathcal{H}}{\lambda \sigma_{\min}^2}$. To plug in Lemma F.1, we first specify the parameters defined therein. We have $\|\boldsymbol{f}\|_{\infty} \leqslant C_0 = \mathcal{H}$ and $\varepsilon_1 = \frac{\lambda \mathcal{H} \sqrt{d}}{K} \sigma_{\min}^2$. By equation F.2, it follows that

$$\log \mathcal{N}(\mathcal{V}^+, \varepsilon_1)$$
$$\leqslant \left[ d \log \left( 1 + \frac{4LK}{\lambda \mathcal{H} \sqrt{d} \sigma_{\min}^2} \right) + d^2 \log \left( 1 + \frac{8B^2 K^2}{\sqrt{d} \lambda^3 \mathcal{H}^2 \sigma_{\min}^4} \right) \right] \cdot dH \log_2 \left( 1 + \frac{K}{\lambda \sigma_{\min}^2} \right)$$
$$\leqslant \frac{2}{\log 2} d^3 H \iota_0^2 \leqslant 3 d^3 H \iota_0^2,$$

By the third condition of Lemma F.1, with probability at least $1 - \frac{\delta}{2H}$, $\left\| (\boldsymbol{\mu}_h^* - \boldsymbol{\mu}_{h,k-1}) \widehat{\boldsymbol{V}}_{h+1}^k \right\| \leqslant \beta_0$ for all $k \in [K]$. Similarly, we can also show that with probability at least $1 - \frac{\delta}{2H}$, $\left\| (\boldsymbol{\mu}_h^* - \boldsymbol{\mu}_{h,k-1}) \check{\boldsymbol{V}}_{h+1}^k \right\| \leqslant \beta_0$ for all $k \in [K]$. Putting them together finishes the proof.

(ii) The analysis on $\underline{\boldsymbol{V}}_{h+1}^k$ is similar to (i).

(iii) The analysis on $[\overline{\boldsymbol{V}}_{h+1}^k]^2$ is similar to (i) except for the following two changes. First, $C_0 = \mathcal{H}^2$ and $\varepsilon_1' = \frac{\lambda \mathcal{H}^2 \sqrt{d}}{K} \sigma_{\min}^2$. Second, with $[\mathcal{V}^+]^2 = \{f^2 : f \in \mathcal{V}^+\}$, we have $[\overline{V}_{h+1}^k]^2 \in [\mathcal{V}^+]^2$ and

$$\log \mathcal{N}([\mathcal{V}^+]^2, \varepsilon_1') \overset{(a)}{\leqslant} \log \mathcal{N}(\mathcal{V}^+, \frac{\varepsilon_1'}{2\mathcal{H}}) \leqslant \log \mathcal{N}(\mathcal{V}^+, \frac{\varepsilon_1}{2}) \leqslant 3 d^3 H \iota_0^2.$$

Here $(a)$ uses the fact that the $\frac{\varepsilon_1'}{2\mathcal{H}}$-cover of $\mathcal{V}^+$ is a $\varepsilon_1$-cover of $[\mathcal{V}^+]^2$ (which is also supported by Lemma G.9).

$\square$

## F.3 Proof of Lemma F.1

*Proof of Lemma F.1.* Since the case of $k = 0$ is trivial, we focus on $k \in [K]$. By definition,

$$\boldsymbol{\mu}_{h,k} = \boldsymbol{H}_{h,k}^{-1} \sum_{j=1}^{k} \sigma_{h,j}^{-2} \boldsymbol{\phi}_{h,j} \boldsymbol{\delta}(s_{h+1,j})^{\top} = \boldsymbol{H}_{h,k}^{-1} \sum_{j=1}^{k} \sigma_{h,j}^{-2} \boldsymbol{\phi}_{h,j} \left( \boldsymbol{\phi}_{h,j}^{\top} \boldsymbol{\mu}_h^* - \boldsymbol{\varepsilon}_{h,j} \right)^{\top}$$

$$= \boldsymbol{\mu}_h^* - \lambda \boldsymbol{H}_{h,k}^{-1} \boldsymbol{\mu}_h^* - \boldsymbol{H}_{h,k}^{-1} \sum_{j=1}^{k} \sigma_{h,j}^{-2} \boldsymbol{\phi}_{h,j} \boldsymbol{\varepsilon}_{h,j}^{\top}.$$

By the triangle inequality, it follows that

$$\|(\boldsymbol{\mu}_h^* - \boldsymbol{\mu}_{h,k})\boldsymbol{f}\|_{\boldsymbol{H}_{h,k}} \leqslant \lambda\|\boldsymbol{H}_{h,k}^{-1}\boldsymbol{\mu}_h^*\boldsymbol{f}\|_{\boldsymbol{H}_{h,k}} + \left\|\boldsymbol{H}_{h,k}^{-1}\sum_{j=1}^{k}\sigma_{h,j}^{-2}\boldsymbol{\phi}_{h,j}\boldsymbol{\varepsilon}_{h,j}^\top\boldsymbol{f}\right\|_{\boldsymbol{H}_{h,k}}$$

$$= \lambda\|\boldsymbol{\mu}_h^*\boldsymbol{f}\|_{\boldsymbol{H}_{h,k}^{-1}} + \left\|\sum_{j=1}^{k}\sigma_{h,j}^{-2}\boldsymbol{\phi}_{h,j}\boldsymbol{\varepsilon}_{h,j}^\top\boldsymbol{f}\right\|_{\boldsymbol{H}_{h,k}^{-1}}$$

$$\leqslant \sqrt{d\lambda}C_0 + \left\|\sum_{j=1}^{k}\sigma_{h,j}^{-2}\boldsymbol{\phi}_{h,j}\boldsymbol{\varepsilon}_{h,j}^\top\boldsymbol{f}\right\|_{\boldsymbol{H}_{h,k}^{-1}}$$

where the last inequality uses $\|\boldsymbol{\mu}_h^*\boldsymbol{f}\| \leqslant \sqrt{d}C_0$.

- Assume $f(\cdot)$ is a deterministic function. To evoke Lemma G.3, we set $\mathcal{G}_j = \mathcal{F}_{h,j}, \boldsymbol{x}_j = \sigma_{h,j}^{-1}\boldsymbol{\phi}_{h,j}, \eta_j = \sigma_{h,j}^{-1}\boldsymbol{\varepsilon}_{h,j}^\top\boldsymbol{f} \cdot 1_{\mathcal{A}_{h,j}}$ and $\boldsymbol{Z}_k = \lambda\boldsymbol{I} + \sum_{j=1}^{k}\sigma_{h,j}^{-2}\boldsymbol{\phi}_{h,j}\boldsymbol{\phi}_{h,j}^\top = \boldsymbol{H}_{h,k}$. Here $1_{\mathcal{A}}$ is the indicator function of the event $\mathcal{A}$.

  Clearly $\boldsymbol{x}_j \in \mathcal{G}_j, \mathbb{E}[\eta_j|\mathcal{G}_j] = 0$ and $\mathbb{E}[\eta_j^2|\mathcal{G}_j] \leqslant C_\sigma^2$. We also have $\|\boldsymbol{x}_j\| \leqslant \sigma_{\min}^{-1}, |\eta_j| \leqslant 2C_0\sigma_{h,j}^{-1}$ and $\|\boldsymbol{x}_j\|_{\boldsymbol{Z}_{j-1}} = w_{h,j}$. As a result, $|\eta_j|\min\left\{1, \|\boldsymbol{x}_j\|_{\boldsymbol{Z}_{j-1}}\right\} \leqslant 2C_0\frac{w_{h,j}}{\sigma_{h,j}} \leqslant \frac{2C_0}{\mathcal{H}d^{2.5}H}$ where the last inequality uses $\sigma_{h,j}^2 \geqslant \mathcal{H}d^{2.5}H\|\boldsymbol{\phi}_{h,j}\|_{\boldsymbol{H}_{h,k-1}^{-1}}$ (which is equivalent to $\frac{w_{h,j}}{\sigma_{h,j}} \leqslant (d^{2.5}H\mathcal{H})^{-1}$). By Lemma Lemma G.3, it follows that with probability $1 - \frac{\delta}{H}$, for all $k \in [K]$,

$$\left\|\sum_{j=1}^{k}\sigma_{h,j}^{-2}\boldsymbol{\phi}_{h,j}\boldsymbol{\varepsilon}_{h,j}^\top\boldsymbol{f}1_{\mathcal{A}_{h,j}}\right\|_{\boldsymbol{H}_{h,k}^{-1}} = \left\|\sum_{j=1}^{k}\boldsymbol{x}_j\eta_j\right\|_{\boldsymbol{Z}_k^{-1}}$$

$$\leqslant 8C_\sigma\sqrt{d\log\left(1 + \frac{K}{\sigma_{\min}^2 d\lambda}\right)\log\frac{4HK^2}{\delta}} + \frac{8C_0}{\mathcal{H}d^{2.5}H}\log\frac{4HK^2}{\delta}.$$

  Finally, on the event $\bigcap_{k\in[K]}\mathcal{A}_{h,k}$, we will have all the indicator functions equal to one.

- If $f(\cdot)$ is a random function, we would use a covering argument to handle the possible correlation between $f(\cdot)$ and history data, which would, unfortunately, enlarge $\beta$.

  Denote the $\varepsilon_0$-net of $\mathcal{V}$ by $\mathcal{N}(\mathcal{V}, \varepsilon_0)$ where $\varepsilon_0 = \min\left\{C_\sigma\sigma_{\min}, \frac{\lambda C_0\sqrt{d}}{K}\sigma_{\min}^2\right\}$. Hence, for any $f \in \mathcal{V}$, there exists $\bar{\boldsymbol{f}} \in \mathcal{N}(\mathcal{V}, \varepsilon_0)$ such that $\|\bar{\boldsymbol{f}} - \boldsymbol{f}\|_\infty = \sup_{s\in\mathcal{S}}|f(s) - \bar{f}(s)| \leqslant \varepsilon_0$. Then,

$$\left\|\sum_{j=1}^{k}\sigma_{h,j}^{-2}\boldsymbol{\phi}_{h,j}\boldsymbol{\varepsilon}_{h,j}^\top\boldsymbol{f}\right\|_{\boldsymbol{H}_{h,k}^{-1}} \leqslant \underbrace{\left\|\sum_{j=1}^{k}\sigma_{h,j}^{-2}\boldsymbol{\phi}_{h,j}\boldsymbol{\varepsilon}_{h,j}^\top\bar{\boldsymbol{f}}\right\|_{\boldsymbol{H}_{h,k}^{-1}}}_{(I)} + \underbrace{\left\|\sum_{j=1}^{k}\sigma_{h,j}^{-2}\boldsymbol{\phi}_{h,j}\boldsymbol{\varepsilon}_{h,j}^\top(\boldsymbol{f} - \bar{\boldsymbol{f}})\right\|_{\boldsymbol{H}_{h,k}^{-1}}}_{(II)}.$$

For the term $(II)$, due to $\|\boldsymbol{\phi}_{h,j}\| \leqslant 1$ and $|\boldsymbol{\varepsilon}_{h,j}^\top(\boldsymbol{f} - \bar{\boldsymbol{f}})| \leqslant \|\boldsymbol{\varepsilon}_{h,j}\|_1\|\boldsymbol{f} - \bar{\boldsymbol{f}}\|_\infty \leqslant 2\varepsilon_0$, we have

$$\left\|\sum_{j=1}^{k}\sigma_{h,j}^{-2}\boldsymbol{\phi}_{h,j}\boldsymbol{\varepsilon}_{h,j}^\top(\boldsymbol{f} - \bar{\boldsymbol{f}})\right\|_{\boldsymbol{H}_{h,k}^{-1}} \leqslant \frac{2K\varepsilon_0}{\sigma_{\min}^2\sqrt{\lambda}} \leqslant 2\sqrt{d\lambda}C_0.$$

For the term $(I)$, we define $\mathcal{V}_{h,k} = \left\{f' \in \mathcal{V} : 4C_\sigma^2\sigma_{h,k}^2 \geqslant (\mathbb{V}_h f')(s_{h,k}, a_{h,k})\right\}$. Since the definition of $\mathcal{V}_{h,k}$ involves only $\sigma_{h,k}, s_{h,k}, a_{h,k} \in \mathcal{F}_{h,k}$, for any fixed function $f \in \mathcal{V}$, $1_{f\in\mathcal{V}_{h,k}}$ is $\mathcal{F}_{h,k}$-measurable. On the event $\mathcal{A}_{h,k}$, by definition of $\varepsilon_0$,

$$(\mathbb{V}_h\bar{f})(s_{h,k}, a_{h,k}) \leqslant 2(\mathbb{V}_h f)(s_{h,k}, a_{h,k}) + 2(\mathbb{V}_h(\bar{f} - f))(s_{h,k}, a_{h,k}) \leqslant 2C_\sigma^2\sigma_{h,k}^2 + 2\varepsilon_0^2 \leqslant 4C_\sigma^2\sigma_{h,k}^2.$$

Hence, $\mathcal{A}_{h,k} \subseteq \left\{\sigma_{h,k}^2 \geqslant (\mathbb{V}_h f)(s_{h,k}, a_{h,k})/C_\sigma^2\right\} \subseteq \{\exists \bar{f} \in \mathcal{N}(\mathcal{V}, \varepsilon_0) \bigcap \mathcal{V}_{h,k}\}$ for all $k \in [K]$.

In the following, we will evoke Lemma G.3 to analyze the term $(I)$. For any fixed $f' \in \mathcal{V}$, we set $\mathcal{G}_j = \mathcal{F}_{h,j}, \boldsymbol{x}_j = \sigma_{h,j}^{-1} \boldsymbol{\phi}_{h,j}, \eta_j = \sigma_{h,j}^{-1} \boldsymbol{\varepsilon}_{h,j}^\top \boldsymbol{f}' \cdot 1_{f' \in \mathcal{V}_{h,k}}$ and $\boldsymbol{Z}_k = \lambda \boldsymbol{I} + \sum_{j=1}^k \sigma_{h,j}^{-2} \boldsymbol{\phi}_{h,j} \boldsymbol{\phi}_{h,j}^\top = \boldsymbol{H}_{h,k}$. Moreover, due to the choice of $\sigma_{h,j}$, it follows that

$$\left| \eta_j \min\left\{1, \|\boldsymbol{x}_j\|_{\boldsymbol{Z}_{j-1}^{-1}}\right\} \right| \leqslant \left| \frac{C_0}{\sigma_{h,j}} \right| \cdot \left\| \frac{\boldsymbol{\phi}_{h,j}}{\sigma_{h,j}} \right\|_{\boldsymbol{H}_{h,j-1}^{-1}} \leqslant C_0 \frac{b_{h,j}}{\sigma_{h,j}^2} \leqslant \frac{C_0}{d^{2.5} H \mathcal{H}}.$$

By Lemma G.3 and the union bound, it follows that with probability $1 - \frac{\delta}{H}$, for all $k \in [K]$,

$$\sup_{f' \in \mathcal{N}(\mathcal{V}, \varepsilon_0)} \left\| \sum_{j=1}^k \sigma_{h,j}^{-2} \boldsymbol{\phi}_{h,j} \boldsymbol{\varepsilon}_{h,j}^\top \boldsymbol{f}' 1_{f' \in \mathcal{V}_{h,k}} \right\|_{\boldsymbol{H}_{h,k}^{-1}}$$
$$\leqslant 8 C_\sigma \sqrt{d \log\left(1 + \frac{K}{\sigma_{\min}^2 d \lambda}\right) \log \frac{4 H K^2 N_0}{\delta}} + \frac{8 C_0}{d^{2.5} H \mathcal{H}} \log \frac{4 H K^2 N_0}{\delta}.$$

where $N_0 = |\mathcal{N}(\mathcal{V}, \varepsilon_0)|$.

As a result, we know that $\left\| \sum_{j=1}^k \sigma_{h,j}^{-2} \boldsymbol{\phi}_{h,j} \boldsymbol{\varepsilon}_{h,j}^\top \bar{\boldsymbol{f}} 1_{\bar{f} \in \mathcal{V}_{h,k}} \right\|_{\boldsymbol{H}_{h,k}^{-1}}$ is no more than the RHS of the last inequality.

On the event $\bigcap_{k \in [K]} \mathcal{A}_{h,k}$, we have $\bar{f} \in \bigcap_{k \in [K]} \mathcal{V}_{h,k}$ and thus all the indicator functions equal to one, completing the proof.

- The proof is almost similar to the second item except that we use Lemma G.4 to analyze the term $(I)$. Noticing we also have $|\eta_j| = |\sigma_{h,j}^{-1} \boldsymbol{\varepsilon}_{h,j}^\top \boldsymbol{f}'| \leqslant \frac{2C_0}{\sigma_{\min}}$. By Lemma G.4 and the union bound, it follows that with probability $1 - \frac{\delta}{H}$, for all $k \in [K]$,

$$\sup_{f' \in \mathcal{N}(\mathcal{V}_f, \varepsilon_1)} \left\| \sum_{j=1}^k \sigma_{h,j}^{-2} \boldsymbol{\phi}_{h,j} \boldsymbol{\varepsilon}_{h,j}^\top \boldsymbol{f}' 1_{f' \in \mathcal{V}_{h,k}} \right\|_{\boldsymbol{H}_{h,k}^{-1}} \leqslant \frac{2C_0}{\sigma_{\min}} \sqrt{d \log\left(1 + \frac{K}{\sigma_{\min}^2 d \lambda}\right) + \log \frac{N_1}{\delta}}.$$

Pay attention that here we don't utilize the variance information so that we change $N_0 := |\mathcal{N}(\mathcal{V}, \varepsilon_0)|$ to $N_1 := |\mathcal{N}(\mathcal{V}, \varepsilon_1)|$ and don't require $\bigcap_{k \in [K]} \mathcal{A}_{h,k}$ is true.

$\square$

## F.4 Proof of Lemma D.3

*Proof of Lemma D.3.* The proof idea of Lemma D.3 is similar to that of Lemma D.1 except that we pay more attention to the reward variance.

Given that $\mathcal{B}_{R^2}$ holds, we have $\boldsymbol{\psi}_h^* \in \widetilde{\mathcal{R}}_{h,k}$ for all $h \in [H]$ and $k \in [K] \bigcup \{0\}$.

We will prove the lemma by induction over $k$. When $k = 0$, we have $\boldsymbol{\theta}_{h,0} = 0, \boldsymbol{H}_{h,0} = \lambda \boldsymbol{I}$ and $\|\boldsymbol{\theta}_{h,0} - \boldsymbol{\theta}_h^*\|_{\boldsymbol{H}_{h,0}} = \sqrt{\lambda} \|\boldsymbol{\theta}_h^*\| \leqslant \sqrt{\lambda} W \leqslant \beta_R$ for all $h \in [H]$. If we suppose $\boldsymbol{\theta}_h^* \in \mathcal{R}_{h,j}$ holds for all $h \in [H]$ and $j \in [k-1]$, we are going to prove $\boldsymbol{\theta}_h^* \in \mathcal{R}_{h,k}$ uniformly for $h \in [H]$. The first thing we will show is

$$\sigma_{h,j}^2 \geqslant [\widehat{\mathbb{V}}_h R_h](s_{h,j}, a_{h,j}) + R_{h,j} \quad \text{for all} \quad h \in [H] \text{ and } j \in [k]. \tag{F.3}$$

Notice that $[\mathbb{V}_h R_h](s_{h,k}, a_{h,k}) = \langle \widetilde{\boldsymbol{\phi}}_{h,k}, \boldsymbol{\psi}_h^* \rangle - \langle \boldsymbol{\phi}_{h,k}, \boldsymbol{\theta}_h^* \rangle^2$. We then have for all $h \in [H], j \in [k]$,

$$|[\widehat{\mathbb{V}}_h R_h - \mathbb{V}_h R_h](s_{h,j}, a_{h,j})|$$
$$\leqslant \left| \langle \widetilde{\boldsymbol{\phi}}_{h,j}, \boldsymbol{\psi}_{h,j-1} \rangle - \langle \widetilde{\boldsymbol{\phi}}_{h,j}, \boldsymbol{\psi}_h^* \rangle \right| + \left| \langle \boldsymbol{\phi}_{h,j}, \boldsymbol{\theta}_h^* \rangle^2 - \langle \boldsymbol{\phi}_{h,j}, \boldsymbol{\theta}_{h,j-1} \rangle_{[0,\mathcal{H}]}^2 \right|$$
$$\leqslant |\langle \widetilde{\boldsymbol{\phi}}_{h,k}, \boldsymbol{\psi}_{h,k-1} - \boldsymbol{\psi}_h^* \rangle| + 2\mathcal{H} |\langle \boldsymbol{\phi}_{h,k}, \boldsymbol{\theta}_{h,j-1} - \boldsymbol{\theta}_h^* \rangle|$$

$$\leqslant \|\widetilde{\phi}_{h,j}\|_{\widetilde{H}_{h,j-1}^{-1}}\|\psi_{h,j-1} - \psi_h^*\|_{\widetilde{H}_{h,j-1}} + 2\mathcal{H}\|\phi_{h,j}\|_{H_{h,j-1}^{-1}}\|\theta_{h,j-1} - \theta_h^*\|_{H_{h,j-1}}$$

$$\leqslant \beta_{R^2}\|\widetilde{\phi}_{h,j}\|_{\widetilde{H}_{h,j-1}^{-1}} + 2\mathcal{H}\beta_R\|\phi_{h,j}\|_{H_{h,j-1}^{-1}} = R_{h,j}$$

where the last inequality uses the hypothesis and the condition that $\mathcal{B}_{R^2}$ holds. As a result, we establish equation F.3.

Let $\mathcal{G}_{h,j} = \sigma(\mathcal{F}_{h-1,j} \bigcup \{s_{h,j}, a_{h,j}\})$. One can show that both $R_{h,j}$ and $\sigma_{h,j}^2$ are $\mathcal{G}_{h,j}$-measurable. As a result, the event $\mathcal{E}_{h,j} := \left\{\sigma_{h,j}^2 \geqslant [\mathbb{V}_h R_h](s_{h,j}, a_{h,j})\right\}$ is also $\mathcal{G}_{h,j}$-measurable. On the event $\mathcal{B}_{R^2}$, it is obvious that $\bigcap_{h\in[H]}\bigcap_{j\in[k]}\mathcal{E}_{h,j}$ is true since equation F.3 is true.

On the other hand, we set $\varepsilon_{h,j} = \frac{r_{h,j} - \langle\phi_{h,j}, \theta_h^*\rangle}{\sigma_{h,j}}1_{\mathcal{E}_{h,j}}$ as the standardized reward. We then have $\varepsilon_{h,j} \in \mathcal{F}_{h,j}$, $\mathbb{E}[\varepsilon_{h,j}|\mathcal{G}_{h,j}] = 0$ and $\text{Var}[\varepsilon_{h,j}|\mathcal{G}_{h,j}] \leqslant 1$. We define $\widehat{\theta}_{h,k}$ as the solution of adaptive Huber regression to the response $\{r_{h,j}1_{\mathcal{E}_{h,j}}\}_{j\in[k]}$ and the feature $\{\phi_{h,j}1_{\mathcal{E}_{h,j}}\}_{j\in[k]}$. We also define $\widehat{H}_{h,k-1}$ as the counterpart matrix of $H_{h,k}$ obtained by replacing $\phi_{h,k}$ with $\phi_{h,k}1_{\mathcal{E}_{h,k}}$. We then apply Theorem 2.1 to analyze the concentration of $\widehat{\theta}_{h,k}$. With probability at least $1 - 3\delta$, it follows that $\|\widehat{\theta}_{h,k} - \theta_h^*\|_{\widehat{H}_{h,k-1}} \leqslant \beta_R$ for all $h \in [H]$ and $k \in [K]$. Because $\mathcal{B}_{R^2}$ is true, all indicator functions are equal to one. Therefore, we have $\widehat{\theta}_{h,k} = \theta_{h,k}$ and $\widehat{H}_{h,k-1} = H_{h,k-1}$, implying $\theta_h^* \in \mathcal{R}_{h,k}$ uniformly for $h \in [H]$. $\square$

## F.5 Proof of Lemma D.4

*Proof of Lemma D.4.* By symmetry, we only prove the RHS inequality, or say, the optimism inequality. We prove it by induction. The statement is true for $h = H + 1$ since both $V_{H+1}^*(\cdot) = \overline{V}_{H+1}^k(\cdot) = 0$ for all $k \in [K]$. Assume the statement is also true for $h + 1$, implying $V_{h+1}^*(\cdot) \leqslant \overline{V}_{h+1}^k(\cdot)$ for all $k \in [K]$. We assume there exists a sequence of updating episodes $1 \leqslant k_1 < \cdots < k_{N_k} \leqslant K$ such that

$$\overline{Q}_h^k(\cdot,\cdot) = \min_{i\in[N_k]}\left\{\langle\phi(\cdot,\cdot), \theta_{h,k_i-1} + \mu_{h,k_i-1}\overline{V}_{h+1}^{k_i}\rangle + \beta\|\phi(\cdot,\cdot)\|_{H_{h,k_i-1}^{-1}}, \mathcal{H}\right\}. \tag{F.4}$$

Using $Q_h^*(s,a) = \langle\phi(s,a), \theta_h^* + \mu_h^* V_{h+1}^*\rangle$, we have for any $(s,a) \in \mathcal{S} \times \mathcal{A}$ and $k \in [K]$,

$$\langle\phi(\cdot,\cdot), \theta_{h,k-1} + \mu_{h,k-1}\overline{V}_{h+1}^k\rangle + \beta\|\phi(\cdot,\cdot)\|_{H_{h,k-1}^{-1}} - Q_h^*(\cdot,\cdot)$$

$$= \langle\phi(\cdot,\cdot), \theta_{h,k-1} - \theta_h^*\rangle + \langle\phi(\cdot,\cdot), \mu_{h,k-1}\overline{V}_{h+1}^k - \mu_h^* V_{h+1}^*\rangle + \beta\|\phi(\cdot,\cdot)\|_{H_{h,k-1}^{-1}}$$

$$\overset{(a)}{\geqslant} \langle\phi(\cdot,\cdot), \theta_{h,k-1} - \theta_h^*\rangle + \langle\phi(\cdot,\cdot), (\mu_{h,k-1} - \mu_h^*)\overline{V}_{h+1}^k\rangle + \beta\|\phi(\cdot,\cdot)\|_{H_{h,k-1}^{-1}}$$

$$\overset{(b)}{\geqslant} \|\phi(\cdot,\cdot)\|_{H_{h,k-1}^{-1}}\left[-\|\theta_{h,k-1} - \theta_h^*\|_{H_{h,k-1}} - \|(\mu_{h,k-1} - \mu_h^*)\overline{V}_{h+1}^k\|_{H_{h,k-1}} + \beta\right] \overset{(c)}{\geqslant} 0$$

where $(a)$ uses $\langle\phi(s,a), \mu_h^*(\overline{V}_{h+1}^k - V_{h+1}^*)\rangle = \mathbb{P}_h(\overline{V}_{h+1}^k - V_{h+1}^*)(s,a) \geqslant 0$ from the hypothesis, $(b)$ follows from Cauchy-Schwarz inequality and $(c)$ uses $\|\theta_{h,k-1} - \theta_h^*\|_{H_{h,k-1}} + \|(\mu_{h,k-1} - \mu_h^*)\overline{V}_{h+1}^k\|_{H_{h,k-1}} \leqslant \beta_R + \beta_V = \beta$ on the event $\mathcal{B}_R \bigcap \mathcal{B}_h$.

As a result, by the last inequality and equation F.4, it follows that for all $k \in [K]$, $\overline{Q}_h^k(\cdot,\cdot) - Q_h^*(\cdot,\cdot) \geqslant 0$. Taking maximum over actions, we have $\overline{V}_h^k(\cdot) \geqslant V_h^*(\cdot)$ for all $k \in [K]$, which implies the case of $h$ is also true. $\square$

## F.6 Proof of Lemma D.5

*Proof of Lemma D.5.* The proof technique has been used in Lemma C.13 in (Hu et al., 2022) and Lemma 7.2 in (He et al., 2022). We include the proof for completeness. By definition,

$$[\mathbb{V}_h V_{h+1}^*](s_{h,k}, a_{h,k}) = \langle\mu_h^*[V_{h+1}^*]^2, \phi_{h,k}\rangle - \langle\mu_h^* V_{h+1}^*, \phi_{h,k}\rangle^2$$

$$[\widehat{\mathbb{V}}_h \overline{V}_{h+1}^k](s_{h,k}, a_{h,k}) = \langle \boldsymbol{\mu}_{h,k-1}[\overline{\boldsymbol{V}}_{h+1}^k]^2, \boldsymbol{\phi}_{h,k-1} \rangle_{[0,\mathcal{H}^2]} - \langle \boldsymbol{\mu}_{h,k-1}\overline{\boldsymbol{V}}_{h+1}^k, \boldsymbol{\phi}_{h,k} \rangle_{[0,\mathcal{H}]}^2.$$

Therefore, it follows that

$$\left| \left[ \mathbb{V}_h V_{h+1}^* - \widehat{\mathbb{V}}_h \overline{V}_{h+1}^k \right](s_{h,k}, a_{h,k}) \right|$$
$$\leqslant \left| \left[ \mathbb{V}_h \overline{V}_{h+1}^k - \widehat{\mathbb{V}}_h \overline{V}_{h+1}^k \right](s_{h,k}, a_{h,k}) \right| + \left| \left[ \mathbb{V}_h V_{h+1}^* - \mathbb{V}_h \overline{V}_{h+1}^k \right](s_{h,k}, a_{h,k}) \right|.$$

We then bound the two terms in the RHS of the last inequality as follows.

$$\left| \left[ \mathbb{V}_h \overline{V}_{h+1}^k - \widehat{\mathbb{V}}_h \overline{V}_{h+1}^k \right](s_{h,k}, a_{h,k}) \right|$$
$$\leqslant \left| \langle \boldsymbol{\mu}_h^*[\overline{\boldsymbol{V}}_{h+1}^k]^2, \boldsymbol{\phi}_{h,k} \rangle - \langle \boldsymbol{\mu}_{h,k-1}[\overline{\boldsymbol{V}}_{h+1}^k]^2, \boldsymbol{\phi}_{h,k} \rangle_{[0,\mathcal{H}^2]} \right|$$
$$\quad + \left| \langle \boldsymbol{\mu}_h^*\overline{\boldsymbol{V}}_{h+1}^k, \boldsymbol{\phi}_{h,k} \rangle^2 - \langle \boldsymbol{\mu}_{h,k-1}\overline{\boldsymbol{V}}_{h+1}^k, \boldsymbol{\phi}_{h,k} \rangle_{[0,\mathcal{H}]}^2 \right|$$
$$\leqslant \left| \langle (\boldsymbol{\mu}_h^* - \boldsymbol{\mu}_{h,k-1})[\overline{\boldsymbol{V}}_{h+1}^k]^2, \boldsymbol{\phi}_{h,k} \rangle \right| + 2\mathcal{H} \cdot \left| \langle \boldsymbol{\mu}_h^*\overline{\boldsymbol{V}}_{h+1}^k, \boldsymbol{\phi}_{h,k} \rangle - \langle \boldsymbol{\mu}_{h,k-1}\overline{\boldsymbol{V}}_{h+1}^k, \boldsymbol{\phi}_{h,k} \rangle_{[0,\mathcal{H}]} \right|$$
$$\leqslant \left\| (\boldsymbol{\mu}_{h,k-1} - \boldsymbol{\mu}_h^*)[\overline{\boldsymbol{V}}_{h+1}^k]^2 \right\|_{\boldsymbol{H}_{h,k-1}} \|\boldsymbol{\phi}_{h,k}\|_{\boldsymbol{H}_{h,k-1}^{-1}} + 2\mathcal{H}\|\boldsymbol{\phi}_{h,k}\|_{\boldsymbol{H}_{h,k-1}^{-1}} \left\| (\boldsymbol{\mu}_{h,k-1} - \boldsymbol{\mu}_h^*)\overline{\boldsymbol{V}}_{h+1}^k \right\|_{\boldsymbol{H}_{h,k-1}}$$

where the second inequality uses the fact that both $\langle \boldsymbol{\mu}_h^*\overline{\boldsymbol{V}}_{h+1}^k, \boldsymbol{\phi}_{h,k} \rangle$ and $\langle \boldsymbol{\mu}_{h,k-1}\overline{\boldsymbol{V}}_{h+1}^k, \boldsymbol{\phi}_{h,k} \rangle_{[0,\mathcal{H}]}$ lie between $0$ and $\mathcal{H}$. Similarly, it follows that

$$\left| \left[ \mathbb{V}_h V_{h+1}^* - \mathbb{V}_h \overline{V}_{h+1}^k \right](s_{h,k}, a_{h,k}) \right|$$
$$\leqslant \left| \mathbb{P}_h[[V_{h+1}^*]^2 - [\overline{V}_{h+1}^k]^2](s_{h,k}, a_{h,k}) \right| + \left| [\mathbb{P}_h V_{h+1}^*]^2(s_{h,k}, a_{h,k}) - [\mathbb{P}_h\overline{V}_{h+1}^k]^2(s_{h,k}, a_{h,k}) \right|$$
$$\leqslant \left| \mathbb{P}_h[(\overline{V}_{h+1}^k - V_{h+1}^*)(\overline{V}_{h+1}^k + V_{h+1}^*)](s_{h,k}, a_{h,k}) \right|$$
$$\quad + \left| [\mathbb{P}_h\overline{V}_{h+1}^k - \mathbb{P}_h V_{h+1}^*][[\mathbb{P}_h\overline{V}_{h+1}^k + \mathbb{P}_h V_{h+1}^*]](s_{h,k}, a_{h,k}) \right|$$
$$\leqslant 4\mathcal{H} \cdot \mathbb{P}_h[\overline{V}_{h+1}^k - V_{h+1}^*](s_{h,k}, a_{h,k})$$
$$\leqslant 4\mathcal{H} \cdot \mathbb{P}_h[\overline{V}_{h+1}^k - \underline{V}_{h+1}](s_{h,k}, a_{h,k})$$
$$\leqslant 4\mathcal{H} \cdot \widehat{\mathbb{P}}_{h,k}[\overline{V}_{h+1}^k - \underline{V}_{h+1}](s_{h,k}, a_{h,k})$$
$$\quad + 4\mathcal{H}\|\boldsymbol{\phi}_{h,k}\|_{\boldsymbol{H}_{h,k-1}^{-1}} \cdot \left[ \left\| (\boldsymbol{\mu}_{h,k-1} - \boldsymbol{\mu}_h^*)\overline{\boldsymbol{V}}_{h+1}^k \right\|_{\boldsymbol{H}_{h,k-1}} + \left\| (\boldsymbol{\mu}_{h,k-1} - \boldsymbol{\mu}_h^*)\underline{\boldsymbol{V}}_{h+1}^k \right\|_{\boldsymbol{H}_{h,k-1}} \right]$$

where for the third and fourth inequalities we use the optimism and pessimism in Lemma D.4 and the last inequality uses the following result.

$$\left| [\mathbb{P}_h\overline{V}_{h+1}^k - \widehat{\mathbb{P}}_{h,k}\overline{V}_{h+1}^k](s_{h,k}, a_{h,k}) \right| = \left| \langle (\boldsymbol{\mu}_h^* - \boldsymbol{\mu}_{h,k-1})\overline{\boldsymbol{V}}_{h+1}^k, \boldsymbol{\phi}_{h,k} \rangle \right|$$
$$\leqslant \|\boldsymbol{\phi}_{h,k}\|_{\boldsymbol{H}_{h,k-1}^{-1}} \left\| (\boldsymbol{\mu}_{h,k-1} - \boldsymbol{\mu}_h^*)\overline{\boldsymbol{V}}_{h+1}^k \right\|_{\boldsymbol{H}_{h,k-1}}.$$

A similar inequality can be derived for $\left| [\mathbb{P}_h\underline{V}_{h+1}^k - \widehat{\mathbb{P}}_{h,k}\underline{V}_{h+1}^k](s_{h,k}, a_{h,k}) \right|$. Finally, we have

$$\left| \left[ \mathbb{V}_h V_{h+1}^* - \widehat{\mathbb{V}}_h \overline{V}_{h+1}^k \right](s_{h,k}, a_{h,k}) \right|$$
$$\leqslant \left\| (\boldsymbol{\mu}_{h,k-1} - \boldsymbol{\mu}_h^*)[\overline{\boldsymbol{V}}_{h+1}^*]^2 \right\|_{\boldsymbol{H}_{h,k-1}} \|\boldsymbol{\phi}_{h,k}\|_{\boldsymbol{H}_{h,k-1}^{-1}} + 4\mathcal{H}\widehat{\mathbb{P}}_{h,k}(\overline{V}_{h+1}^k - \underline{V}_{h+1}^k)(s_{h,k}, a_{h,k})$$
$$\quad + \mathcal{H}\|\boldsymbol{\phi}_{h,k}\|_{\boldsymbol{H}_{h,k-1}^{-1}} \cdot \left[ 6\left\| (\boldsymbol{\mu}_{h,k-1} - \boldsymbol{\mu}_h^*)\overline{\boldsymbol{V}}_{h+1}^k \right\|_{\boldsymbol{H}_{h,k-1}} + 4\left\| (\boldsymbol{\mu}_{h,k-1} - \boldsymbol{\mu}_h^*)\underline{\boldsymbol{V}}_{h+1}^k \right\|_{\boldsymbol{H}_{h,k-1}} \right].$$

We complete the proof by noting that on the event $\mathcal{B}_0$, we have

$$\max\left\{ \left\| (\boldsymbol{\mu}_h^* - \boldsymbol{\mu}_{h,k-1})\overline{\boldsymbol{V}}_{h+1}^k \right\|_{\boldsymbol{H}_{h,k-1}}, \left\| (\boldsymbol{\mu}_h^* - \boldsymbol{\mu}_{h,k-1})\underline{\boldsymbol{V}}_{h+1}^k \right\|_{\boldsymbol{H}_{h,k-1}} \right\} \leqslant \beta_0,$$

$$\left\| (\boldsymbol{\mu}_h^* - \boldsymbol{\mu}_{h,k-1})[\overline{\boldsymbol{V}}_{h+1}^k]^2 \right\|_{\boldsymbol{H}_{h,k-1}} \leqslant \mathcal{H}\beta_0.$$

$\square$

### F.7 Proof of Lemma D.6

*Proof of Lemma D.6.* For any $j \leqslant k$, we have

$$\left[ \mathbb{V}_h(\overline{V}_{h+1}^k - V_{h+1}^*) \right](s_{h,j}, a_{h,j}) \overset{(a)}{\leqslant} \left[ \mathbb{P}_h(\overline{V}_{h+1}^k - V_{h+1}^*)^2 \right](s_{h,j}, a_{h,j})$$

$$\overset{(b)}{\leqslant} \mathcal{H}\left[ \mathbb{P}_h(\overline{V}_{h+1}^k - \underline{V}_{h+1}^k) \right](s_{h,j}, a_{h,j})$$

$$\overset{(c)}{\leqslant} \mathcal{H}\left[ \mathbb{P}_h(\overline{V}_{h+1}^j - \underline{V}_{h+1}^j) \right](s_{h,j}, a_{h,j})$$

where $(a)$ uses the fact that $\mathrm{Var}(X) \leqslant \mathbb{E}X^2$ for any random variable $X$, $(b)$ uses $0 \leqslant \underline{V}_{h+1}^k(\cdot) \leqslant V_{h+1}^*(\cdot) \leqslant \overline{V}_{h+1}^k(\cdot) \leqslant \mathcal{H}$ on the event $\mathcal{B}_R \bigcap \mathcal{B}_{h+1}$ from Lemma D.4, and $(c)$ uses that $\overline{V}_{h+1}^j(\cdot) \geqslant \overline{V}_{h+1}^k(\cdot)$ and $\underline{V}_{h+1}^j(\cdot) \leqslant \underline{V}_{h+1}^k(\cdot)$ by definition. On the other hand, in the event $\mathcal{B}_0$, we have

$$\max\left\{ \left\| (\boldsymbol{\mu}_h^* - \boldsymbol{\mu}_{h,j-1})\overline{\boldsymbol{V}}_{h+1}^j \right\|_{\boldsymbol{H}_{h,j-1}}, \left\| (\boldsymbol{\mu}_h^* - \boldsymbol{\mu}_{h,j-1})\underline{\boldsymbol{V}}_{h+1}^j \right\|_{\boldsymbol{H}_{h,j-1}} \right\} \leqslant \beta_0.$$

As a result,

$$\left[ \left( \mathbb{P}_h - \widehat{\mathbb{P}}_{h,j} \right) \overline{V}_{h+1}^j \right](s_{h,j}, a_{h,j}) = \langle \boldsymbol{\phi}_{h,j}, (\boldsymbol{\mu}_h^* - \boldsymbol{\mu}_{h,j-1})\overline{\boldsymbol{V}}_{h+1}^j \rangle$$

$$\leqslant \|\boldsymbol{\phi}_{h,j}\|_{\boldsymbol{H}_{h,j-1}^{-1}} \|(\boldsymbol{\mu}_h^* - \boldsymbol{\mu}_{h,j-1})\overline{\boldsymbol{V}}_{h+1}^j\|_{\boldsymbol{H}_{h,j-1}} \leqslant \beta_0 \|\boldsymbol{\phi}_{h,j}\|_{\boldsymbol{H}_{h,j-1}^{-1}}.$$

Similarily, we have $\left[ \left( \mathbb{P}_h - \widehat{\mathbb{P}}_{h,j} \right) \underline{V}_{h+1}^j \right](s_{h,j}, a_{h,j}) \leqslant \beta_0 \|\boldsymbol{\phi}_{h,j}\|_{\boldsymbol{H}_{h,j-1}^{-1}}$. Therefore,

$$\left[ \mathbb{V}_h(\overline{V}_{h+1}^k - V_{h+1}^*) \right](s_{h,j}, a_{h,j})$$

$$\leqslant \mathcal{H}\left[ 2\beta_0 \|\boldsymbol{\phi}_{h,k}\|_{\boldsymbol{H}_{h,j-1}^{-1}} + \left[ \widehat{\mathbb{P}}_{h,j}(\overline{V}_{h+1}^j - \underline{V}_{h+1}^j) \right](s_{h,j}, a_{h,j}) \right] =: E_{h,j}$$

Repeating the above argument, we have a similar inequality for $\overline{V}_{h+1}^k$ due to symmetry. $\square$

### F.8 Proof of Lemma D.7

*Proof of Lemma D.7.* Due to the backward recursion structure, we will use induction (over horizon $h$) to prove this lemma. First, equation D.2 is true for $h = H$ since $\overline{V}_{H+1}^k(\cdot) = \underline{V}_{H+1}^k(\cdot) = 0$ for all $k \in [K]$. Therefore, we have $\mathcal{B}_H$ holds. Assume equation D.2 holds for horizons no smaller than $h + 1$, i.e., $\mathcal{B}_{h+1}$ holds with $h + 1 \leqslant H$. In the following, we will show, once $\mathcal{B}_{h+1} \bigcap \mathcal{B}_0$ holds, $\mathcal{B}_h$ holds with probability at least than $1 - \frac{2\delta}{H}$. Repeating the argument, we have, given $\mathcal{B}_H \bigcap \mathcal{B}_0$ holds, with probability at least $1 - 2\delta$, $\mathcal{B}_1 \bigcap \mathcal{B}_0$ holds. Hence, $\mathbb{P}(\mathcal{B}_0 \bigcap \mathcal{B}_1) \geqslant 1 - 5\delta$.

Note that

$$\max\left\{ \left\| (\boldsymbol{\mu}_h^* - \boldsymbol{\mu}_{h,k-1})\overline{\boldsymbol{V}}_{h+1}^k \right\|_{\boldsymbol{H}_{h,k-1}}, \left\| (\boldsymbol{\mu}_h^* - \boldsymbol{\mu}_{h,k-1})\underline{\boldsymbol{V}}_{h+1}^k \right\|_{\boldsymbol{H}_{h,k-1}} \right\} \leqslant \left\| (\boldsymbol{\mu}_h^* - \boldsymbol{\mu}_{h,k-1})V_{h+1}^* \right\|_{\boldsymbol{H}_{h,k-1}}$$

$$+ \max\left\{ \left\| (\boldsymbol{\mu}_h^* - \boldsymbol{\mu}_{h,k-1})(\overline{\boldsymbol{V}}_{h+1}^k - V_{h+1}^*) \right\|_{\boldsymbol{H}_{h,k-1}}, \left\| (\boldsymbol{\mu}_h^* - \boldsymbol{\mu}_{h,k-1})(\underline{\boldsymbol{V}}_{h+1}^k - V_{h+1}^*) \right\|_{\boldsymbol{H}_{h,k-1}} \right\}.$$

we would analyze the two terms in the RHS separately to proceed with the proof.

**For the first term** Since $V_{h+1}^*$ is a deterministic function, we apply the first item in Lemma F.1 to bound it. In the following, we specify the parameters defined therein. First, we have $C_0 = \mathcal{H}$ and $\mathcal{A}_{h,k} = \left\{\sigma_{h,k}^2 \geqslant (\mathbb{V}_h V_{h+1}^*)(s_{h,k}, a_{h,k})\right\}$ is $\mathcal{F}_{h,k}$-measurable. By Lemma D.5, on the event $\mathcal{B}_0 \bigcap \mathcal{B}_{h+1}$, we have for all $k \in [K]$, $\left|\left[\mathbb{V}_h V_{h+1}^* - \widehat{\mathbb{V}}_h \overline{V}_{h+1}^k\right](s_{h,k}, a_{h,k})\right| \leqslant U_{h,k}$ with $U_{h,k}$ defined in equation A.9. Hence, $\sigma_{h,k}^2 \geqslant [\widehat{\mathbb{V}}_h \widehat{V}_{h+1}^k](s_{h,k}, a_{h,k}) + U_{h,k} \geqslant [\mathbb{V}_h V_{h+1}^*](s_{h,k}, a_{h,k})$ for all $k \in [K]$, implying $\bigcap_{k \in [K]} \mathcal{A}_{h,k}$ holds under $\mathcal{B}_0 \bigcap \mathcal{B}_{h+1}$ and $C_\sigma = 1$. By Lemma F.1, with probability at least $1 - \frac{\delta}{H}$, $\left\|(\boldsymbol{\mu}_h^* - \boldsymbol{\mu}_{h,k-1}) V_{h+1}^*\right\|_{\boldsymbol{H}_{h,k-1}} \leqslant \beta_1$ for all $k \in [K]$ with $\beta_1$ defined in the following. Finally, we simplify $\beta_1$ as

$$\beta_1 := 8\sqrt{d \log\left(1 + \frac{K}{\sigma_{\min}^2 d\lambda}\right) \log \frac{4HK^2}{\delta}} + \frac{8}{d^{2.5}H} \log \frac{4HK^2}{\delta} + \sqrt{d\lambda}\mathcal{H}$$

$$\leqslant 8\sqrt{d}\iota_1 + \frac{8\iota_1}{d^{2.5}H} + \sqrt{d\lambda}\mathcal{H} \leqslant 16\sqrt{d}\iota_1 + \sqrt{d\lambda}\mathcal{H}.$$

**For the second term** Since both $\overline{V}_{h+1}^k - V_{h+1}^*$ and $\underline{V}_{h+1}^k - V_{h+1}^*$ are $\mathcal{F}_{H,k-1}$-measurable random functions, we apply the second item in Lemma F.1 to analyze the second term. In the following, we specify the parameters defined therein. First, $C_0 = \mathcal{H}$ and $\mathcal{A}_{h,k} = \left\{\sigma_{h,k}^2 \geqslant d^3 H \cdot E_{h,k}\right\}$ is $\mathcal{F}_{h,k}$-measurable. By Lemma D.6, on the event $\mathcal{B}_0 \bigcap \mathcal{B}_R \bigcap \mathcal{B}_{h+1}$, we have simultaneously $\left[\mathbb{V}_h(\overline{V}_{h+1}^k - V_{h+1}^*)\right](s_{h,j}, a_{h,j}) \leqslant E_{h,j}$ and $\left[\mathbb{V}_h(\underline{V}_{h+1}^k - V_{h+1}^*)\right](s_{h,j}, a_{h,j}) \leqslant E_{h,j}$ for all $j \leqslant k \leqslant K$ with $E_{h,j}$ defined in equation A.7. As a result, for all $j \leqslant k$,

$$\sigma_{h,j}^2 \geqslant d^3 H \cdot E_{h,j} \geqslant d^3 H \cdot \max\left\{\left[\mathbb{V}_h(\overline{V}_{h+1}^k - V_{h+1}^*)\right](s_{h,j}, a_{h,j}), \left[\mathbb{V}_h(\underline{V}_{h+1}^k - V_{h+1}^*)\right](s_{h,j}, a_{h,j})\right\}.$$

It implies $C_\sigma = \frac{1}{\sqrt{d^3 H}}$ and for any $j \in [k]$,

$$\mathcal{A}_{h,j} \subseteq \left\{\sigma_{h,j}^2 \geqslant C_\sigma^{-2} \max\left\{\left[\mathbb{V}_h(\overline{V}_{h+1}^k - V_{h+1}^*)\right](s_{h,j}, a_{h,j}), \left[\mathbb{V}_h(\underline{V}_{h+1}^k - V_{h+1}^*)\right](s_{h,j}, a_{h,j})\right\}\right\}.$$

Finally, with by Lemma G.8 and G.7, the covering entropy for $\varepsilon_0 = \min\left\{\frac{\sigma_{\min}}{\sqrt{d^3 H}}, \frac{\lambda \mathcal{H}\sqrt{d}}{K}\sigma_{\min}^2\right\}$ and the function class to which $\overline{V}_{h+1}^k - V_{h+1}^*$ and $\underline{V}_{h+1}^k - V_{h+1}^*$ belong is

$$\log N_0 = |\mathcal{N}(\mathcal{V}^\pm, \varepsilon_0)| \leqslant \left[d \log\left(1 + \frac{4L}{\varepsilon_0}\right) + d^2 \log\left(1 + \frac{8\sqrt{d}B^2}{\lambda\varepsilon_0^2}\right)\right] \cdot dH \log_2\left(1 + \frac{K}{\lambda\sigma_{\min}^2}\right)$$

$$= \mathcal{O}(d^3 H \iota_1^2)$$

By Lemma F.1, with probability at least $1 - \frac{\delta}{H}$,

$$\max\left\{\left\|(\boldsymbol{\mu}_h^* - \boldsymbol{\mu}_{h,k-1})(\overline{\boldsymbol{V}}_{h+1}^k - \boldsymbol{V}_{h+1}^*)\right\|_{\boldsymbol{H}_{h,k-1}}, \left\|(\boldsymbol{\mu}_h^* - \boldsymbol{\mu}_{h,k-1})(\underline{\boldsymbol{V}}_{h+1}^k - \boldsymbol{V}_{h+1}^*)\right\|_{\boldsymbol{H}_{h,k-1}}\right\} \leqslant \beta_2$$

for all $k \in [K]$ with $\beta_2$ defined in the following. Finally, we simplify $\beta_2$ as

$$\beta_2 = \frac{8}{\sqrt{d^3 H}}\sqrt{d \log\left(1 + \frac{K}{\sigma_{\min}^2 d\lambda}\right) \log \frac{4N_0 HK^2}{\delta}} + \frac{8}{d^{2.5}H} \log \frac{4N_0 HK^2}{\delta} + \sqrt{d\lambda}\mathcal{H}$$

$$\leqslant 8\sqrt{\frac{\iota_1}{d^2 H} \cdot (\iota_1 + \mathcal{O}(d^3 H \iota_1^2))} + \frac{8}{d^{2.5}H}\left(\iota_1 + \mathcal{O}(d^3 H \iota_1^2)\right) + \sqrt{d\lambda}\mathcal{H}$$

$$= \mathcal{O}\left(\sqrt{d}\iota_1^{1.5} + \iota_1 + \sqrt{d}\iota_1^2 + \sqrt{d\lambda}\mathcal{H}\right) = \mathcal{O}\left(\sqrt{d}\iota_1^2 + \sqrt{d\lambda}\mathcal{H}\right).$$

**Putting pieces together** we have shown that given $\mathcal{B}_{h+1} \bigcap \mathcal{B}_0$ is true, with probability at least $1 - 2\delta$, for all $h \in [H]$ and $k \in [K]$,

$$\max\left\{\left\|(\boldsymbol{\mu}_h^* - \boldsymbol{\mu}_{h,k-1})\overline{\boldsymbol{V}}_{h+1}^k\right\|_{\boldsymbol{H}_{h,k-1}}, \left\|(\boldsymbol{\mu}_h^* - \boldsymbol{\mu}_{h,k-1})\underline{\boldsymbol{V}}_{h+1}^k\right\|_{\boldsymbol{H}_{h,k-1}}\right\} \leqslant \beta_1 + \beta_2 = \mathcal{O}\left(\sqrt{d}\iota_1^2 + \sqrt{d\lambda}\mathcal{H}\right).$$

Therefore, $\mathcal{B}_V := \mathcal{B}_1$ holds. $\qquad\square$

### F.9 Proof of Lemma D.8

*Proof of Lemma D.8.* For a given $k$, let $k_{\text{last}}$ denote the latest update episode before episode $k$, that is $k_{\text{last}} \leqslant k < k_{\text{last}} + 1$. By Lemma G.6, due to $\boldsymbol{H}_{h,k-1} \geq \boldsymbol{H}_{h,k_{\text{last}}-1}$ and $\det(\boldsymbol{H}_{h,k-1}) \leqslant 2\det(\boldsymbol{H}_{h,k_{\text{last}}-1})$, it follows that for any $\boldsymbol{x} \in \mathbb{R}^d$,

$$\|\boldsymbol{x}\|_{\boldsymbol{H}_{h,k_{\text{last}}-1}^{-1}} \leqslant 2\|\boldsymbol{x}\|_{\boldsymbol{H}_{h,k-1}^{-1}}. \tag{F.5}$$

By definition, $\overline{Q}_h^k(\cdot,\cdot) \leqslant \langle\boldsymbol{\phi}(\cdot,\cdot), \boldsymbol{\theta}_{h,k_{\text{last}}-1} + \boldsymbol{\mu}_{h,k_{\text{last}}-1}\overline{\boldsymbol{V}}_{h+1}^{k_{\text{last}}}\rangle + \beta\|\boldsymbol{\phi}(\cdot,\cdot)\|_{\boldsymbol{H}_{h,k_{\text{last}}-1}^{-1}}$ and $Q_h^{\pi_k}(s,a) = \langle\boldsymbol{\phi}(s,a), \boldsymbol{\theta}_h^* + \boldsymbol{\mu}_h^*\boldsymbol{V}_{h+1}^{\pi_k}\rangle$. Using $a_{h,k} = \pi_h^k(s_{h,k}) = \text{argmax}_{a \in \mathcal{A}} \overline{Q}_h^k(s_{h,k}, a)$, we then have

$$(\overline{V}_h^k - V_h^{\pi_k})(s_{h,k}) \leqslant (\overline{Q}_h^k - Q_h^{\pi_k})(s_{h,k}, a_{h,k})$$

$$\leqslant \langle\boldsymbol{\phi}_{h,k}, \boldsymbol{\theta}_{h,k_{\text{last}}-1} + \boldsymbol{\mu}_{h,k_{\text{last}}-1}\overline{\boldsymbol{V}}_{h+1}^{k_{\text{last}}} - (\boldsymbol{\theta}_h^* + \boldsymbol{\mu}_h^*\boldsymbol{V}_{h+1}^{\pi_k})\rangle + \beta\|\boldsymbol{\phi}(s_{h,k}, a_{h,k})\|_{\boldsymbol{H}_{h,k_{\text{last}}-1}^{-1}}$$

$$\overset{(a)}{\leqslant} \langle\boldsymbol{\phi}_{h,k}, (\boldsymbol{\theta}_{h,k_{\text{last}}-1} - \boldsymbol{\theta}_h^*) + (\boldsymbol{\mu}_{h,k_{\text{last}}-1} - \boldsymbol{\mu}_h^*)\overline{\boldsymbol{V}}_{h+1}^{k_{\text{last}}}\rangle + \langle\boldsymbol{\phi}_{h,k}, \boldsymbol{\mu}_h^*(\overline{\boldsymbol{V}}_{h+1}^{k_{\text{last}}} - \boldsymbol{V}_{h+1}^{\pi_k})\rangle + 2\beta\|\boldsymbol{\phi}_{h,k}\|_{\boldsymbol{H}_{h,k-1}^{-1}}$$

$$\overset{(b)}{\leqslant} 4\beta\|\boldsymbol{\phi}_{h,k}\|_{\boldsymbol{H}_{h,k-1}^{-1}} + \langle\boldsymbol{\phi}_{h,k}, \boldsymbol{\mu}_h^*(\overline{\boldsymbol{V}}_{h+1}^k - \boldsymbol{V}_{h+1}^{\pi_k})\rangle$$

$$\overset{(c)}{=} 4\beta\|\boldsymbol{\phi}_{h,k}\|_{\boldsymbol{H}_{h,k-1}^{-1}} + \mathbb{P}_h(\overline{V}_{h+1}^k - V_{h+1}^{\pi_k})(s_{h,k}, a_{h,k})$$

$$\overset{(d)}{=} 4\beta\|\boldsymbol{\phi}_{h,k}\|_{\boldsymbol{H}_{h,k-1}^{-1}} + (\overline{V}_{h+1}^k - V_{h+1}^{\pi_k})(s_{h+1,k}) + X_{h,k}.$$

Here $(a)$ uses equation F.5, $(b)$ uses

$$|\langle\boldsymbol{\phi}_{h,k}, (\boldsymbol{\theta}_{h,k_{\text{last}}-1} - \boldsymbol{\theta}_h^*) + (\boldsymbol{\mu}_{h,k_{\text{last}}-1} - \boldsymbol{\mu}_h^*)\overline{\boldsymbol{V}}_{h+1}^{k_{\text{last}}}\rangle|$$

$$\leqslant \|\boldsymbol{\phi}_{h,k}\|_{\boldsymbol{H}_{h,k_{\text{last}}-1}^{-1}}\|(\boldsymbol{\theta}_{h,k_{\text{last}}-1} - \boldsymbol{\theta}_h^*) + (\boldsymbol{\mu}_{h,k_{\text{last}}-1} - \boldsymbol{\mu}_h^*)\overline{\boldsymbol{V}}_{h+1}^{k_{\text{last}}}\|_{\boldsymbol{H}_{h,k_{\text{last}}-1}}$$

$$\leqslant \beta\|\boldsymbol{\phi}_{h,k}\|_{\boldsymbol{H}_{h,k_{\text{last}}-1}^{-1}} \leqslant 2\beta\|\boldsymbol{\phi}_{h,k}\|_{\boldsymbol{H}_{h,k-1}^{-1}}$$

on $\mathcal{B}_R \bigcap \mathcal{B}_V$, $(c)$ uses $\langle\boldsymbol{\phi}_{h,k}, \boldsymbol{\mu}_h^*(\overline{\boldsymbol{V}}_{h+1}^{k_{\text{last}}} - V_{h+1}^{\pi_k})\rangle = \mathbb{P}_h(\overline{\boldsymbol{V}}_{h+1}^{k_{\text{last}}} - \boldsymbol{V}_{h+1}^{\pi_k})(s_{h,k}, a_{h,k}) = \mathbb{P}_h(\overline{\boldsymbol{V}}_{h+1}^k - \boldsymbol{V}_{h+1}^{\pi_k})(s_{h,k}, a_{h,k})$, and $(d)$ uses the notation

$$X_{h,k} := \mathbb{P}_h(\overline{V}_{h+1}^k - V_{h+1}^{\pi_k})(s_{h,k}, a_{h,k}) - (\overline{V}_{h+1}^k - V_{h+1}^{\pi_k})(s_{h+1,k}).$$

The last inequality implies

$$(\overline{V}_h^k - V_h^{\pi_k})(s_{h,k}) \leqslant (\overline{V}_{h+1}^k - V_{h+1}^{\pi_k})(s_{h+1,k}) + X_{h,k} + 4\beta\|\boldsymbol{\phi}_{h,k}\|_{\boldsymbol{H}_{h,k-1}^{-1}}.$$

Iterating the above inequality over $h$ and using $\overline{V}_{H+1}^k(\cdot) = V_{H+1}^{\pi_k}(\cdot) = 0$, we have

$$(\overline{V}_h^k - V_h^{\pi_k})(s_{h,k}) \leqslant \sum_{i=h}^H \left[X_{i,k} + 4\beta\|\boldsymbol{\phi}_{i,k}\|_{\boldsymbol{H}_{i,k-1}^{-1}}\right]. \tag{F.6}$$

Therefore, setting $h = 1$ and summing equation F.6 over $k \in [K]$, we have

$$\sum_{k=1}^K (\overline{V}_1^k - V_1^{\pi_k})(s_{1,k}) \leqslant \sum_{k=1}^K \sum_{h=1}^H \left[X_{h,k} + 4\beta\|\boldsymbol{\phi}_{h,k}\|_{\boldsymbol{H}_{h,k-1}^{-1}}\right]. \tag{F.7}$$

We then need to analyze $\sum_{k=1}^{K}\sum_{h=1}^{H}X_{h,k}$. Since $s_{h+1,k}$ is $\mathcal{F}_{h+1,k}$-measurable, $\pi_k = \{\pi_h^k\}_{h\in[H]}, \overline{V}_{h+1}^k$ is $\mathcal{F}_{H,k-1}$-measurable, we have $X_{h,k}$ is $\mathcal{F}_{h+1,k}$-measurable. We also have $\mathbb{E}[X_{h,k}|\mathcal{F}_{h,k}] = 0$, $|X_{h,k}| \leqslant 2\mathcal{H}$ and

$$\mathbb{E}[X_{h,k}^2|\mathcal{F}_{h,k}] \leqslant \mathbb{E}[(\overline{V}_{h+1}^k - V_{h+1}^{\pi_k})^2(s_{h+1,k})|\mathcal{F}_{h,k}] \overset{(a)}{\leqslant} \mathcal{H}\mathbb{E}[|\overline{V}_{h+1}^k - V_{h+1}^{\pi_k}|(s_{h+1,k})|\mathcal{F}_{h,k}]$$

$$\overset{(b)}{=} \mathcal{H}\mathbb{E}[(\overline{V}_{h+1}^k - V_{h+1}^{\pi_k})(s_{h+1,k})|\mathcal{F}_{h,k}] = \mathcal{H}\mathbb{P}_h(\overline{V}_{h+1}^k - V_{h+1}^{\pi_k})(s_{h,k}, a_{h,k})$$

where $(a)$ uses $|\overline{V}_{h+1}^k - V_{h+1}^{\pi_k}|(\cdot) \leqslant \mathcal{H}$ and $(b)$ uses the optimism in Lemma D.4. By the variance-aware Freedman inequality in Lemma G.2, with probability at least $1 - \frac{\delta}{2}$, it follows that

$$\left|\sum_{k=1}^{K}\sum_{h=1}^{H}X_{h,k}\right| \leqslant 3\sqrt{\iota} \cdot \sqrt{\mathcal{H} \cdot \sum_{k=1}^{K}\sum_{h=1}^{H}\mathbb{P}_h(\overline{V}_{h+1}^k - V_{h+1}^{\pi_k})(s_{h,k}, a_{h,k}) + 10\mathcal{H} \cdot \iota} \tag{F.8}$$

where $\iota = \log\frac{4\lceil\log_2 HK\rceil}{\delta}$. On the other hand, it follows that

$$\sum_{k=1}^{K}\sum_{h=1}^{H}\mathbb{P}_h(\overline{V}_{h+1}^k - V_{h+1}^{\pi_k})(s_{h,k}, a_{h,k}) = \sum_{k=1}^{K}\sum_{h=2}^{H}(\overline{V}_h^k - V_h^{\pi_k})(s_{h,k}) + \sum_{k=1}^{K}\sum_{h=1}^{H}X_{h,k}$$

$$\overset{(a)}{\leqslant} \sum_{k=1}^{K}\sum_{h=2}^{H}\sum_{i=h}^{H}\left[X_{i,k} + 4\beta\|\phi_{i,k}\|_{\boldsymbol{H}_{i,k-1}^{-1}}\right] + \sum_{k=1}^{K}\sum_{h=1}^{H}X_{h,k}$$

$$= \sum_{k=1}^{K}\sum_{h=2}^{H}(H - h + 1)\left[X_{h,k} + 4\beta\|\phi_{h,k}\|_{\boldsymbol{H}_{h,k-1}^{-1}}\right] + \sum_{k=1}^{K}\sum_{h=1}^{H}X_{h,k}$$

$$\overset{(b)}{\leqslant} 4H\beta\sum_{k=1}^{K}\sum_{h=2}^{H}\|\phi_{h,k}\|_{\boldsymbol{H}_{h,k-1}^{-1}} + \sum_{k=1}^{K}\sum_{h=1}^{H}X_{h,k}b_h$$

where $(a)$ uses equation F.6 and $(b)$ uses the notation $b_h = 1$ if $h = 1$; otherwise $= H - h + 2$ for $2 \leqslant h \leqslant H$. Clearly, we have $|b_h| \leqslant H$ for all $h \in [H]$. By the variance-aware Freedman inequality in Lemma G.2, with probability at least $1 - \frac{\delta}{2}$, it follows that

$$\left|\sum_{k=1}^{K}\sum_{h=1}^{H}X_{h,k}b_h\right| \leqslant 3H\sqrt{\iota} \cdot \sqrt{\mathcal{H} \cdot \sum_{k=1}^{K}\sum_{h=1}^{H}\mathbb{P}_h(\overline{V}_{h+1}^k - V_{h+1}^{\pi_k})(s_{h,k}, a_{h,k}) + 10H\mathcal{H} \cdot \iota}.$$

As a result, we have

$$\sum_{k=1}^{K}\sum_{h=1}^{H}\mathbb{P}_h(\overline{V}_{h+1}^k - V_{h+1}^{\pi_k})(s_{h,k}, a_{h,k}) \leqslant 3H\sqrt{\iota} \cdot \sqrt{\mathcal{H} \cdot \sum_{k=1}^{K}\sum_{h=1}^{H}\mathbb{P}_h(\overline{V}_{h+1}^k - V_{h+1}^{\pi_k})(s_{h,k}, a_{h,k})}$$

$$+ 4H\beta\sum_{k=1}^{K}\sum_{h=2}^{H}\|\phi_{h,k}\|_{\boldsymbol{H}_{h,k-1}^{-1}} + 10H\mathcal{H}\iota.$$

Using the inequality that $x \leqslant 2(a^2 + b^2)$ for any $x \leqslant |a|\sqrt{x} + b^2$, we have

$$\sum_{k=1}^{K}\sum_{h=1}^{H}\mathbb{P}_h(\overline{V}_{h+1}^k - V_{h+1}^{\pi_k})(s_{h,k}, a_{h,k}) \leqslant 8H\beta\sum_{k=1}^{K}\sum_{h=1}^{H}\|\phi_{h,k}\|_{\boldsymbol{H}_{h,k-1}^{-1}} + 38H^2\mathcal{H}\iota. \tag{F.9}$$

Putting pieces together, we have

$$\sum_{k=1}^{K}(\overline{V}_1^k - V_1^{\pi_k})(s_{1,k}) \overset{equation\ F.7}{\leqslant} \sum_{k=1}^{K}\sum_{h=1}^{H}\left[X_{h,k} + 4\beta\|\phi_{h,k}\|_{\boldsymbol{H}_{h,k-1}^{-1}}\right]$$

$$\overset{\text{equation } F.8}{\leqslant} 4\beta \sum_{k=1}^{K}\sum_{h=1}^{H}\|\boldsymbol{\phi}_{h,k}\|_{\boldsymbol{H}_{h,k-1}^{-1}} + 3\sqrt{\iota} \cdot \sqrt{\mathcal{H} \cdot \sum_{k=1}^{K}\sum_{h=1}^{H}\mathbb{P}_h(\overline{V}_{h+1}^k - V_{h+1}^{\pi_k})(s_{h,k},a_{h,k}) + 10\mathcal{H} \cdot \iota}$$

$$\overset{\text{equation } F.9}{\leqslant} 4\beta \sum_{k=1}^{K}\sum_{h=1}^{H}\|\boldsymbol{\phi}_{h,k}\|_{\boldsymbol{H}_{h,k-1}^{-1}} + 3\sqrt{\iota} \cdot \sqrt{\mathcal{H} \cdot \left[8H\beta \sum_{k=1}^{K}\sum_{h=1}^{H}\|\boldsymbol{\phi}_{h,k}\|_{\boldsymbol{H}_{h,k-1}^{-1}} + 38H^2\mathcal{H}\iota\right] + 10\mathcal{H} \cdot \iota}$$

$$\leqslant 6\beta \sum_{k=1}^{K}\sum_{h=1}^{H}\|\boldsymbol{\phi}_{h,k}\|_{\boldsymbol{H}_{h,k-1}^{-1}} + 38H\mathcal{H}\iota$$

where the last inequality uses $\sqrt{a+b} \leqslant \sqrt{a} + \sqrt{b}$ and $2\sqrt{ab} \leqslant a + b$ for non-negative numbers $a,b \geqslant 0$. $\qquad\square$

### F.10 Proof of Lemma D.9

*Proof of Lemma D.9.* The proof main idea is similar to that in Lemma D.8. For a given $k$, let $k_{\text{last}}$ denote the latest update episode before episode $k$, that is $k_{\text{last}} \leqslant k < k_{\text{last}}+1$. By definition, $\overline{Q}_h^k(\cdot,\cdot) \leqslant \langle\boldsymbol{\phi}(\cdot,\cdot), \boldsymbol{\theta}_{h,k_{\text{last}}-1} + \boldsymbol{\mu}_{h,k_{\text{last}}-1}\overline{\boldsymbol{V}}_{h+1}^{k_{\text{last}}}\rangle + \beta\|\boldsymbol{\phi}(\cdot,\cdot)\|_{\boldsymbol{H}_{h,k_{\text{last}}-1}^{-1}}$ and $\underline{Q}_h^k(\cdot,\cdot) \geqslant \langle\boldsymbol{\phi}(\cdot,\cdot), \boldsymbol{\theta}_{h,k_{\text{last}}-1} + \boldsymbol{\mu}_{h,k_{\text{last}}-1}\underline{\boldsymbol{V}}_{h+1}^{k_{\text{last}}}\rangle - \beta\|\boldsymbol{\phi}(\cdot,\cdot)\|_{\boldsymbol{H}_{h,k_{\text{last}}-1}^{-1}}$. Using

$$a_{h,k} = \pi_h^k(s_{h,k}) = \underset{a\in\mathcal{A}}{\operatorname{argmax}}\,\overline{Q}_h^k(s_{h,k},a),$$

we then have

$$(\overline{V}_h^k - \underline{V}_h^k)(s_{h,k}) \leqslant (\overline{Q}_h^k - \underline{Q}_h^k)(s_{h,k},a_{h,k})$$

$$\leqslant \langle\boldsymbol{\phi}_{h,k}, \boldsymbol{\mu}_{h,k_{\text{last}}-1}(\overline{\boldsymbol{V}}_{h+1}^{k_{\text{last}}} - \underline{\boldsymbol{V}}_{h+1}^{k_{\text{last}}})\rangle + 2\beta\|\boldsymbol{\phi}_{h,k}\|_{\boldsymbol{H}_{h,k_{\text{last}}-1}^{-1}}$$

$$\overset{(a)}{\leqslant} \langle\boldsymbol{\phi}_{h,k}, (\boldsymbol{\mu}_{h,k-1} - \boldsymbol{\mu}_h^*)(\overline{\boldsymbol{V}}_{h+1}^{k_{\text{last}}} - \underline{\boldsymbol{V}}_{h+1}^{k_{\text{last}}})\rangle + \langle\boldsymbol{\phi}_{h,k}, \boldsymbol{\mu}_h^*(\overline{\boldsymbol{V}}_{h+1}^{k_{\text{last}}} - \underline{\boldsymbol{V}}_{h+1}^{k_{\text{last}}})\rangle + 4\beta\|\boldsymbol{\phi}_{h,k}\|_{\boldsymbol{H}_{h,k-1}^{-1}}$$

$$\overset{(b)}{\leqslant} 6\beta\|\boldsymbol{\phi}_{h,k}\|_{\boldsymbol{H}_{h,k-1}^{-1}} + \langle\boldsymbol{\phi}_{h,k}, \boldsymbol{\mu}_h^*(\overline{\boldsymbol{V}}_{h+1}^{k_{\text{last}}} - \underline{\boldsymbol{V}}_{h+1}^{k_{\text{last}}})\rangle$$

$$\overset{(c)}{=} 6\beta\|\boldsymbol{\phi}_{h,k}\|_{\boldsymbol{H}_{h,k-1}^{-1}} + \mathbb{P}_h(\overline{V}_{h+1}^k - \underline{V}_{h+1}^k)(s_{h,k},a_{h,k})$$

$$\overset{(d)}{=} 6\beta\|\boldsymbol{\phi}_{h,k}\|_{\boldsymbol{H}_{h,k-1}^{-1}} + (\overline{V}_{h+1}^k - \underline{V}_{h+1}^k)(s_{h+1,k}) + X_{h,k}.$$

Here $(a)$ uses equation F.5, $(b)$ uses

$$|\langle\boldsymbol{\phi}_{h,k}, (\boldsymbol{\mu}_{h,k_{\text{last}}-1} - \boldsymbol{\mu}_h^*)(\overline{\boldsymbol{V}}_{h+1}^{k_{\text{last}}} - \underline{\boldsymbol{V}}_{h+1}^{k_{\text{last}}})\rangle|$$

$$\leqslant \|\boldsymbol{\phi}_{h,k}\|_{\boldsymbol{H}_{h,k_{\text{last}}-1}^{-1}}\|(\boldsymbol{\mu}_{h,k_{\text{last}}-1} - \boldsymbol{\mu}_h^*)(\overline{\boldsymbol{V}}_{h+1}^{k_{\text{last}}} - \underline{\boldsymbol{V}}_{h+1}^{k_{\text{last}}})\|_{\boldsymbol{H}_{h,k_{\text{last}}-1}}$$

$$\leqslant 2\beta\|\boldsymbol{\phi}_{h,k}\|_{\boldsymbol{H}_{h,k_{\text{last}}-1}^{-1}} \leqslant 2\beta\|\boldsymbol{\phi}_{h,k}\|_{\boldsymbol{H}_{h,k-1}^{-1}}$$

on $\mathcal{B}_V \bigcap \mathcal{B}_R$, $(c)$ uses $\langle\boldsymbol{\phi}_{h,k}, \boldsymbol{\mu}_h^*(\overline{\boldsymbol{V}}_{h+1}^{k_{\text{last}}} - \underline{\boldsymbol{V}}_{h+1}^{k_{\text{last}}})\rangle = \mathbb{P}_h(\overline{V}_{h+1}^{k_{\text{last}}} - \underline{V}_{h+1}^{k_{\text{last}}})(s_{h,k},a_{h,k}) = \mathbb{P}_h(\overline{V}_{h+1}^k - \underline{V}_{h+1}^k)(s_{h,k},a_{h,k})$, and $(d)$ uses the notation

$$X_{h,k} := \mathbb{P}_h(\overline{V}_{h+1}^k - \underline{V}_{h+1}^k)(s_{h,k},a_{h,k}) - (\overline{V}_{h+1}^k - \underline{V}_{h+1}^k)(s_{h+1,k}). \tag{F.10}$$

The last inequality implies

$$(\overline{V}_h^k - \underline{V}_h^k)(s_{h,k}) \leqslant (\overline{V}_{h+1}^k - \underline{V}_{h+1}^k)(s_{h+1,k}) + X_{h,k} + 6\beta\|\boldsymbol{\phi}_{h,k}\|_{\boldsymbol{H}_{h,k-1}^{-1}}.$$

Iterating the above inequality over $h$ and using $\overline{V}_{H+1}^k(\cdot) = \underline{V}_{H+1}^k(\cdot) = 0$, we have

$$(\overline{V}_h^k - \underline{V}_h^k)(s_{h,k}) \leqslant \sum_{i=h}^{H}\left[X_{i,k} + 6\beta\|\boldsymbol{\phi}_{i,k}\|_{\boldsymbol{H}_{i,k-1}^{-1}}\right]. \tag{F.11}$$

Using the last inequality, it follows that

$$\sum_{k=1}^{K}\sum_{h=1}^{H}\mathbb{P}_h(\overline{V}_{h+1}^k - \underline{V}_{h+1}^k)(s_{h,k}, a_{h,k}) = \sum_{k=1}^{K}\sum_{h=2}^{H}(\overline{V}_h^k - \underline{V}_h^k)(s_{h,k}) + \sum_{k=1}^{K}\sum_{h=1}^{H}X_{h,k}$$

$$\overset{(a)}{\leqslant} \sum_{k=1}^{K}\sum_{h=2}^{H}\sum_{i=h}^{H}\left[X_{i,k} + 8\beta\|\phi_{i,k}\|_{\boldsymbol{H}_{i,k-1}^{-1}}\right] + \sum_{k=1}^{K}\sum_{h=1}^{H}X_{h,k}$$

$$= \sum_{k=1}^{K}\sum_{h=2}^{H}(H - h + 1)\left[X_{h,k} + 6\beta\|\phi_{h,k}\|_{\boldsymbol{H}_{h,k-1}^{-1}}\right] + \sum_{k=1}^{K}\sum_{h=1}^{H}X_{h,k}$$

$$\overset{(b)}{\leqslant} 6H\beta\sum_{k=1}^{K}\sum_{h=2}^{H}\|\phi_{h,k}\|_{\boldsymbol{H}_{h,k-1}^{-1}} + \sum_{k=1}^{K}\sum_{h=1}^{H}X_{h,k}b_h \tag{F.12}$$

where $(a)$ uses equation F.6 and $(b)$ uses the notation $b_h = 1$ if $h = 1$; otherwise $= H - h + 2$ for $2 \leqslant h \leqslant H$. Clearly, we have $|b_h| \leqslant H$ for all $h \in [H]$.

We then need to analyze $\sum_{k=1}^{K}\sum_{h=1}^{H}X_{h,k}b_h$ with $X_{h,k}$'s defined in equation F.10. Since $s_{h+1,k}$ is $\mathcal{F}_{h+1,k}$-measurable, $\overline{V}_{h+1}^k, \underline{V}_{h+1}^k$ is $\mathcal{F}_{H,k-1}$-measurable, we have $X_{h,k}$ is $\mathcal{F}_{h+1,k}$-measurable. We also have $\mathbb{E}[X_{h,k}|\mathcal{F}_{h,k}] = 0$, $|X_{h,k}| \leqslant 2\mathcal{H}$ and

$$\mathbb{E}[X_{h,k}^2|\mathcal{F}_{h,k}] \leqslant \mathbb{E}[(\overline{V}_{h+1}^k - \underline{V}_{h+1}^k)^2(s_{h+1,k})|\mathcal{F}_{h,k}]$$

$$\overset{(a)}{\leqslant} \mathcal{H}\mathbb{E}[|\overline{V}_{h+1}^k - \underline{V}_{h+1}^k|(s_{h+1,k})|\mathcal{F}_{h,k}] = \mathcal{H}\mathbb{P}_h(\overline{V}_{h+1}^k - \underline{V}_{h+1}^k)(s_{h,k}, a_{h,k})$$

where $(a)$ uses $|\overline{V}_{h+1}^k - \underline{V}_{h+1}^k|(\cdot) \leqslant \mathcal{H}$. By the variance-aware Freedman inequality in Lemma G.2, with probability at least $1 - \delta$, it follows that

$$\left|\sum_{k=1}^{K}\sum_{h=1}^{H}X_{h,k}b_h\right| \leqslant 3H\sqrt{\iota} \cdot \sqrt{\mathcal{H} \cdot \sum_{k=1}^{K}\sum_{h=1}^{H}\mathbb{P}_h(\overline{V}_{h+1}^k - \underline{V}_{h+1}^k)(s_{h,k}, a_{h,k})} + 10H\mathcal{H} \cdot \iota \tag{F.13}$$

where $\iota = \log\frac{4\lceil\log_2 HK\rceil}{\delta}$. As a result, plugging equation F.13 into equation F.12, we have

$$\sum_{k=1}^{K}\sum_{h=1}^{H}\mathbb{P}_h(\overline{V}_{h+1}^k - \underline{V}_{h+1}^k)(s_{h,k}, a_{h,k}) \leqslant 3H\sqrt{\iota} \cdot \sqrt{\mathcal{H} \cdot \sum_{k=1}^{K}\sum_{h=1}^{H}\mathbb{P}_h(\overline{V}_{h+1}^k - \underline{V}_{h+1}^k)(s_{h,k}, a_{h,k})}$$

$$+ 6H\beta\sum_{k=1}^{K}\sum_{h=2}^{H}\|\phi_{h,k}\|_{\boldsymbol{H}_{h,k-1}^{-1}} + 10H\mathcal{H}\iota.$$

Using the inequality that $x \leqslant 2(a^2 + b^2)$ for any $x \leqslant |a|\sqrt{x} + b^2$, we have

$$\sum_{k=1}^{K}\sum_{h=1}^{H}\mathbb{P}_h(\overline{V}_{h+1}^k - \underline{V}_{h+1}^k)(s_{h,k}, a_{h,k}) \leqslant 12H\beta\sum_{k=1}^{K}\sum_{h=1}^{H}\|\phi_{h,k}\|_{\boldsymbol{H}_{h,k-1}^{-1}} + 38H^2\mathcal{H}\iota.$$

$\square$

## F.11  Proof of Lemma D.10

*Proof of Lemma D.10.* Recall that $b_{h,k} = \max\{\|\phi_{h,k}\|_{\boldsymbol{H}_{h,k-1}^{-1}}, \|\widetilde{\phi}_{h,k}\|_{\widetilde{\boldsymbol{H}}_{h,k-1}^{-1}}\}, w_{h,k} = \sigma_{h,k}^{-1}\|\phi_{h,k}\|_{\boldsymbol{H}_{h,k-1}^{-1}}$ and $\widetilde{w}_{h,k} = \sigma_{h,k}^{-1}\|\widetilde{\phi}_{h,k}\|_{\widetilde{\boldsymbol{H}}_{h,k-1}^{-1}}$. As a result, we have $\sigma_{h,k}^{-1}b_{h,k} = \max\{w_{h,k}, \widetilde{w}_{h,k}\}$. On the other hand,

$$\sigma_{h,k}^2 = \max\left\{\sigma_{\min}^2, d^3H \cdot E_{h,k}, J_{h,k}, c_0^{-2}b_{h,k}^2, \left(\frac{W}{\sqrt{c_1d}} + \mathcal{H}d^{2.5}H\right)b_{h,k}\right\}. \tag{A.5}$$

Based on what value $\sigma_{h,k}$ takes, we compose the full index set $\mathcal{I} := [H] \times [K]$ into three disjoint sets with ties broken arbitrarily:

$$\mathcal{J}_1 = \left\{ (h,k) \subseteq [H] \times [K] : \sigma_{h,k}^2 \in \left\{ \sigma_{\min}^2, d^3 H \cdot E_{h,k}, U_{h,k} \right\} \right\},$$

$$\mathcal{J}_2 = \left\{ (h,k) \subseteq [H] \times [K] : \sigma_{h,k}^2 = c_0^{-2} b_{h,k}^2 \right\},$$

$$\mathcal{J}_3 = \left\{ (h,k) \subseteq [H] \times [K] : \sigma_{h,k}^2 = \left( \frac{W}{\sqrt{c_1 d}} + \mathcal{H} d^{2.5} H \right) b_{h,k} \right\}.$$

For simplicity, we denote $z_{h,k} := \frac{b_{h,k}}{\sigma_{h,k}} = \max\{w_{h,k}, \widetilde{w}_{h,k}\}$. Therefore,

$$\sum_{k=1}^{K} \sum_{h=1}^{H} b_{h,k} = \sum_{(h,k) \in \mathcal{I}} \sigma_{h,k} z_{h,k} = \sum_{i=1}^{3} \sum_{(h,k) \in \mathcal{J}_i} \sigma_{h,k} z_{h,k}. \tag{F.14}$$

Recall that $\kappa = d \log\left(1 + \frac{K}{d\lambda\sigma_{\min}^2}\right)$, we have $\sum_{(h,k) \in \mathcal{I}} z_{h,k}^2 \leqslant 4H\kappa$. This is because

$$\sum_{(h,k) \in \mathcal{I}} z_{h,k}^2 \leqslant \sum_{(h,k) \in \mathcal{I}} \left( w_{h,k}^2 + \widetilde{w}_{h,k}^2 \right) \overset{(a)}{=} \sum_{k=1}^{K} \sum_{h=1}^{H} \min\left\{1, w_{h,k}^2\right\} + \sum_{k=1}^{K} \sum_{h=1}^{H} \min\left\{1, \widetilde{w}_{h,k}^2\right\}$$

$$\overset{(b)}{\leqslant} 4Hd \log\left(1 + \frac{K}{d\lambda\sigma_{\min}^2}\right) = 4H\kappa.$$

where $(a)$ uses $z_{h,k} \leqslant c_0 \leqslant 1$ due to $\sigma_{h,k} \geqslant c_0^{-1} b_{h,k}, c_0 \leqslant 1$ and $(b)$ uses Lemma G.5. We will frequently use the above inequality.

Now, we are ready to analyze the three terms in the RHS of equation F.14 respectively.

- For the first term, it follows that

$$\sum_{(h,k) \in \mathcal{J}_1} \sigma_{h,k} z_{h,k} \leqslant \sqrt{\sum_{(h,k) \in \mathcal{J}_1} \sigma_{h,k}^2} \sqrt{\sum_{(h,k) \in \mathcal{J}_1} z_{h,k}^2}$$

$$\leqslant \sqrt{\sum_{(h,k) \in \mathcal{J}_1} (\sigma_{\min}^2 + d^3 H \cdot E_{h,k} + J_{h,k})} \sqrt{\sum_{(h,k) \in \mathcal{J}_1} z_{h,k}^2}$$

$$\leqslant \sqrt{\sum_{(h,k) \in \mathcal{I}} (\sigma_{\min}^2 + d^3 H \cdot E_{h,k} + J_{h,k})} \sqrt{\sum_{(h,k) \in \mathcal{I}} z_{h,k}^2}$$

$$\leqslant \sqrt{HK\sigma_{\min}^2 + \sum_{(h,k) \in \mathcal{I}} (d^3 H \cdot E_{h,k} + J_{h,k})} \cdot \sqrt{4H\kappa}.$$

We provide a upper bound for $\sum_{k=1}^{K} \sum_{h=1}^{H} E_{h,k}$ in Lemma F.2 whose proof is deferred in Appendix F.11.1.

**Lemma F.2** (Sum of $E_{h,k}$). On the event $\mathcal{B}_0 \bigcap \mathcal{A}_0$,

$$\sum_{k=1}^{K} \sum_{h=1}^{H} E_{h,k} = \mathcal{O}\left( (\beta_0 + H\beta)\mathcal{H} \cdot \sum_{k=1}^{K} \sum_{h=1}^{H} \|\phi_{h,k}\|_{\boldsymbol{H}_{h,k-1}^{-1}} + H^2 \mathcal{H}^2 \log \frac{4\lceil \log_2 HK \rceil}{\delta} \right).$$

where $\mathcal{O}(\cdot)$ hides universal positive constants.

We also provide a upper bound for $\sum_{k=1}^{K} \sum_{h=1}^{H} J_{h,k}$ in Lemma F.3 whose proof is deferred in Appendix F.11.2.

**Lemma F.3** (Sum of $J_{h,k}$). Recall that $J_{h,k} = [\widehat{\mathbb{V}}_h R_h + \widehat{\mathbb{V}}_h \overline{V}_{h+1}^k](s_{h,k}, a_{h,k}) + R_{h,k} + U_{h,k}$ with $R_{h,k}, U_{h,k}$ defined in equation A.8 and equation A.9 respectively. On the event $\mathcal{B}_R \bigcap \mathcal{B}_V \bigcap \mathcal{B}_0 \bigcap \mathcal{A}_0$, with probability at least $1 - 2\delta$,

$$\sum_{k=1}^{K} \sum_{h=1}^{H} J_{h,k} = \mathcal{O}\left( \mathcal{G}^* K + [(\beta_0 + H\beta)\mathcal{H} + \beta_{R^2}] \sum_{k=1}^{K} \sum_{h=1}^{H} b_{h,k} + H^2 \mathcal{H}^2 \log \frac{4\lceil \log_2 HK \rceil}{\delta} + H\sigma_R^2 \log \frac{1}{\delta} \right).$$

where $\mathcal{G}^*$ is defined in equation 3.2 and $\mathcal{O}(\cdot)$ hides universal positive constants.

Putting pieces together and using $\sqrt{a+b+c} \leqslant \sqrt{a} + \sqrt{b} + \sqrt{c}$, we have

$$
\begin{aligned}
\sum_{(h,k)\in\mathcal{J}_1} b_{h,k} = \sum_{(h,k)\in\mathcal{J}_1} \sigma_{h,k} z_{h,k} &= \mathcal{O}\left(\sqrt{H\kappa} \cdot \sqrt{K\left(H\sigma_{\min}^2 + \mathcal{G}^*\right)}\right) \\
&+ \mathcal{O}\left(\sqrt{H\kappa} \cdot \sqrt{H^3 d^3 \mathcal{H}^2 \log \frac{4\lceil\log_2 HK\rceil}{\delta} + H\sigma_R^2 \log\frac{1}{\delta}}\cdot\right) \\
&+ \mathcal{O}\left(\sqrt{H\kappa} \cdot \sqrt{\left[(\beta_0 + H\beta)\mathcal{H}d^3 H + \beta_{R^2}\right] \sum_{(h,k)\in\mathcal{I}} b_{h,k}}\right).
\end{aligned}
\tag{F.15}
$$

- For the second term, due to $\sigma_{h,k} = c_0^{-1} b_{h,k}$, we have $z_{h,k} = b_{h,k}/\sigma_{h,k} = c_0 \leqslant 1$ for all $(h,k) \in \mathcal{J}_2$. Hence,

$$
\begin{aligned}
\sum_{(h,k)\in\mathcal{J}_2} b_{h,k} = \sum_{(h,k)\in\mathcal{J}_2} \sigma_{h,k} z_{h,k} &= \frac{1}{c_0} \sum_{(h,k)\in\mathcal{J}_2} \sigma_{h,k} z_{h,k}^2 \leqslant \frac{\sup_{(h,k)\in\mathcal{I}} \sigma_{h,k}}{c_0} \sum_{(h,k)\in\mathcal{J}_2} z_{h,k}^2 \\
&\leqslant \sup_{(h,k)\in\mathcal{I}} \frac{\max\{\|\boldsymbol{\phi}_{h,k}\|_{\boldsymbol{H}_{h,k-1}^{-1}}, \|\widetilde{\boldsymbol{\phi}}_{h,k}\|_{\widetilde{\boldsymbol{H}}_{h,k-1}^{-1}}\}}{c_0^2} \cdot \sum_{(h,k)\in\mathcal{I}} z_{h,k}^2 \leqslant \frac{4H\kappa}{c_0^2 \sqrt{\lambda}}
\end{aligned}
\tag{F.16}
$$

  where the last inequality uses $\|\boldsymbol{\phi}_{h,k}\|_{\boldsymbol{H}_{h,k-1}^{-1}} \leqslant \frac{1}{\sqrt{\lambda}}\|\boldsymbol{\phi}_{h,k}\| \leqslant \frac{1}{\sqrt{\lambda}}$ and $\|\widetilde{\boldsymbol{\phi}}_{h,k}\|_{\widetilde{\boldsymbol{H}}_{h,k-1}^{-1}} \leqslant \frac{1}{\sqrt{\lambda}}\|\widetilde{\boldsymbol{\phi}}_{h,k}\| \leqslant \frac{1}{\sqrt{\lambda}}$ for any $(h,k) \in \mathcal{I}$.

- For the third term, $\sigma_{h,k}^2 = \left(\frac{W}{\sqrt{c_1 d}} + \mathcal{H}d^{2.5}H\right) b_{h,k}$ and thus $\sigma_{h,k} = \left(\frac{W}{\sqrt{c_1 d}} + \mathcal{H}d^{2.5}H\right) z_{h,k}$. Hence,

$$
\begin{aligned}
\sum_{(h,k)\in\mathcal{J}_3} b_{h,k} = \sum_{(h,k)\in\mathcal{J}_3} \sigma_{h,k} z_{h,k} &= \left(\frac{W}{\sqrt{c_1 d}} + \mathcal{H}d^{2.5}H\right) \sum_{(h,k)\in\mathcal{J}_3} z_{h,k}^2 \\
&\leqslant \left(\frac{W}{\sqrt{c_1 d}} + \mathcal{H}d^{2.5}H\right) \sum_{(h,k)\in\mathcal{I}} z_{h,k}^2 \leqslant 4H\kappa \cdot \left(\frac{W}{\sqrt{c_1 d}} + \mathcal{H}d^{2.5}H\right).
\end{aligned}
\tag{F.17}
$$

Combing equation F.15, equation F.16 and equation F.17, we have

$$
\sum_{(h,k)\in\mathcal{I}} b_{h,k} = \mathcal{O}\left(C + \sqrt{H\kappa}\sqrt{\left[(\beta_0 + H\beta)\mathcal{H}d^3 H + \beta_{R^2}\right] \cdot \sum_{(h,k)\in\mathcal{I}} b_{h,k}}\right)
$$

where

$$
\begin{aligned}
C &= \sqrt{H\kappa} \cdot \sqrt{K\left(H\sigma_{\min}^2 + \mathcal{G}^*\right)} + H\kappa \cdot \left(\frac{W}{\sqrt{c_1 d}} + \frac{1}{c_0^2 \sqrt{\lambda}} + \mathcal{H}d^{2.5}H\right) \\
&+ \sqrt{H\kappa} \cdot \sqrt{H^3 d^3 \mathcal{H}^2 \log \frac{4\lceil\log_2 HK\rceil}{\delta} + H\sigma_R^2 \log\frac{1}{\delta}}.
\end{aligned}
$$

Using the inequality that $x \leqslant 2(a^2 + b^2)$ for any $x \leqslant |a|\sqrt{x} + b^2$, we have

$$
\sum_{(h,k)\in\mathcal{I}} b_{h,k} = \mathcal{O}\left(C + H^2 \mathcal{H}\kappa d^3 (\beta_0 + H\beta) + H\kappa\beta_{R^2}\right).
$$

In the following, we are going to simplify the last inequality. For simplicity, we will use $\widetilde{\mathcal{O}}(\cdot)$ to hide logarithmic factors. Notice that $\kappa = \widetilde{\mathcal{O}}(d)$. By setting $\lambda = \frac{1}{\mathcal{H}^2 + W^2}$, we have $\beta_R = \beta_V = \widetilde{\mathcal{O}}(\sqrt{d})$ and

thus $\beta = \beta_V + \beta_R = \tilde{\mathcal{O}}(\sqrt{d})$. Moreover, $\beta_{R^2} = \tilde{\mathcal{O}}\left(\sqrt{d} + \sqrt{d}\frac{\sigma_{R^2}}{\sigma_{\min}} + \sqrt{\lambda}W\right) = \tilde{\mathcal{O}}\left(\sqrt{d} + \sqrt{d}\frac{\sigma_{R^2}}{\sigma_{\min}}\right)$ and $\beta_0 = \tilde{\mathcal{O}}\left(\frac{\sqrt{d^3H}\mathcal{H}}{\sigma_{\min}} + \sqrt{d\lambda}\mathcal{H}\right) = \tilde{\mathcal{O}}\left(\frac{\sqrt{d^3H}\mathcal{H}}{\sigma_{\min}} + \sqrt{d}\right)$. Therefore,

$$\sum_{(h,k)\in\mathcal{I}} b_{h,k} = \mathcal{O}\left(C + H^2\mathcal{H}\kappa d^3\left(\beta_0 + H\beta\right) + H\kappa\beta_{R^2}\right)$$

$$= \mathcal{O}(C) + \tilde{\mathcal{O}}\left(\frac{H^{2.5}d^{5.5}\mathcal{H}^2 + Hd^{1.5}\sigma_{R^2}}{\sigma_{\min}} + H^3d^{4.5}\mathcal{H} + Hd^{1.5}\right).$$

We then analyze $C$. Using $\sqrt{a+b} \leqslant \sqrt{a} + \sqrt{b}$ for non-negative numbers $a, b \geqslant 0$, we have

$$C = \tilde{\mathcal{O}}\left(\sqrt{dHK\mathcal{G}^*} + Hd^{0.5}K^{0.5}\sigma_{\min} + H^2d^{3.5}\mathcal{H} + H^2d^2\mathcal{H} + Hd^{0.5}\sigma_R + Hd\right).$$

Putting the results together, we have

$$\sum_{(h,k)\in\mathcal{I}} b_{h,k} = \tilde{\mathcal{O}}\left(\sqrt{dHK\mathcal{G}^*} + Hd^{0.5}K^{0.5}\sigma_{\min} + \frac{H^{2.5}d^{5.5}\mathcal{H}^2 + Hd^{1.5}\sigma_{R^2}}{\sigma_{\min}} + H^3d^{4.5}\mathcal{H} + Hd^{0.5}\sigma_R + Hd^{1.5}\right).$$

$\square$

### F.11.1 Proof of Lemma F.2

*Proof of Lemma F.2.* By the definition of $E_{h,k}$ in equation A.7, it follows that

$$\sum_{k=1}^{K}\sum_{h=1}^{H} E_{h,k} \leqslant \sum_{k=1}^{K}\sum_{h=1}^{H}\left[2\mathcal{H}\beta_0\|\phi_{h,k}\|_{\boldsymbol{H}_{h,k-1}^{-1}} + \mathcal{H}\cdot\left[\widehat{\mathbb{P}}_{h,k}(\overline{V}_{h+1}^k - \underline{V}_{h+1}^k)\right](s_{h,k}, a_{h,k})\right]$$

$$\overset{(a)}{\leqslant} \sum_{k=1}^{K}\sum_{h=1}^{H}\left[4\mathcal{H}\beta_0\|\phi_{h,k}\|_{\boldsymbol{H}_{h,k-1}^{-1}} + \mathcal{H}\cdot\left[\mathbb{P}_h(\overline{V}_{h+1}^k - \underline{V}_{h+1}^k)\right](s_{h,k}, a_{h,k})\right]$$

$$\overset{(b)}{\leqslant} (4\beta_0 + 16H\beta)\mathcal{H}\cdot\sum_{k=1}^{K}\sum_{h=1}^{H}\|\phi_{h,k}\|_{\boldsymbol{H}_{h,k-1}^{-1}} + 38H^2\mathcal{H}^2\log\frac{4\lceil\log_2 HK\rceil}{\delta}$$

$$= \mathcal{O}\left((\beta_0 + H\beta)\mathcal{H}\cdot\sum_{k=1}^{K}\sum_{h=1}^{H}\|\phi_{h,k}\|_{\boldsymbol{H}_{h,k-1}^{-1}} + H^2\mathcal{H}^2\log\frac{4\lceil\log_2 HK\rceil}{\delta}\right)$$

where $(a)$ uses $\left|[(\widehat{\mathbb{P}}_{h,k} - \mathbb{P}_h)\overline{V}_{h+1}^k](s_{h,k}, a_{h,k})\right| = |\langle\phi_{h,k}, (\boldsymbol{\mu}_{h,k-1} - \boldsymbol{\mu}_h^*)\overline{V}_{h+1}^k\rangle| \leqslant \beta_0\|\phi_{h,k}\|_{\boldsymbol{H}_{h,k-1}^{-1}}$ on $\mathcal{B}_0$ and $(b)$ follows from Lemma D.9. $\square$

### F.11.2 Proof of Lemma F.3

*Proof of Lemma F.3.* By Lemma D.3, on the event $\mathcal{B}_R$, we have $\left|[\widehat{\mathbb{V}}_h\widehat{R}_h - \mathbb{V}_hR_h](s_{h,k}, a_{h,k})\right| \leqslant R_{h,k}$ for all $h \in [H]$ and $k \in [K]$. By Lemma D.5, on the event $\mathcal{B}_0\bigcap\mathcal{B}_V$, $[\widehat{\mathbb{V}}_h\overline{V}_{h+1}^k](s_{h,k}, a_{h,k}) \leqslant [\mathbb{V}_hV_{h+1}^*](s_{h,k}, a_{h,k}) + U_{h,k}$ for all $h \in [H]$ and $k \in [K]$. Therefore,

$$\sum_{k=1}^{K}\sum_{h=1}^{H} J_{h,k} \leqslant \sum_{k=1}^{K}\sum_{h=1}^{H}[\mathbb{V}_hR_h + \mathbb{V}_hV_{h+1}^*](s_{h,k}, a_{h,k}) + 2\sum_{k=1}^{K}\sum_{h=1}^{H} R_{h,k} + 2\sum_{k=1}^{K}\sum_{h=1}^{H} U_{h,k}$$

$$:= (I) + (II) + (III).$$

For the term $(III)$, we have

$$\sum_{k=1}^{K}\sum_{h=1}^{H} U_{h,k} = \sum_{k=1}^{K}\sum_{h=1}^{H}\left[11\mathcal{H}\beta_0\cdot\|\phi_{h,k}\|_{\boldsymbol{H}_{h,k-1}^{-1}} + 4\mathcal{H}\cdot\widehat{\mathbb{P}}_{h,k}(\overline{V}_{h+1}^k - \underline{V}_{h+1}^k)(s_{h,k}, a_{h,k})\right]$$

$$\overset{(a)}{\leqslant} \sum_{k=1}^{K} \sum_{h=1}^{H} \left[ 19\mathcal{H}\beta_0 \cdot \|\phi_{h,k}\|_{\boldsymbol{H}_{h,k-1}^{-1}} + 4\mathcal{H} \cdot \mathbb{P}_h(\overline{V}_{h+1}^k - \underline{V}_{h+1}^k)(s_{h,k}, a_{h,k}) \right]$$

$$\overset{(b)}{\leqslant} (19\beta_0 + 64H\beta)\mathcal{H} \cdot \sum_{k=1}^{K} \sum_{h=1}^{H} \|\phi_{h,k}\|_{\boldsymbol{H}_{h,k-1}^{-1}} + 152H^2\mathcal{H}^2 \log \frac{4\lceil \log_2 HK \rceil}{\delta}, \qquad \text{(F.18)}$$

where $(a)$ uses $\left| [(\widehat{\mathbb{P}}_{h,k} - \mathbb{P}_h)\overline{V}_{h+1}^k](s_{h,k}, a_{h,k}) \right| = |\langle \phi_{h,k}, (\boldsymbol{\mu}_{h,k-1} - \boldsymbol{\mu}_h^*)\overline{V}_{h+1}^k \rangle| \leqslant \beta_0 \|\phi_{h,k}\|_{\boldsymbol{H}_{h,k-1}^{-1}}$ on $\mathcal{B}_0$; and $(b)$ follows from Lemma D.9.

For the term $(II)$, we have

$$\sum_{k=1}^{K} \sum_{h=1}^{H} R_{h,k} = \beta_{R^2} \sum_{k=1}^{K} \sum_{h=1}^{H} \|\widetilde{\phi}_{h,k}\|_{\widetilde{\boldsymbol{H}}_{h,k-1}^{k-1}} + 2\mathcal{H}\beta_R \sum_{k=1}^{K} \sum_{h=1}^{H} \|\phi_{h,k}\|_{\boldsymbol{H}_{h,k-1}^{-1}}. \qquad \text{(F.19)}$$

We provide two ways to analyze the term $(I)$.

- On one hand, we denote $X_k = \sum_{h=1}^{H} [\mathbb{V}_h R_h + \mathbb{V}_h V_{h+1}^*](s_{h,k}, a_{h,k})$ for simplicity. Let $\mathcal{G}_k := \mathcal{F}_{H,k}$ be the $\sigma$-field generated by all the random variables over the first $k$ episodes. Then $\pi_k$ is $\mathcal{G}_{k-1}$-measurable, $X_k \geqslant 0$ is $\mathcal{G}_k$-measurable, and $|X_k| \leqslant H(\sigma_R^2 + \mathcal{H}^2)$. Therefore, $|X_k - \mathbb{E}[X_k|\mathcal{G}_{k-1}]| \leqslant H(\sigma_R^2 + \mathcal{H}^2)$ and $\mathrm{Var}[X_k|\mathcal{G}_{k-1}] \leqslant H(\sigma_R^2 + \mathcal{H}^2) \cdot \mathbb{E}[X_k|\mathcal{G}_{k-1}]$. By the variance-aware Freedman inequality in Lemma G.2, with probability at least $1 - \delta$, we have

$$\sum_{k=1}^{K} X_k \leqslant \sum_{k=1}^{K} \mathbb{E}[X_k|\mathcal{F}_{k-1}] + 3\sqrt{H(\sigma_R^2 + \mathcal{H}^2) \sum_{k=1}^{K} \mathbb{E}[X_k|\mathcal{G}_{k-1}] \log \frac{2\lceil \log_2 K \rceil}{\delta}}$$

$$+ 5H(\sigma_R^2 + \mathcal{H}^2) \log \frac{2\lceil \log_2 K \rceil}{\delta}$$

$$\leqslant 3 \sum_{k=1}^{K} \mathbb{E}[X_k|\mathcal{F}_{k-1}] + 7H(\sigma_R^2 + \mathcal{H}^2) \log \frac{2\lceil \log_2 K \rceil}{\delta}.$$

Notice that

$$\mathbb{E}[X_k|\mathcal{F}_{k-1}] = \mathbb{E}\left[ \sum_{h=1}^{H} [\mathbb{V}_h R_h + \mathbb{V}_h V_{h+1}^*](s_{h,k}, a_{h,k}) \middle| \mathcal{G}_{k-1} \right]$$

$$= \sum_{h=1}^{H} \mathbb{E}_{(s,a) \sim d_h^{\pi_k}} [\mathbb{V}_h R_h + \mathbb{V}_h V_{h+1}^*](s, a)$$

where $d_h^{\pi_k}(s, a) = \mathbb{P}^{\pi_k}(s_h = s, a_h = a|s_0 = s_{1,k})$ is the probability reaching $(s_{h,k}, a_{h,k}) = (s, a)$ at the $h$-th step when the agent starts from $s_{1,k}$ and follows the policy $\pi_k$. Therefore, we have

$$(I) \leqslant 3 \sum_{k=1}^{K} \sum_{h=1}^{H} \mathbb{E}_{(s,a) \sim d_h^{\pi_k}} [\mathbb{V}_h R_h + \mathbb{V}_h V_{h+1}^*](s, a) + 7H(\sigma_R^2 + \mathcal{H}^2) \log \frac{2\lceil \log_2 K \rceil}{\delta}$$

$$\leqslant 3\mathcal{G}_0^* K + 7H(\sigma_R^2 + \mathcal{H}^2) \log \frac{2\lceil \log_2 K \rceil}{\delta}$$

where

$$\mathcal{G}_0^* = \frac{1}{K} \sum_{k=1}^{K} \sum_{h=1}^{H} \mathbb{E}_{(s,a) \sim d_h^{\pi_k}} [\mathbb{V}_h R_h + \mathbb{V}_h V_{h+1}^{\pi_k}](s, a).$$

- On the other hand, we have

$$(I) = \sum_{k=1}^{K} \sum_{h=1}^{H} \left[ \mathbb{V}_h V_{h+1}^* - \mathbb{V}_h V_{h+1}^{\pi_k} \right](s_{h,k}, a_{h,k}) + \sum_{k=1}^{K} \sum_{h=1}^{H} \left[ \mathbb{V}_h R_h + \mathbb{V}_h V_{h+1}^{\pi_k} \right](s_{h,k}, a_{h,k})$$

$$\overset{equation\ F.20}{\leqslant} 2\mathcal{H} \cdot \sum_{k=1}^{K} \sum_{h=1}^{H} \mathbb{P}_h(\overline{V}_{h+1}^k - V_{h+1}^{\pi_k})(s_{h,k}, a_{h,k}) + \sum_{k=1}^{K} \sum_{h=1}^{H} [\mathbb{V}_h R_h + \mathbb{V}_h V_{h+1}^{\pi_k}](s_{h,k}, a_{h,k})$$

$$\leqslant 2\mathcal{H} \cdot \sum_{k=1}^{K} \sum_{h=1}^{H} \mathbb{P}_h(\overline{V}_{h+1}^k - V_{h+1}^{\pi_k})(s_{h,k}, a_{h,k}) + 2\mathcal{V}^2 K + 2H(\sigma_R^2 + \mathcal{H}^2) \log \frac{1}{\delta}$$

$$\leqslant 2\mathcal{V}^2 K + 2H(\sigma_R^2 + \mathcal{H}^2) \log \frac{1}{\delta} + 16H\beta\mathcal{H} \sum_{k=1}^{K} \sum_{h=1}^{H} \|\phi_{h,k}\|_{\boldsymbol{H}_{h,k-1}^{-1}} + 76H^2\mathcal{H}^2 \log \frac{4\lceil \log_2 HK \rceil}{\delta}$$

$$\leqslant 2\mathcal{V}^2 K + 16H\beta\mathcal{H} \sum_{k=1}^{K} \sum_{h=1}^{H} \|\phi_{h,k}\|_{\boldsymbol{H}_{h,k-1}^{-1}} + 78H^2\mathcal{H}^2 \log \frac{4\lceil \log_2 HK \rceil}{\delta} + 2H\sigma_R^2 \log \frac{1}{\delta}$$

where the first inequality uses equation F.20, the second inequality uses Lemma F.4, and the third inequality uses Lemma D.8.

$$\begin{aligned}
\left[\mathbb{V}_h V_{h+1}^* - \mathbb{V}_h V_{h+1}^{\pi_k}\right](s_{h,k}, a_{h,k}) &= \mathbb{P}_h[V_{h+1}^*]^2(s_{h,k}, a_{h,k}) - [\mathbb{P}_h V_{h+1}^*(s_{h,k}, a_{h,k})]^2 \\
&\quad - \left(\mathbb{P}_h[V_{h+1}^{\pi_k}]^2(s_{h,k}, a_{h,k}) - [\mathbb{P}_h V_{h+1}^{\pi_k}(s_{h,k}, a_{h,k})]^2\right) \\
&\overset{(a)}{\leqslant} \mathbb{P}_h[V_{h+1}^*]^2(s_{h,k}, a_{h,k}) - \mathbb{P}_h[V_{h+1}^{\pi_k}]^2(s_{h,k}, a_{h,k}) \\
&\overset{(b)}{\leqslant} 2\mathcal{H} \cdot \mathbb{P}_h(V_{h+1}^* - V_{h+1}^{\pi_k})(s_{h,k}, a_{h,k}) \\
&\overset{(c)}{\leqslant} 2\mathcal{H} \cdot \mathbb{P}_h(\overline{V}_{h+1}^k - V_{h+1}^{\pi_k})(s_{h,k}, a_{h,k})
\end{aligned} \tag{F.20}$$

where $(a)$ uses $V_{h+1}^*(\cdot) \geqslant V_{h+1}^{\pi_k}(\cdot)$, $(b)$ uses $V_{h+1}^{\pi_k}(\cdot) \leqslant V_{h+1}^*(\cdot) \leqslant \mathcal{H}$, and $(c)$ uses Lemma D.4.

Finally, we are going to put the pieces together. In order to simplicity notation, we use $b_{h,k} = \max\{\|\phi_{h,k}\|_{\boldsymbol{H}_{h,k-1}^{-1}}, \|\widetilde{\phi}_{h,k}\|_{\widetilde{\boldsymbol{H}}_{h,k-1}^{-1}}\}$ and $\beta = \beta_V + \beta_R$. From the first bullet point, we have

$$\sum_{k=1}^{K} \sum_{h=1}^{H} J_{h,k} = \mathcal{O}\left(\mathcal{G}_0^* \cdot K + [(\beta_0 + H\beta)\mathcal{H} + \beta_{R^2}] \cdot \sum_{k=1}^{K} \sum_{h=1}^{H} b_{h,k} + H^2\mathcal{H}^2 \log \frac{4\lceil \log_2 HK \rceil}{\delta} + H\sigma_R^2 \log \frac{1}{\delta}\right).$$

From the second bullet point, we have

$$\sum_{k=1}^{K} \sum_{h=1}^{H} J_{h,k} = \mathcal{O}\left(\mathcal{V}^2 K + [(\beta_0 + H\beta)\mathcal{H} + \beta_{R^2}] \sum_{k=1}^{K} \sum_{h=1}^{H} b_{h,k} + H^2\mathcal{H}^2 \log \frac{4\lceil \log_2 HK \rceil}{\delta} + H\sigma_R^2 \log \frac{1}{\delta}\right).$$

Taking minimum of the last two inequalities and using $\min\{\mathcal{G}_0^*, \mathcal{V}^2\} \leqslant \mathcal{G}^*$ complete the proof. □

### F.11.3 Proof of Lemma F.4

**Lemma F.4** (Total variance lemma). With probability at least $1 - \delta$, we have

$$\sum_{k=1}^{K} \sum_{h=1}^{H} [\mathbb{V}_h R_h + \mathbb{V}_h V_{h+1}^{\pi_k}](s_{h,k}, a_{h,k}) \leqslant 2\mathcal{V}^2 K + 2H(\sigma_R^2 + \mathcal{H}^2) \log \frac{1}{\delta}.$$

*Proof of Lemma F.4.* The proof uses a similar argument as Lemma C.5 in (Jin et al., 2018). Notice that the first state $s_{1,k}$ is fixed and $a_{h,k} = \pi_h^k(s_{h,k})$. Therefore, $(s_{2,k}, \cdots, s_{H,k})$ is a sequence generated by following policy $\pi_k$ starting at $s_{1,k}$. Let $\mathcal{G}_k$ be the $\sigma$-field generated by all the random variables over the first $k$ episodes. $X_k = \sum_{h=1}^{H} [\mathbb{V}_h R_h + \mathbb{V}_h V_{h+1}^{\pi_k}](s_{h,k}, a_{h,k})$. We have the following properties about $X_k$. Clearly $\pi_k$ is $\mathcal{G}_{k-1}$-measurable, $X_k \geqslant 0$ is $\mathcal{G}_k$-measurable, and $|X_k| \leqslant H(\sigma_R^2 + \mathcal{H}^2)$.

Let $\mathbb{E}_k(\cdot) := \mathbb{E}[\cdot|\mathcal{G}_k]$ for simplicity.

$$\mathcal{V}^2 \geqslant \mathbb{E}_{k-1}\left[\sum_{h=1}^{H} R_h(s_{h,k}, a_{h,k}) - V_1^{\pi_k}(s_{1,k})\right]^2$$

$$\overset{(a)}{=} \mathbb{E}_{k-1}\left[\sum_{h=1}^{H}\left(R_h(s_{h,k},a_{h,k}) + V_{h+1}^{\pi_k}(s_{h+1,k}) - V_h^{\pi_k}(s_{h,k})\right)\right]^2$$

$$\overset{(b)}{=} \sum_{h=1}^{H}\mathbb{E}_{k-1}\left[R_h(s_{h,k},a_{h,k}) + V_{h+1}^{\pi_k}(s_{h+1,k}) - V_h^{\pi_k}(s_{h,k})\right]^2$$

$$\overset{(c)}{=} \sum_{h=1}^{H}\mathbb{E}_{k-1}\left[[R_h - r_h]^2(s_{h,k},a_{h,k}) + \left[r_h(s_{h,k},a_{h,k}) + V_{h+1}^{\pi_k}(s_{h+1,k}) - V_h^{\pi_k}(s_{h,k})\right]^2\right]$$

$$\overset{(d)}{=} \mathbb{E}_{k-1}\sum_{h=1}^{H}[\mathbb{V}_h R_h + \mathbb{V}_h V_{h+1}^{\pi_k}](s_{h,k},a_{h,k}) = \mathbb{E}[X_k|\mathcal{F}_{k-1}]$$

where $(a)$ uses $V_{H+1}^{\pi_k}(\cdot) = 0$, $(b)$ uses the independence due to the Markov property, $(c)$ holds since $R_h(s_{h,k},a_{h,k})$ is independent with $s_{h+1,k}$ conditioning on $(s_{h,k},a_{h,k})$, and $(d)$ uses $V_h^{\pi_k}(s_{h,k}) = r_h(s_{h,k},a_{h,k}) + \mathbb{E}_{s_{h+1,k}\sim\mathbb{P}_h(\cdot|s_{h,k},a_{h,k})}[V_{h+1}^{\pi_k}(s_{h+1,k})]$. Using $\text{Var}[X_k|\mathcal{G}_{k-1}] \leqslant H(\sigma_R^2 + \mathcal{H}^2)\cdot\mathbb{E}[X_k|\mathcal{G}_{k-1}]$, we have

$$\sum_{k=1}^{K}\text{Var}[X_k|\mathcal{G}_{k-1}] \leqslant H(\sigma_R^2 + \mathcal{H}^2)\cdot\sum_{k=1}^{K}\mathbb{E}[X_k|\mathcal{G}_{k-1}] \leqslant (\sigma_R^2 + \mathcal{H}^2)\mathcal{V}^2 HK.$$

By the Freedman inequality in Lemma G.1, with probability at least $1 - \delta$, we have

$$\sum_{k=1}^{K}\sum_{h=1}^{H}[\mathbb{V}_h R_h + \mathbb{V}_h V_{h+1}^{\pi_k}](s_{h,k},a_{h,k})$$

$$= \sum_{k=1}^{K}X_k \leqslant \sum_{k=1}^{K}\mathbb{E}[X_k|\mathcal{F}_{k-1}] + \sqrt{2(\sigma_R^2 + \mathcal{H}^2)\mathcal{V}^2 HK\log\frac{1}{\delta}} + \frac{2}{3}H(\sigma_R^2 + \mathcal{H}^2)\log\frac{1}{\delta}$$

$$\leqslant \mathcal{V}^2 K + 2\sqrt{\mathcal{V}^2 K\cdot H(\sigma_R^2 + \mathcal{H}^2)\log\frac{1}{\delta}} + \frac{2}{3}H(\sigma_R^2 + \mathcal{H}^2)\log\frac{1}{\delta}$$

$$\leqslant 2\mathcal{V}^2 K + 2H(\sigma_R^2 + \mathcal{H}^2)\log\frac{1}{\delta}.$$

$\square$

# G  Auxiliary Lemmas

## G.1  Concentration Inequalities

**Lemma G.1** (Freedman inequality (Freedman, 1975)). Let $\{X_t\}_{t\in[T]}$ be a stochastic process that adapts to the filtration $\mathcal{F}_t$ so that $X_t$ is $\mathcal{F}_t$-measurable, $\mathbb{E}[X_t|\mathcal{F}_{t-1}] = 0$, $|X_t| \leqslant M$ and $\sum_{t=1}^{T}\mathbb{E}[X_t^2|\mathcal{F}_{t-1}] \leqslant V$ where $M > 0$ and $V > 0$ are positive constants. Then with probability at least $1 - \delta$, we have

$$\sum_{t=1}^{T}X_t \leqslant \sqrt{2V\ln\frac{1}{\delta}} + \frac{2M}{3}\ln\frac{1}{\delta}.$$

**Lemma G.2** (Variance-aware Freedman inequality). Let $\{X_t\}_{t\in[T]}$ be a stochastic process that adapts to the filtration $\mathcal{F}_t$ so that $X_t$ is $\mathcal{F}_t$-measurable, $\mathbb{E}[X_t|\mathcal{F}_{t-1}] = 0$, $|X_t| \leqslant M$ and $\sum_{t=1}^{T}\mathbb{E}[X_t^2|\mathcal{F}_{t-1}] \leqslant V^2$ where $M > 0$ and $V > 0$ are positive constants. Then with probability at least $1 - \delta$, we have

$$\left|\sum_{t=1}^{T}X_t\right| \leqslant 3\sqrt{\sum_{t=1}^{T}\mathbb{E}[X_t^2|\mathcal{F}_{t-1}]\cdot\log\frac{2K}{\delta}} + 5M\log\frac{2K}{\delta}$$

where $K = 1 + \lceil 2\log_2\frac{V}{M}\rceil$.

*Proof of Lemma G.2.* By Theorem 5 in (Li et al., 2021), we have for any positive integer $K \geqslant 1$,

$$\mathbb{P}\left(\left|\sum_{t=1}^{T} X_t\right| \leqslant \sqrt{8 \max\left\{\sum_{t=1}^{T} \mathbb{E}[X_t^2|\mathcal{F}_{t-1}], \frac{V^2}{2^K}\right\} \cdot \ln \frac{2K}{\delta}} + \frac{4M}{3} \ln \frac{2K}{\delta}\right) \geqslant 1 - \delta.$$

By setting $K = 1 + \lceil 2 \log_2 \frac{V}{M} \rceil$, we have $\frac{V^2}{2^K} \leqslant M^2$. Using $\max\{a, b\} \leqslant a + b$, $\sqrt{a+b} \leqslant \sqrt{a} + \sqrt{b}$ for any $a, b \geqslant 0$ and $\ln \frac{2K}{\delta} \geqslant 1$, we complete the proof. $\qquad\square$

The following two lemmas are the counterpart lemmas of Theorem 2.1 under the light-tail assumption.

**Lemma G.3** (Bernstein's inequality for self-normalized martingales, Lemma F.4 in (Hu et al., 2022)). Let $\{\mathcal{G}_t\}_{t\geqslant 0}$ be a filtration and $\{\boldsymbol{x}_t, \eta_t\}_{t\geqslant 0}$ be a stochastic process so that $\boldsymbol{x}_t \in \mathbb{R}^d$ is $\mathcal{G}_t$-measurable and $\eta_t \in \mathbb{R}$ is $\mathcal{G}_{t+1}$-measurable.. If $\|\boldsymbol{x}_t\| \leqslant L$ and $\{\eta_t\}_{t\geqslant 1}$ satisfies that $\mathbb{E}[\eta_t|\mathcal{G}_t] = 0$, $\mathbb{E}[\eta_t^2|\mathcal{G}_t] \leqslant \sigma^2$ and $|\eta_t \min\left\{1, \|\boldsymbol{x}_t\|_{\boldsymbol{Z}_{t-1}^{-1}}\right\}| \leqslant M$ for all $t \geqslant 1$. Then, for any $\delta \in (0, 1)$, with probability at least $1 - \delta$, we have for all $t \geqslant 1$,

$$\left\|\sum_{j=1}^{t} \boldsymbol{x}_j \eta_j\right\|_{\boldsymbol{Z}_t^{-1}} \leqslant 8\sigma \sqrt{d \log\left(1 + \frac{tL^2}{d\lambda}\right) \log \frac{4t^2}{\delta}} + 4M \log \frac{4t^2}{\delta}$$

where $\boldsymbol{Z}_t = \lambda \boldsymbol{I} + \sum_{j=1}^{t} \boldsymbol{x}_j \boldsymbol{x}_j^\top$ for $t \geqslant 1$ and $\boldsymbol{Z}_0 = \lambda \boldsymbol{I}$.

**Lemma G.4** (Hoeffding inequality for self-normalized martingales, Theorem 1 in (Abbasi-Yadkori et al., 2011)). Let $\{\mathcal{G}_t\}_{t\geqslant 0}$ be a filtration and $\{\boldsymbol{x}_t, \eta_t\}_{t\geqslant 0}$ be a stochastic process so that $\boldsymbol{x}_t \in \mathbb{R}^d$ is $\mathcal{G}_t$-measurable and $\eta_t \in \mathbb{R}$ is $\mathcal{G}_{t+1}$-measurable.. If $\|\boldsymbol{x}_t\| \leqslant L$ and $\{\eta_t\}_{t\geqslant 1}$ satisfies that $\mathbb{E}[\eta_t|\mathcal{G}_t] = 0$ and $|\eta_t| \leqslant M$ for all $t \geqslant 1$. Then, for any $\delta \in (0, 1)$, with probability at least $1 - \delta$, we have for all $t \geqslant 1$,

$$\left\|\sum_{j=1}^{t} \boldsymbol{x}_j \eta_j\right\|_{\boldsymbol{Z}_t^{-1}} \leqslant M\sqrt{d \log\left(1 + \frac{tL^2}{d\lambda}\right) + \log \frac{1}{\delta}}$$

where $\boldsymbol{Z}_t = \lambda \boldsymbol{I} + \sum_{j=1}^{t} \boldsymbol{x}_j \boldsymbol{x}_j^\top$ for $t \geqslant 1$ and $\boldsymbol{Z}_0 = \lambda \boldsymbol{I}$.

## G.2 Elliptical Lemmas

**Lemma G.5** (Lemma 11 in (Abbasi-Yadkori et al., 2011)). Let $\{\boldsymbol{x}_t\}_{t\geqslant 1} \subset \mathbb{R}^d$ and assume $\|\boldsymbol{x}_t\| \leqslant L$ for all $t \geqslant 1$. Set $\boldsymbol{Z}_t = \sum_{s=1}^{t} \boldsymbol{x}_t \boldsymbol{x}_t^\top + \lambda \boldsymbol{I}$. Then it follows that

$$\sum_{t=1}^{T} \min\left\{1, \|\boldsymbol{x}_t\|_{\boldsymbol{Z}_{t-1}}^2\right\} \leqslant 2d \log\left(\frac{d\lambda + TL^2}{d\lambda}\right).$$

**Lemma G.6** (Lemma 12 in (Abbasi-Yadkori et al., 2011)). Suppose $\boldsymbol{A}, \boldsymbol{B} \in \mathbb{R}^{d \times d}$ are two positive definite matrices satisfying that $\boldsymbol{A} \succeq \boldsymbol{B}$, then for any $\boldsymbol{x} \in \mathbb{R}^d$,

$$\|\boldsymbol{x}\|_{\boldsymbol{B}^{-1}} \leqslant \|\boldsymbol{x}\|_{\boldsymbol{A}^{-1}} \sqrt{\frac{\det(\boldsymbol{A})}{\det(\boldsymbol{B})}}.$$

## G.3 Function Class and Covering Number

This subsection collects important lemmas in (He et al., 2022). Let $\mathcal{K} = \{k_1, k_2, \cdots\}$ denote the set of episodes where the algorithm updates the value function in Algorithm 3. For a given total number of episodes $K$, it follows that $|\mathcal{K}| \leqslant K$. Furthermore, due to the mechanism of rare-switching value function updates, $|\mathcal{K}|$ is much smaller than $K$.

**Lemma G.7.**

$$|\mathcal{K}| \leqslant dH \log_2\left(1 + \frac{K}{\lambda \sigma_{\min}^2}\right).$$

*Proof of Lemma G.7.* The proof is almost identical to Lemma E.1 in (He et al., 2022) except that we maintain the dependence on $\sigma_{\min}$. According to the determinant-based criterion, for each episode $k_i$, there exists a stage $h' \in [H]$ such that $\det(\boldsymbol{H}_{h',k_i-1}) \geqslant 2\det(\boldsymbol{H}_{h',k_{i-1}-1})$. Since we always have $\boldsymbol{H}_{h,k_i-1} \succeq \boldsymbol{H}_{h,k_{i-1}-1}$ for all $h \in [H]$, it then follows that

$$\prod_{h\in[H]} \det(\boldsymbol{H}_{h,k_i-1}) \geqslant 2 \prod_{h\in[H]} \det(\boldsymbol{H}_{h,k_{i-1}-1}).$$

By induction, it follows that

$$\prod_{h\in[H]} \det(\boldsymbol{H}_{h,k_{|\mathcal{K}|}-1}) \geqslant 2^{|\mathcal{K}|} \prod_{h\in[H]} \det(\boldsymbol{H}_{h,k_1-1}) \geqslant 2^{|\mathcal{K}|} \prod_{h\in[H]} \det(\lambda\boldsymbol{I}) = 2^{|\mathcal{K}|} \lambda^{dH}$$

On the other hand, due to $\boldsymbol{H}_{h,k_{|\mathcal{K}|}-1} \preceq \boldsymbol{H}_{h,K}$ the determinant $\det(\boldsymbol{H}_{h,k_{|\mathcal{K}|}-1}$ is upper bounded by

$$\prod_{h\in[H]} \det(\boldsymbol{H}_{h,k_{|\mathcal{K}|}-1}) \leqslant \prod_{h\in[H]} \det(\boldsymbol{H}_{h,K}) \leqslant \left(\lambda + \frac{K}{\sigma_{\min}^2}\right)^{dH}.$$

Combining the last two inequalities, we have

$$|\mathcal{K}| \leqslant dH \log_2 \left(1 + \frac{K}{\lambda\sigma_{\min}^2}\right).$$

$\square$

The optimistic value function $\overline{V}_h^k(\cdot) = \min_{k_i \leqslant k} \max_a \overline{Q}_h^{k_i}(\cdot, a)$ belong to the function class $\mathcal{V}^+$

$$\mathcal{V}^+ = \left\{ f | f(\cdot) = \max_{a\in\mathcal{A}} \min_{i\leqslant|\mathcal{K}|} \min \left\{ \boldsymbol{w}_i^\top \boldsymbol{\phi}(\cdot, a) + \beta\|\boldsymbol{\phi}(\cdot, a)\|_{\boldsymbol{H}_i^{-1}}, \mathcal{H} \right\}, \beta \in [0, B], \|\boldsymbol{w}_i\| \leqslant L, \boldsymbol{H}_i \succeq \lambda\boldsymbol{I} \right\}. \quad \text{(G.1)}$$

while the pessimistic value function $\underline{V}_h^k(\cdot) = \max_{k_i \leqslant k} \max_a \underline{Q}_h^{k_i}(\cdot, a)$ belong to the function class $\mathcal{V}^-$,

$$\mathcal{V}^- = \left\{ f | f(\cdot) = \max_{a\in\mathcal{A}} \max_{i\leqslant|\mathcal{K}|} \max \left\{ \boldsymbol{w}_i^\top \boldsymbol{\phi}(\cdot, a) - \beta\|\boldsymbol{\phi}(\cdot, a)\|_{\boldsymbol{H}_i^{-1}}, \mathcal{H} \right\}, \beta \in [0, B], \|\boldsymbol{w}_i\| \leqslant L, \boldsymbol{H}_i \succeq \lambda\boldsymbol{I} \right\}. \quad \text{(G.2)}$$

Here $B$ upper bounds $\beta$ and $L = W + \mathcal{H}\sqrt{\frac{dK}{\lambda}}$ is a uniformly bound for $\boldsymbol{\theta}_{h,k-1} + \boldsymbol{\mu}_{h,k-1}\overline{\boldsymbol{V}}_{h+1}^k$ because

$$\|\boldsymbol{\theta}_{h,k-1} + \boldsymbol{\mu}_{h,k-1}\overline{\boldsymbol{V}}_{h+1}^k\| \leqslant \|\boldsymbol{\theta}_{h,k-1}\| + \|\boldsymbol{\mu}_{h,k-1}\overline{\boldsymbol{V}}_{h+1}^k\| \leqslant W + \mathcal{H}\sqrt{\frac{dK}{\lambda}}$$

where the last inequality uses the boundedness of $\boldsymbol{\theta}_{h,k-1}$'s and the inequality $\|\boldsymbol{\mu}_{h,k-1}\overline{\boldsymbol{V}}_{h+1}^k\| \leqslant \mathcal{H}\sqrt{\frac{dK}{\lambda}}$ (whose proof can be found in Lemma E.2 of He et al. (2022)).

**Lemma G.8** (Covering number of value functions). Let $\mathcal{V}^\pm$ denote the class of optimistic or pessimistic value functions with definition in equation G.1 and equation G.2 respectively. Assume $\|\boldsymbol{\phi}(s, a)\| \leqslant 1$ for all $(s, a)$ pairs, and let $\mathcal{N}(\mathcal{V}, \varepsilon)$ be the $\varepsilon$-covering number of $\mathcal{V}$ with respective to the distance $\text{dist}(f, f') := \sup_{s\in\mathcal{S}} |f(s) - f'(s)|$. Then,

$$\log \mathcal{N}(\mathcal{V}^\pm, \varepsilon) \leqslant \left[ d\log\left(1 + \frac{4L}{\varepsilon}\right) + d^2\log\left(1 + \frac{8d^{1/2}B^2}{\lambda\varepsilon^2}\right) \right] \cdot |\mathcal{K}|.$$

*Proof of Lemma G.8.* The result about $\mathcal{V}_f^+$ follows from Lemma E.6 in (He et al., 2022). The result about $\mathcal{V}_f^-$ follows from Lemma E.7 in (He et al., 2022). $\square$

**Lemma G.9** (Covering number of squared functions, Lemma E.8 in (He et al., 2022)). For the squared function class $[\mathcal{V}^+]^2 := \{f^2 | f \in \mathcal{V}^+\}$, let $\mathcal{N}([\mathcal{V}^+]^2, \varepsilon)$ be the $\varepsilon$-covering number of $[\mathcal{V}^+]^2$ with respective to the distance $\text{dist}(f, f') := \sup_{s\in\mathcal{S}} |f(s) - f'(s)|$. Then

$$\log \mathcal{N}([\mathcal{V}^+]^2, \varepsilon) \leqslant \left[ d\log\left(1 + \frac{8HL}{\varepsilon}\right) + d^2\log\left(1 + \frac{32d^{1/2}H^2B^2}{\lambda\varepsilon^2}\right) \right] \cdot |\mathcal{K}|.$$

