# OpenReview forum: "Variance-aware decision making with linear function approximation under heavy-tailed rewards"
_TMLR — Accepted by TMLR_

### Review · Reviewer_oNBg · 2023-11-20

**Summary Of Contributions:**

This paper achieves variance-based regret bounds for linear bandits with heavy-tailed rewards. While the task of attaining variance-based regret or handling heavy-tailed rewards have supposedly been studied in isolation w.r.t. linear bandits, this work claims to be the first to jointly study the two problems. The main approach is to combine OFUL with Huber regression techniques. As an application, they show similar regret bounds can be achieved for linear MDPs, which yields tighter instance-dependent rates than previously known.

**Audience:**

Yes

**Broader Impact Concerns:**

No broader impact concerns.

**Claims And Evidence:**

Yes

**Requested Changes:**

## Writing Suggestions/Typos
* section 1.1, second paragraph "Adatptive" should be "Adaptive"
* page 4, below display (2.5) "deinfed" should be "defined"
* page 6, below display (2.10), "Combing" should be "Combining".
* Is dependence on $b$ being hidden in Lemma 2.2 or are you taking $b=1$ WLOG? I'm a bit confused by the $b=1$ remark at the end of page 6.
* In Lemma G.2, it seems the inequality with V^2 is unnecessary as you just want to say Freedman's inequality holds for the actual sum of conditiional variances rather than an upper bound.

**Strengths And Weaknesses:**

## Strengths
* The paper is well written and the techniques are presented in a natural order for ease of reading.
* It is nice that their results for linear MDP's does not require knowledge of the new variance parameter and is tighter than previously known rates.
* The theoretical findings are supported by experiments which show the superior performance of their new algorithm.

## Weaknesses
* It is a bit unclear to me how the setting is "heavy-tailed linear bandits". In Definition 2.1, one is merely saying the conditional variance can be a generic quantity $\nu_t^2$. While this recovers the setting with bounded second moments, what about the typical heavy-tailed setting where one assumes rewards have $(1+\epsilon)$ moments which are bounded (e.g., see the cited paper Xue et al., 2021)? Without considering the full setup for generic $\epsilon>0$, it just seems like this paper is restricted to the subgaussian case. See also the work "Variance-aware Spare Linear Bandits" (Dai et al., 2023).
* I'm also concerned about the similarity to "Variance-Dependent Regret Bounds for Linear Bandits and Reinforcement Learning: Adaptivity and Computational Efficiency" (Zhao et al., 2023). It seems like they can already get the variance-aware regret in the same setting as this paper. The citation of this paper is noticeably absent despite the similarities.
* Please also compare with the work "Heavy-tailed Linear Bandit with Huber Regression" by Kang and Kim, 2023, which is also not cited.

In conclusion, I would like to see the questions of (1) whether the results of this paper can really capture the full "heavy tailed" setting and (2) how the result compares to the recent and similar mentioned papers above. It is concerning that these recent works are not cited, and so I am left unsure of whether the results of this paper are not already contained in other papers, technical contributions notwithstanding.

---

> ### Author Response · Authors · 2023-11-21
> **Thanks for your reviews.**
>
> Thank you for your valuable feedback and the time in reviewing our work. We sincerely appreciate your efforts. In the next revision, we will include citations to all the papers you mentioned and compare them with our study. Below, we provide our point-by-point responses to your queries.
>
> **1. Why not consider $(1+\epsilon)$-moment?**
>
> We agree with your comment that the typical heavy-tailed setting assumes $(1+\epsilon)$-moment. Note that our focus is on variance-aware regrets. So the weakest moment condition is bounded second moments, which is exactly what we used in our paper.
>
> Although we didn’t consider the full "heavy-tailed" setting, it is definitely possible to extend our technique and analysis to the $(1+\epsilon)$-moment setting. Actually, we are recently aware that this extension has been made by the following work [1]. As you can see, their algorithm is a variant of ours and their theories utilize the key idea that we depicted in Section 2.4.
>
> As the first work focusing on the variance-awareness and heavy-tail rewards, we believe our technique would inspire future works. Our focus on the bounded second moments might provide a familiar context for readers, aiding in quicker comprehension of our techniques.
>
> [1] Tackling Heavy-Tailed Rewards in Reinforcement Learning with Function Approximation: Minimax Optimal and Instance-Dependent Regret Bounds, https://arxiv.org/pdf/2306.06836.pdf
>
> **2. Is our approach limited to the subgaussian case?**
>
>  A sub-Gaussian random variable has moments of any order, but our assumption is that rewards have only bounded second moments. Note that a variable with bounded second moments isn’t necessarily subgaussian, though the reverse is always true.
>
> For example, in our experiments, we used Student t-distributions with degrees of freedom denoted by $df$. This distribution does not have moments of all orders (the finite order depends on $df$), and lacks a moment-generating function, as detailed on [Wikipedia](https://en.wikipedia.org/wiki/Student%27s_t-distribution). Therefore, it is not subgaussian, but our theory still provides theoretical guarantees for it.
>
> Therefore, our paper addresses a broader range of cases than just the subgaussian. This distinction is critical when discussing the paper "Variance-aware Sparse Linear Bandits" (Dai et al., 2023). Their related work indeed provides a good summary of the literature on variance-aware regrets. It's important to note that these studies primarily focus on establishing variance-aware regrets under the assumption of either bounded or sub-Gaussian rewards. This assumption suggests that their settings do not encompass ours. Consequently, our results extend beyond the scope of their coverage.
>
> **3. Similarity to (Zhao et al., 2023).**
>
> We will cite this recent work. We studied the same problem with (Zhao et al., 2023) but with different settings. Note that (Zhao et al., 2023) still assume bounded rewards (Please see their Theorem 2.1 at https://arxiv.org/pdf/2302.10371.pdf). As we explained in Point 2, their setting doesn’t cover heave-tail rewards. What’s more, we believe that once we replace their ridge regression with our proposed Adaptive pseudo-Huber regression, their results would hold again with a similar regret guarantee in heavy-tailed reward settings.

---

> ### Author Response · Authors · 2023-11-21
> **Thanks for your reviews (Cont'd).**
>
> **4. Comparison with (Kang and Kim, 2023).**
>
>  We appreciate the introduction to this relevant work. Kang and Kim (2023) explored heavy-tailed linear bandits using Huber regression, but their approach and conclusions differ significantly from ours:
>
> - (1) Their regret analysis is worst-case, whereas ours is variance-aware.
>
> - (2) They investigate context bandits where the arm $\phi_t$ is independently and identically distributed (i.i.d.) from a static distribution. In contrast, our study allows the agent to actively select $\phi_t$ from a dynamically changing action set $D_t$. This makes our approach more general, as it can be adapted to the context bandit framework by simply fixing the action set $D_t$
>  and having the agent sample $\phi_t$ from the same constant distribution. Therefore, our framework encompasses a broader range of scenarios than theirs, indicating that their results do not fully capture the complexities addressed in our work.
>
> - (3) They assume the distribution used for sampling arms has many good properties such as a positive sub-optimality gap (their Assumption 3) and sufficient exploration (their Assumption 4). As we already explained in Point (2), we don’t need these conditions.
>
> - (4) Our algorithm significantly differs from theirs. They employ an elimination-based method, whereas our approach relies on upper confidence bounds. While both our algorithm and theirs use (adaptive) Huber regression to select the linear coefficient, we adaptively set the robustification parameter $\tau_i$, in contrast to their approach of using a constant $\tau_i \equiv \tau$. As discussed in our contribution section, a constant $\tau_i$ is effective primarily in i.i.d. settings, which is why their study focuses on context bandits with i.i.d. sampled arms $\phi_t$. This difference leads us to believe that their method might not be suitable for adaptive online settings where the arms are not i.i.d., further distinguishing our approach from theirs.
>
>  Considering those differences, we conclude that their results and theories can’t encompass ours. In fact, we shouldn’t compare their work with ours directly, because we consider different settings. However, we believe a variant of our AdaOFUL would perform similarly well as their algorithm.
>
> **5. Typos.**
>
> (1) About the $b$ in Lemma 2.2. This is a typo. We should replace all $\kappa$ with $\kappa b^2$ in the current statement of Lemma 2.2. In the proof of Theorem 2.1, it is indeed true that $b=1$ because all reward noises are divided by their variance. So this typo doesn't make a difference. The reason we incorporate a general constant bound $b$ in Lemma 2.2 is that the theory for linear MDPs requires a non-constant $b$.
>
> (2) The function $V^2$ in Lemma G.2. The inequality with $V^2$ is actually necessary. The original Freedman's inequality requires that the sum of conditional variance is bounded by a constant. Please see https://www.jstor.org/stable/2959268 for more details.

---

### Review · Reviewer_hW9L · 2023-11-23

**Summary Of Contributions:**

This paper presents and analyzes the AdaOFUL linear bandit algorithm designed to handle heavy tailed reward distributions and the VARA algorithm for fixed horizon, linear MDPs under heavy tailed reward distributions. VARA utilizes AdaOFUL as a subroutine to estimate parameters for the first and second moments of the linear reward function and LSVI-UCB++ for optimistic policy improvement. The performance of AdaOFUL and VARA are analyzed in terms of regret bounds, computation, and memory compared with previous algorithms, where AdaOFUL and VARA have substantial advantages. Experiments suggest that AdaOFUL is indeed more effective practically than previous linear bandit algorithms in light tailed, heavy tailed, and unbounded reward distribution scenarios.

**Audience:**

Yes

**Broader Impact Concerns:**

No broader impact concerns.

**Claims And Evidence:**

Yes

**Requested Changes:**

No critical changes but I have some small questions and suggestions.

1. Algorithm 1 is difficult to understand with its many references to equations in the text. I would like to see the equations inlined with the current references changed to comments.
2. The $\tilde{O}$ notation is explained in Corollary 2.1 but it is used in Section 1.1. Could that explanation be moved closer to its first use?
3. The experiment figures are not very readable in greyscale, it would be nice if they were, especially since there are only three lines in each plot to differentiate.
4. For the experiment figures, I am not sure why the regret of AdaOFUL continues to grow substantially even when its distance to $\theta^*$ is relatively small. Especially in Cases (b) and (c), it looks like AdaOFUL's regret is growing only marginally slower than TOFU's, but AdaOFUL's convergence is much better. What am I missing?
5. Also for the experiment, 10 repetitions were run, but I do not see the variability reported in the main paper. A histogram of the final regret and convergence across replicas could be illuminating.
6. When defining MDPs, the state value function is defined as a sum of action values. I think that $a$ is unbound and instead the statement should be $V_h^{\pi}(\cdot) = Q_h^{\pi}(\cdot, \pi_h(\cdot))$.
7. Clause 2 of Assumption 3.2, it appears like $\mathcal{H}$ and $\mathcal{V}$ are used without a previous definition. Upon checking that they were not defined previously, I guessed that they were being defined here implicitly as bounding values but making that clear here would have saved me some time backtracking through the paper.
8. I cannot make sense of the second last sentence before the "Learning protocol" subsubsection near the end of Section 3.1, I think there is a typo.
9. Is $V_1^*$ defined before its use at the end of Section 3.1?

**Strengths And Weaknesses:**

This paper is easy to very read and it is well structured. The algorithms are straightforward improvements on previous iterations but the additions, particularly the hyperparameter settings, are entirely non-obvious without the associated theory. I am familiar with but not an expert on this area of theory, and these results are clear, their relationships to previous results are made clear, and I believe they are impactful. The one linear bandit experiment illustrates the practical implication of the theory and is a meaningful companion to the regret bounds. I have not checked the proofs carefully but the proof sketches and results look reasonable to me.

No meaningful weaknesses.

---

> ### Author Response · Authors · 2023-11-27
> **Thanks for your reviews.**
>
> Thank you for your insightful feedback and the time invested in reviewing our manuscript. We deeply value your constructive comments and support. Below are our detailed responses to each of your points:
>
> **1. Presentation of Algorithm 1**
>
> In the forthcoming revision, we will embed the equations directly into Algorithm 1 for clearer and more immediate comprehension. Nonetheless, we maintain our preference to defer the expression of $\beta_t$ to Theorem 2.1, facilitating a more direct path to understanding the final regret as outlined in Corollary 2.1.
>
> **2. Notation Clarification**
>
> The notation $\widetilde{O}(\cdot)$ will be explicitly explained in Section 1.1 for better clarity.
>
> **3. Figure Enhancements**
>
> To aid in distinguishing the curves in greyscale, we will employ different line styles for each in the revised figures. However, because we didn't save all the history estimates, we had to redo the experiments. Even though the curves are slightly different from the previous results, the observed patterns are the same.
>
> **4. Explanation of Gradual Regret Increase**
>
> First, we want to highlight that in prior experimental results, such as those presented by Shao et al. (2018), the continuous substantial growth in regret is also observed across all proposed algorithms studied therein.
>
> Second, the primary cause of the gradual increase in regrets is the presence of many suboptimal arms together with too large noises. With 20 arms in each action set and small optimality gaps (the difference between the highest and the selected rewards, whose expectation is typically in the range of 0.01-0.1), identifying the optimal arm becomes challenging. This difficulty is compounded by the existence of heavy-tail rewards and varying action sets. In particular, for the student distribution $t(df)$ with $df =1$, the added noise has infinite variance (which corresponds to Case (c)).
>
> In contrast, if we simplify the problem by reducing the action set to just fewer (such as two or five) arms and making the reward noises have smaller variance, all algorithms in our study would quickly identify the optimal arm, halting the growth of regret. Therefore, the increase in regrets is attributable to the problem's complexity which is caused by the abundance of suboptimal arms and large noise variance.
>
> **5. Variability Representation**
>
> We plan to use shadowing to depict variability. A brief report of the results will follow shortly.
>
> **6. Typographical Error**
>
> This is a typo. We will correct it as you suggest.
>
>
> **7. Constants $\mathcal{H}$ and $\mathcal{V}$**
>
> These are known constant upper bounds. In practice, their values can be estimated using a "doubling trick," as discussed in Footnote 1.
>
> **8. Regret Dominant Term**
>
> We will revise the text to clarify that guessing very large values for $\mathcal{H}$ and $\mathcal{V}$ does not affect the dominant term of our regret, provided the iteration $T$ is sufficiently large. These optimistic guesses influence only the non-leading terms in the regret.
>
> **9. Definition Addition**
>
> We will include the definition of $V_1^{\star}$ at the end of Section 3.1 for completeness.

---

### Review · Reviewer_eYei · 2023-11-27

**Summary Of Contributions:**

This paper investigates bandits and Markov Decision Processes (MDP) with heavy-tailed rewards. The main contribution of this paper is the construction of two variance-dependent regret bounds for bandits and linear MDP, respectively.

**Audience:**

Yes

**Broader Impact Concerns:**

I do not have any concerns on the ethical implications of the work.

**Claims And Evidence:**

Yes

**Requested Changes:**

Please see the comments in Strengths And Weaknesses.

**Strengths And Weaknesses:**

## Major concerns:

1. Lemma 2.2 is crucial for constructing the confidence region and addressing the heavy-tailed issue. However, I am uncertain about the correctness of the proof of Lemma 2.2.

   Specifically, at the top of page 30, the authors utilize Lemma C.1 to complete the proof, where Lemma C.1 requires $A_t$ to hold for any $t\leq1$. However, Lemma 2.2 aims to establish that $A_t$ holds with a probability of at least $1-\delta$ for any $t\leq1$. Lemma 2.2 is a condition of Lemma C.1, and Lemma C.1 is used to prove Lemma 2.2, which creates a contradiction.

   If mathematical induction is being used, please provide detailed steps for the induction, particularly the change of probability parameter $\delta$. I am open to the possibility of being completely mistaken about this point.

2. Based on the aforementioned concerns, I have doubts about the effectiveness of the Huber loss (Eq 2.3). Specifically, the Huber loss adaptively adjusts between squared loss and L1 loss to handle heavy-tailed problems, particularly the robustness of L1 loss when the loss value is extreme. However, both squared loss and L1 loss lack the ability to handle heavy-tailed issues, and their combination intuitively cannot reduce extreme values.

   Can the authors provide some intuitive examples to explain the effectiveness of this loss? In my opinion, if the Huber loss is effective, it would be a more elegant strategy than truncation and the median of means. I am open to the possibility of being completely mistaken about this point.

3. Can the authors conduct real-world experiments? Currently, most of the bandit models are verified using simulated datasets. If experiments can be conducted on real-world datasets, it would be a significant contribution to the community.

## Minor comments:

1. $R_h(s,a)$ above the Assumption is not defined. $r_h(s,a)$ represents the expected reward, while $R_h(s,a)$ represents the random reward. In step 17 of the formal VARA, it is observed that $r_{h,k}\sim R_h(s_h, a_{h,k})$, where $R_h(s_h, a_$h,k$)$ seems to be a distribution. The authors may consider distinguishing between random variables and expectations using uppercase and lowercase letters, as the current usage can easily cause confusion.
2. The MDP process has already been modeled through parameters $\theta_h^\*$ and $\mu_h^\*$, can the author provide an expression for $Q_h^\*(\cdot, \cdot)$ regarding  $\theta_h^\*$ and $\mu_h^\*$?
3. $V_1^\*$ in the learning protocol is not defined.
4. Lemma D.2 and Lemma D.4 are the most important tools for extending the bandit model to MDP. Regarding D.2, the author states: "The proof is quite standard (Jin et al., 2020b; Wagenmaker et al., 2022a; Hu et al., 2022)." What is the novelty of the analysis for VARA in Lemma D.2?
5. For VARA, I did not understand the meaning of the lower bound $\underline{V}_h^k(\cdot)$. It seems the lower bound does not used in the formal VARS. Can the author provide further explanation?

---

> ### Author Response · Authors · 2023-11-28
> **Thanks for your reviews.**
>
> Thank you for your detailed feedback and the time you dedicated to reviewing our manuscript. We greatly appreciate your insightful comments. Here are our responses to each of your main points:
>
> **1.Proof of Lemma 2.2**
>
> We utilized mathematical induction in proving Lemma 2.2. The mention of a "change of probability parameter $\delta$" was a typo.
>
>  We clarify the establishment of Lemma 2.2 in three steps:
> - First, Lemma C.1 and Lemma C.2 do not require that the event $A_t$ is true.
>
> - Second, Lemma C.1 and Lemma C.2 are proved simply using the Freedman inequality and a union bound. I guess the reviewer’s confusion partially comes from the way we use the union bound.   For each finite $T \ge 1$, we show that $\sum_{t=1}^T Y_t \le z_t$ (here we use $z_t$ to denote the bound for simplicity) is true with probability at least $1-\frac{\delta}{2T^2}$. Then, by the union bound, the probability of the event where there exists one $T \ge 1$ such that $\sum_{t=1}^T Y_t > z_t$ is no more than the sum of probabilities where $\sum_{t=1}^T Y_t > z_t$ over all possible $T \ge 1$.
>
> Note that this probability is no more than $\sum_{t=1}^{\infty} \frac{\delta}{2t^2} \le \delta$, which is the reason why we say there is a typo about the $\delta$. The correct probability is at least $1-\delta$. Hence, Lemma C.1 and Lemma C.2 are true with probability at least $1-\delta$ respectively. Our previous version showing that they are true with probability at least $1-\delta/2$ is a typo and has been corrected.
>
> - Third, assuming Lemmas C.1 and C.2 hold (which is true with probability at least $1-2\delta$), we use mathematical induction to assert all $A_t$ are true. For a detailed derivation, we invite the reviewer to consult our revised paper
>
> **2. The intuition behind Adaptive Huber Regression**
>
> The following are two intuitive examples to illustrate the effectiveness of the Huber loss:
>
> - Handling Outliers: Imagine a dataset with mostly regular values but a few extreme outliers. A squared loss function would disproportionately amplify the impact of these outliers (since the loss grows quadratically with the error), potentially skewing the model. L1 loss, while robust to outliers, might underrepresent the influence of smaller errors. The Huber loss, by smoothly transitioning from squared to L1 loss, can mitigate the influence of outliers while still being responsive to smaller, more typical data points.
>
> - Balancing Sensitivity and Robustness: Consider a scenario where most data points are close to the model's prediction, but a few are significantly off. A model trained solely on squared loss might overfit these anomalies. Conversely, a model trained on L1 loss might overlook subtle but important patterns in the data. The Huber loss strikes a balance, allowing the model to be sensitive to general trends while not being overly influenced by rare, extreme deviations.
>
> Comparatively, truncation and the median of means are more explicit methods for handling heavy-tailed data. Truncation directly limits the influence of extreme values, while the median of means divides data into subsets to mitigate the impact of outliers. However, these methods can sometimes be too aggressive, potentially discarding useful information or oversimplifying the data’s complexity.
>
> The elegance of the Huber loss lies in its adaptability: it doesn't discard data or require the data to be segmented, as in truncation or median of means, respectively. Instead, it adjusts its sensitivity dynamically, based on the magnitude of the error. This makes it a potentially more nuanced and less invasive approach to handling heavy-tailed data, offering a middle ground between sensitivity to data trends and robustness against extreme values.
>
> **3. Real-world Dataset**
>
> We acknowledge the value of real-world datasets in verifying bandit models. However, finding suitable heavy-tailed online datasets is challenging. While such datasets would benefit the community, we believe that our simulated datasets are sufficient for evaluating our algorithm's performance.

---

> ### Author Response · Authors · 2023-11-28
> **Thanks for your reviews (Cont'd).**
>
> ## Minor comments:
>
> **1.Definition of $R_h(s, a)$:** We will formally define $R_h(s,a)$ in the "Learning protocol" section of Section 3.1.
>
> **2. Derivation of $Q_h^{*}$:**
> This derivation has been done by the paper [1]. Their Proposition 2.3 shows that for any policy $\pi$, $Q_{h}^{\pi}(s, a) = \langle \phi(s, a), w_h^{\pi} \rangle$ where $w_h^{\pi} = \theta_h^* + \int_{\mathcal{S}} V_{h+1}^{\pi}(s’)d \mu_h(s’)$. Given our interest is $\pi^*$, we could simply replace $\pi$ in the above equation with $\pi^*$.
>
> [1] Provably efficient reinforcement learning with linear function approximation, Chi Jin, Zhuoran Yang, Zhaoran Wang, and Michael I Jordan, 2020 COLT, https://arxiv.org/pdf/1907.05388.pdf
>
> **3. $V_1^{\star}$** will be defined at the end of Section 3.1 for completeness.
>
> **4. Lemma D.2 and Lemma D.4** are quite classic in RL literature. Though many papers established similar counterparts of Lemma D.2 and Lemma D.4, the notation or the way of defining the exploration radiuses (i.e. various $\beta$’s) is slightly different. We formally state Lemma D.2 and Lemma D.4 and present their proofs mainly for completeness. We claim little novelty.
>
> **5. Variance Estimation Bounds:** The lower bound $\underline{V}_h^k$ together with the upper bound $\overline{V}_h^k$ is mainly used to accurate the estimation of variance.
>
> To be more specific, in the definition of  $\sigma_{h}^k$ (see the equation (A.5)), both $E_{h,k}$ and $J_{h, k}$ involved these two value functions $\underline{V}_h^k$ and $\overline{V}_h^k$ (see the equation (A.6) and (A.7) respectively).
>
> We highlight the point in the informal presentation of VARA.

---

> > ### Comment · Reviewer_eYei · 2023-11-29
> >
> > Thank you for your detailed response. I am still uncertain about the correctness of using mathematical induction in Lemma 2.2. Suppose that for all $0\leq t\leq T-1$, the event $A_t$ is true with probability 1. The revised part has demonstrated that $A_T$ is true with probability at least $1-2\delta$. It is important to note that the probability has decreased from 1 to $1-2\delta$. This raises the question of whether the induction method was applied correctly. Whether the induction method can be applied with uncertainty?

---

> > > ### Author Response · Authors · 2023-11-29
> > > **Hope we could address your concern.**
> > >
> > > The induction method was applied in an almost sure manner. It has nothing to do with the uncertainty $\delta$.
> > >
> > > ============================================
> > >
> > > Let $B$ denote the event that Lemma C.1 and Lemma C.2 are true.
> > >
> > > In other words, on the event $B$, FOR ANY $T \ge$ 1, we have the following two inequalities,
> > >
> > > $(*) \sum_{t=1}^T \frac{2 \tau_t z_t^{\star} 1_{A_{t-1}} }{ \sqrt{\tau_t^2+\left(z_t^{\star}\right)^2} } \frac{1}{1+w_t^2} \frac{ d_{t-1}^T H_{t-1}^{-1} {\phi}_t}{\sigma_t} \le \frac{\alpha_T^2}{2},$
> > >
> > > $(**) \sum_{t=1}^T \frac{\tau_t^2\left(z_t^{\star}\right)^2}{\tau_t^2+\left(z_t^{\star}\right)^2} \frac{w_t^2}{1+w_t^2} \le\frac{\alpha_T^2}{2},$.
> > >
> > > Please keep in mind that the last two inequalities are true for ANY $T \ge$ 1, which is important for our induction.
> > >
> > > The mathematical induction tries to prove $B \subset C$ where $C$ is the event defined by $C = { \forall T \ge 1,\| d_T\|_{H_T^{-1}}^2 \le \alpha_T^2\}$
> > >
> > > As you can see, it has nothing to do with the uncertainty $\delta$.
> > >
> > > By Lemma C.1 and Lemma C.2, we know that $\mathbb{P}(B) \ge 1- 2\delta$. As a result, we would have $\mathbb{P}(C) \ge \mathbb{P}(B) \ge 1- 2\delta$.
> > >
> > > ============================================
> > >
> > > The added argument on Page 30 is trying to establish that $B \subset C$. Let's explain them in more detailed in the following.
> > >
> > > The first inequality is the following inequality:
> > >
> > > $\| d_T\|_{H_T^{-1}}^2 \le $
> > >
> > > $\sum_{t=1}^T \frac{2 \tau_t z_t^{\star} }{ \sqrt{\tau_t^2+\left(z_t^{\star}\right)^2} } \frac{1}{1+w_t^2} \frac{ d_{t-1}^T H_{t-1}^{-1} {\phi}_t}{\sigma_t} $
> > >
> > > $+ \sum_{t=1}^T \frac{\tau_t^2\left(z_t^{\star}\right)^2}{\tau_t^2+\left(z_t^{\star}\right)^2} \frac{w_t^2}{1+w_t^2}.
> > > $
> > >
> > > This inequality is always true for ANY $T \ge1$ because it simply results from a direct calculation.
> > >
> > > The second equality uses the induction condition so that
> > >
> > > $\sum_{t=1}^T \frac{2 \tau_t z_t^{\star} }{ \sqrt{\tau_t^2+\left(z_t^{\star}\right)^2} } \frac{1}{1+w_t^2} \frac{ d_{t-1}^T H_{t-1}^{-1} {\phi}_t}{\sigma_t}$
> > >
> > > =
> > > $\sum_{t=1}^T \frac{2 \tau_t z_t^{\star} 1_{A_{t-1}} }{ \sqrt{\tau_t^2+\left(z_t^{\star}\right)^2} } \frac{1}{1+w_t^2} \frac{ d_{t-1}^T H_{t-1}^{-1} {\phi}_t}{\sigma_t} $.
> > >
> > > This uses the induction condition all $A_{t-1}$ holds for $t \le T$.
> > >
> > > The last inequality uses the fact that we proceed with the proof on the event $B$ so that (*) and (**) are true.
> > >
> > > As you can see, all the arguments above don't manipulate the uncertainty $\delta$. It applies the induction method in an almost sure manner.

---

### Comment · Reviewer_hW9L · 2023-12-22
**Follow-Up Review Comments**

I have read the replies from the authors and the other reviews. I am satisfied with the responses to my questions and comments. I have looked over some of the theory further and, while I still do not understand it deeply, I do not see a problem with the high level reasoning.

In re-reading parts of the paper, I noticed use of the the $\gtrsim$ symbol at the end of Section 2.2. I think it should be a $\ge$ symbol, otherwise I'm not sure what $\gtrsim$ is supposed to mean.

I also noticed that just after the statement of Theorem 2.2, there is a reference to equation 2.3, saying that it shows $\beta_t$ is a hyperbolic function of $\tau_0$ and I do not see how equation 2.3 shows this fact. I think the defintion of $\beta_t$, equation 2.6, was meant to be referenced instead.

---

> ### Author Response · Authors · 2023-12-22
> **Thanks for your feedback.**
>
> Thanks for your feedback. The following are our feedbacks.
>
> 1. $\gtrsim$ could be replaced by $\ge$. Typically, $a \gtrsim b$ means there exists a positive constant $c>0$ such that $a \ge c * b$.
> Because such a constant $c$ is usually universal and unimportant, we don't want to express the dependence explicitly for simplicity.
> As a result, we use $a \gtrsim b$.
>
> At the end of Section 2.2, we want to require that $\tau_0^2  \ge c* d$ for a sufficiently large constant $c$.
>  So we use $\gtrsim$ for simplicity. As we mentioned, it could be replaced by $\ge$ (if we explicitly introduce the positive constant).
>
> 2. Thanks for your pointing out. We will correct it in the next revision.

---

### Decision · Action_Editor_VBte · 2024-03-12

**Recommendation:** Accept with minor revision

**Comment:**

While I am enthusiastic about accepting this work, my excitement is somewhat dampened by it being unclear what the technical challenges were in obtaining these results, in light of previous work. This opinion is shared by at least one reviewer. The authors did mention explaining analytical novelty of the linear MDP results in an appendix, but this is really something that should take center stage for such a theory-focused paper. Also, from my look at the appendix, I saw the author's quickly emphasizing Lemma D.8 but was left wanting a more detailed discussion. In the future, I suggest the authors give proper focus to any novel technical tools/ideas within the main text (you can see well-known RL Theory papers to see how people do this); this might make the difference between an Accept decision versus an Accept decision along with a certification (e.g. Featured Certification in this case). It can also help others adopt your ideas. Still, I believe this paper is definitely worth accepting and congratulate the authors on their impressive work.

For the camera-ready version, please do a full pass over the paper to check for spelling issues and typos. You can run a spellchecker on LaTeX, but that won't fix all issues so you really should carefully look over the full paper (especially the appendix, which is where I found the most issues). There is one place in the paper (page 42) where you wrote "forth" instead of "fourth". Also, for intersections of events where the intersection has a subscript (like $h \in [H]$), use the command \bigcap to achieve the better $\bigcap$ rather than the command \cap which only gives  $\cap$.

**Audience:**

These results are definitely of interest to both the bandits and RL Theory communities. As one example of such interest, as mentioned under "Claims and Evidence", another recent work by different authors compared their work to the present paper.

**Claims And Evidence:**

The claims are well-supported by theoretical evidence, and empirical results are convincing. I now expand on this in detail.

This paper makes contributions both in the setting of linear bandits and the setting of linear MDPs. First, the authors establishes new, variance-dependent regret bounds for linear bandits in heavy-tailed situations using their algorithm AdaOFUL (where here, heavy-tailed means that the second moment is bounded but higher moments might fail to be bounded). Previous work had shown variance-dependent regret bounds either under sub-Gaussian conditions (including the special case of bounded rewards), or had shown less adaptive (not variance-dependent in the sense of the authors) bounds for heavy-tailed situations. AdaOFUL exhibits excellent empirical performance compared to competing algorithms. In the linear MDP setting, I believe the authors’ results (again, variance-dependent regret bounds in heavy-tailed situations) are the first of their kind.